# Phenotypic plasticity and genetic control in colorectal cancer evolution

Jacob Househam[1,2,11], Timon Heide[1,3,11], George D. Cresswell[1], Inmaculada Spiteri[1], Chris Kimberley[2], Luis Zapata[1], Claire Lynn[1], Chela James[1], Maximilian Mossner[1,2], Javier Fernandez-Mateos[1], Alessandro Vinceti[3], Ann-Marie Baker[1,2], Calum Gabbutt[1,2], Alison Berner[2], Melissa Schmidt[2], Bingjie Chen[1], Eszter Lakatos[1,2], Vinaya Gunasri[1,2], Daniel Nichol[1], Helena Costa[4], Miriam Mitchinson[5], Daniele Ramazzotti[6], Benjamin Werner[2], Francesco Iorio[3], Marnix Jansen[4], Giulio Caravagna[1,7], Chris P. Barnes[8], Darryl Shibata[9], John Bridgewater[4], Manuel Rodriguez-Justo[4], Luca Magnani[10], Andrea Sottoriva[1,3 ✉] & Trevor A. Graham[1,2 ✉]

Genetic and epigenetic variation, together with transcriptional plasticity, contribute to intratumour heterogeneity[1]. The interplay of these biological processes and their respective contributions to tumour evolution remain unknown. Here we show that intratumour genetic ancestry only infrequently affects gene expression traits and subclonal evolution in colorectal cancer (CRC). Using spatially resolved paired whole-genome and transcriptome sequencing, we find that the majority of intratumour variation in gene expression is not strongly heritable but rather 'plastic'. Somatic expression quantitative trait loci analysis identified a number of putative genetic controls of expression by *cis*-acting coding and non-coding mutations, the majority of which were clonal within a tumour, alongside frequent structural alterations. Consistently, computational inference on the spatial patterning of tumour phylogenies finds that a considerable proportion of CRCs did not show evidence of subclonal selection, with only a subset of putative genetic drivers associated with subclone expansions. Spatial intermixing of clones is common, with some tumours growing exponentially and others only at the periphery. Together, our data suggest that most genetic intratumour variation in CRC has no major phenotypic consequence and that transcriptional plasticity is, instead, widespread within a tumour.

Genetic intratumour heterogeneity (gITH) is an inevitable consequence of tumour evolution[2]. Extensive gITH has been documented across human cancer types[1], and its precise pattern within an individual cancer is a direct consequence of the evolutionary dynamics driving the development of the tumour[3]. Consequently, clones that undergo positive, negative or neutral selection can be identified through analysis of gITH[4–6]. However, clonal selection in cancer operates on the phenotypic characteristics of a cell—for example, the ability of a cancer cell to evade predation by the immune system[7] or to survive in oxygen-poor environments[8,9], and can be modulated by spatial competition[9–14]. Knowledge of the genotype–phenotype map of cancer cells is limited and thus, while genomics offers us a window into determination of which clones are selected, the methodology provides limited information on precisely why they are selected. Interrelatedly, the extent to which subclonal mutations in tumours lead to phenotypic change is unclear.

RNA sequencing (RNA-seq) enables high-throughput profiling of phenotypic characteristics of cancer cells by quantitative measurement of gene expression levels[15]. Historically, studies have focused on intertumour differences in gene expression patterns and have led to the identification of gene expression signatures that correlate with clinical outcomes. In colorectal cancer (CRC), the focus of this study, consensus molecular subtypes (CMS)[16] or cancer cell-intrinsic gene expression subtypes (CRIS)[17] exemplify this approach. Because the transcriptome is a feature of the cancer cell phenotype, it is natural to view changes in expression, and the pattern of transcriptomic intratumour heterogeneity (tITH), as 'functional' and the substrate for tumour evolution. Potentially tITH could be driven entirely by underlying heritable (epi)genetic variation that evolves during tumour growth. However, the observation that local invasion is polyclonal in both CRC[18] and early breast cancer[19] challenges the notion that cancer cell phenotype (here, the ability to invade) is driven solely by the accrual of genetic mutations. Furthermore, observations of rapid transcriptional shifts following treatment (for example, in melanoma[20]) and, in CRC, variation in

[1]Centre for Evolution and Cancer, The Institute of Cancer Research, London, UK. [2]Centre for Genomics and Computational Biology, Barts Cancer Institute, Queen Mary University of London, London, UK. [3]Computational Biology Research Centre, Human Technopole, Milan, Italy. [4]UCL Cancer Institute, University College London, London, UK. [5]Histopathology Department, University College London Hospitals NHS Foundation Trust, London, UK. [6]Department of Medicine and Surgery, University of Milano-Bicocca, Milan, Italy. [7]Department of Mathematics and Geosciences, University of Trieste, Trieste, Italy. [8]Department of Cell and Developmental Biology, University College London, London, UK. [9]Department of Pathology, University of Southern California Keck School of Medicine, Los Angeles, CA, USA. [10]Department of Surgery and Cancer, Imperial College London, London, UK. [11]These authors contributed equally: Jacob Househam, Timon Heide. ✉e-mail: andrea.sottoriva@fht.org; trevor.graham@icr.ac.uk

subclone proliferation rates through serial retransplantation despite largely stable patterns of genetic alterations[21], discount the notion that transcriptomic phenotypes are determined solely by clonal replacement. It has previously been determined that most driver mutations are clonal in metastatic CRC, meaning that intratumoral transcriptional variation often happens in the absence of the acquisition of new key driver mutations[22]. Collectively, these studies suggest that phenotypic characteristics are at least partially plastic—they can vary without acquiring a new heritable (epi)genetic alteration to drive expression changes, for instance as a response to the cellular environment. In patient samples we cannot measure longitudinally the exact same clones or cells, and so here we define a trait as plastic if it varies independently of evolutionary history. Conversely, non-plastic traits are fixed through tumour evolution.

Here we analyse spatially resolved paired genomic (whole-genome sequencing), epigenomic (assay for transposase-accessible chromatin using sequencing, or ATAC-seq), and transcriptomic (whole-transcript RNA-seq) profiling, coupled with computational modelling, to characterize the evolution of phenotypic heterogeneity in CRC. Paired DNA–RNA data enable assessment of the interrelationship between genetic evolution and gene expression patterns, and of the functional consequence of gene expression change for cancer evolution.

We analysed our spatially resolved, multiomic, single-gland profiling dataset from primary CRCs[23] that were part of our Evolutionary Predictions in Colorectal Cancer (EPICC) study. Single-gland profiling allowed multimodal DNA, chromatin and RNA characterization of the same small clonal unit of tissue (glands or crypts). We focused our analysis on 297 samples from 27 CRCs (mean, 11 samples per tumour; range, 1–38) in which we had obtained high-quality, full-transcript RNA-seq data. Paired deep and shallow whole-genome sequencing and chromatin accessibility analysis by ATAC-seq were available for a subset of these samples. An analysis of the ATAC-seq data is available in the associated paper[23].

## Expression heterogeneity in CRC

First, we explored the heterogeneity of gene expression within and between CRCs. We clustered a filtered set of 11,401 genes (including removal of very lowly expressed genes and those significantly negatively correlated with purity; Methods) using both the mean and variance of gene expression within each tumour (Fig. 1a), and separated the dendrogram into four groups (Methods): group 1 had high average expression and relatively low variance in gene expression ('highly expressed, limited heterogeneity'); groups 2 and 3 had progressively lower average gene expression and high variance in expression, whereas group 4 genes had low average gene expression and low variability between samples from the same tumour (Fig. 1b,c and Supplementary Table 1). Meta-pathway analysis showed weak, non-significant enrichment for pathways involved in cell growth and death in group 1, and significant enrichment for cancer-related genes in group 2 and pathways related to replication and repair in group 3 (Fig. 1d). Group 4 was weakly and non-significantly enriched for signalling pathways but, due to generally low expression, it was excluded from further analyses. We confirmed that transcriptional heterogeneity evident in group 2 genes in tumours was less prominent in an equivalent analysis of normal colon single-cell RNA-seq (scRNA-seq) data, thus excluding the possibility that the gene expression variation we observed was simply the natural transcriptional noise of colon cells (Methods and Supplementary Figs. 1 and 2).

We repeated the clustering analysis using hallmark pathways[24] (Methods) rather than individual genes (Extended Data Fig. 1a), and separated the dendrogram into four groups of pathways based on the degree and heterogeneity of enrichment score within and between cancers, respectively (Extended Data Fig. 1b,c). Hallmark pathways were grouped into 'classes' according to their biological mechanism (oncogenic, immune, stromal and so on)[25]. Homogeneously enriched pathways (pathway group 1) showed moderate but not significant enrichment for cellular stress response; heterogeneously enriched pathways (pathway group 2) were moderately but not significantly enriched for oncogenic signalling (Extended Data Fig. 1d), congruent with the gene-level result. Pathway group 4 (low average pathway enrichment and high heterogeneity) contained two pathways, epithelial–mesenchymal transition and angiogenesis; these were both classed as stromal, meaning that pathway group 4 was enriched for stroma-related pathways (Extended Data Fig. 1d).

Consensus molecular subtypes[16] and CRIS[17] are useful approaches in classification of CRC by gene expression patterns. We investigated the intratumour heterogeneity of these classifiers. For CMS, only 2 out of 17 tumours with sufficient samples for analysis were homogeneously classified (both CMS3; Extended Data Fig. 2a). For CRIS, only a single tumour was homogeneously classified (CRIS-A; Extended Data Fig. 2b). CRIS classification exhibited higher intratumour expression heterogeneity than CMS (Extended Data Fig. 2a,b), and heterogeneity remained when the analysis was limited to only those samples that could be subtyped with high accuracy (Extended Data Fig. 2e–h). Correspondence between CRIS and CMS type calls was weak (Extended Data Fig. 2c). We note that others have published data showing the heterogeneity of molecular subtypes in CRC[26,27] and the discordance between CRIS and CMS classifications[17,28]. The genes used for both CMS and CRIS classification were depleted for highly homogeneously expressed genes (group 1; Extended Data Fig. 2d). Consequently, both CRIS and CMS classifiers exhibited extensive ITH.

Together, these analyses showed that gene expression programmes that define cancer cell biology and interactions with the surrounding tumour microenvironment were not uniformly expressed across CRCs.

## Evolution of expression heterogeneity

We sought to understand the genetic determinants of the observed tITH. If variability in gene expression was caused by genetic change within the tumour (that is, if tITH is caused by gITH), then gene expression variability should mirror genetic ancestry. Phylogenetic signal is a statistical method derived from evolutionary biology that measures the degree to which phenotypic (dis)similarity between species is explained by genetic ancestry, and can be quantified by Pagel's $\lambda$ statistic[29,30] (Supplementary Fig. 3). We assessed the phylogenetic signal of gene expression heterogeneity in each of our CRCs with sufficient paired RNA-seq whole-genome sequencing (WGS) data (114 samples from eight tumours; median 11 samples per tumour, range 6–31). Phylogenetic trees for each tumour were constructed from WGS data (Methods) and terminal nodes overlaid with gene expression profiles (Fig. 1e,f and Extended Data Fig. 3). Pagel's $\lambda$ was computed for 8,368 genes from groups 1–3 (as defined in Fig. 1a), with group 4 genes removed due to low average expression. Within each tumour a median of 166 genes (range 67–2,335) had expression levels with detectable phylogenetic signal ($P < 0.05$), though with the exception of cancer C559 no associations remained after multiple testing correction. The number of genes with phylogenetic signal (at $P < 0.05$) did not significantly correlate with the number of samples per tumour ($P = 0.25$; Supplementary Fig. 4). The above analyses were rerun using standard log-normalization of gene expression and there was a high overlap between genes with evidence of phylogenetic signal, indicating that the normalization method has a negligible impact on results (Supplementary Fig. 5). Adjustment of expression for tumour content (purity) before running phylogenetic signal analysis was also found to have a minimal impact on results (Methods and Supplementary Fig. 6). Post hoc power analysis indicated that our dataset was sufficiently sized to enable detection of the heritability of early subclonal, large-effect changes in gene expression (Supplementary Fig. 7); the expression of most genes did not show this pattern of heritability.

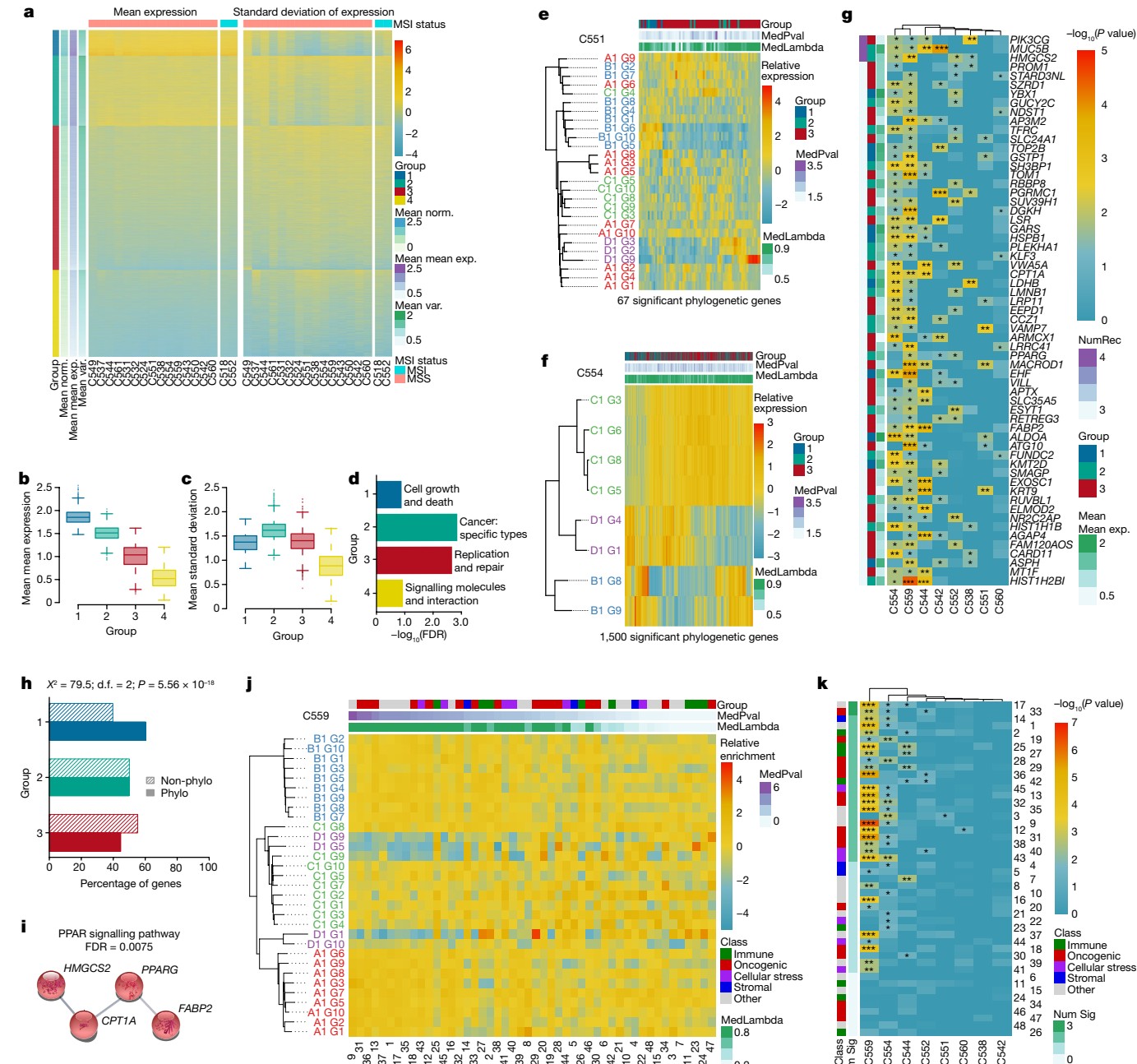

**Fig. 1 | Heterogeneity of gene expression and phylogenetic signal in CRC.**
**a**, Heatmaps showing clustering of genes by expression level across tumours (left) and expression variation within tumours (right). Hierarchical clustering showed four distinct groups, groups 1–4. Units are scaled by column in each heatmap. **b**,**c**, Summary box plots per gene group (group 1, 891 genes; group 2, 2,444 genes; group 3, 5,033 genes; group 4, 3,033 genes). Mean expression level (**b**) and intratumour heterogeneity of expression (**c**) per group, as measured by s.d. **d**, Meta-KEGG pathway analysis showing which pathway categories are most over-represented in each group (after removal of 'infectious disease: bacterial' and 'neurodegenerative disease'—most significant in group 1). **e**,**f**, Phylogenetic trees and heatmaps of genes with evidence of phylogenetic signal (at $P < 0.05$) for tumours C551 (**e**) and C554 (**f**). **g**, Heatmap of genes with recurrent phylogenetic signal across tumours (those which were found to have evidence of phylogenetic signal in at least three tumours). **h**, Results of chi-squared test showing whether gene groups were enriched for phylogenetic genes (those with evidence of phylogenetic signal in at least one tumour—"Phylo") compared to all other genes ("Non-phylo"). **i**, Enrichment of KEGG PPAR signalling pathway for recurrently phylogenetic genes. **j**, Example phylogenetic tree and pathway enrichment heatmap for tumour C559. Pathways are ordered by decreasing significance of phylogenetic signal. **k**, Heatmap showing recurrence of phylogenetic signal of pathways across tumours. Pathways are ordered by decreasing recurrence. Refer to pathway key in Extended Data Fig. 4 for pathway names. *$P < 0.05$, **$P < 0.01$, ***$P < 0.001$; Mean norm., mean gene expression in normal samples; Mean mean exp., mean of mean gene expression per tumour; Mean var., mean standard deviation of gene expression; MedPval, median $P$-value from forest of 100 trees; MedLambda, median $\lambda$ value from forest of 100 trees; NumRec, number of tumours in which gene has evidence of phylogenetic signal; Num Sig, number of tumours in which pathway has evidence of phylogenetic signal; d.f., degrees of freedom.

Only 61 genes had expression patterns that recurrently mirrored phylogenetic ancestry in at least three tumours (Fig. 1g). Group 1 genes (highly expressed, limited heterogeneity) were enriched for phylogenetic signal whereas group 3 genes (moderately expressed, moderate heterogeneity) were significantly depleted for phylogenetic signal (Fig. 1h). Interestingly, the Kyoto Encyclopedia of Genes and

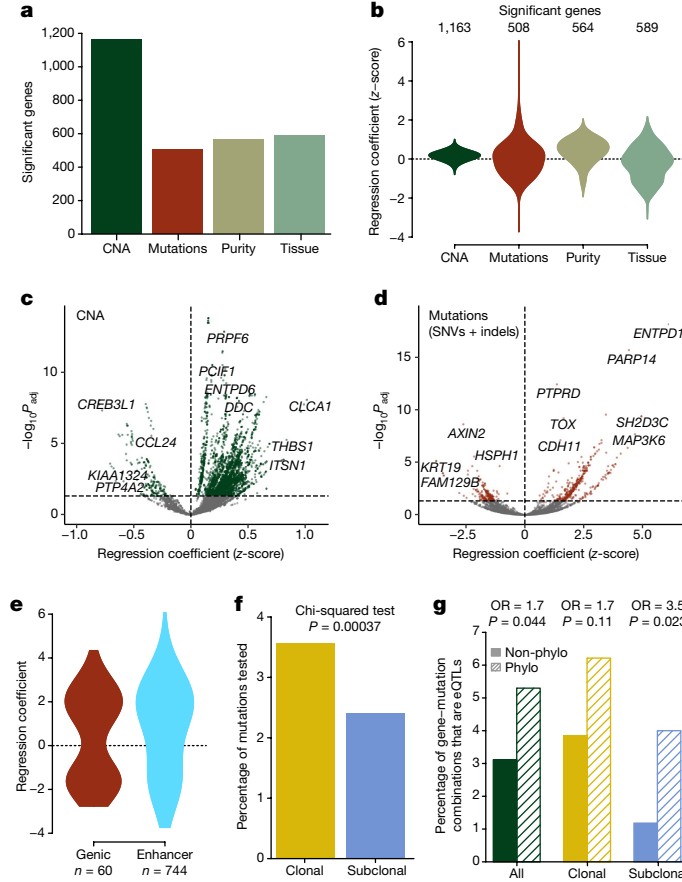

**Fig. 2 | Genetic control of expression with eQTL. a**, The number of genes with significant models for each data type. **b**, Distribution of regression coefficients (effect sizes) for each data type. **c**,**d**, Volcano plots highlighting selected genes significant for SCNA (**c**) and Mut eQTLs (**d**) (linear regression two-sided *t*-tests; $P_{adj}$, FDR-adjusted *P* values). **e**, In comparison with non-synonymous mutations (NS), enhancer (Enh) mutations tended to have large effect sizes and a higher proportion of positive effect sizes. **f**, The proportion of subclonal mutations associated with detectable changes in *cis* gene expression was significantly lower than for clonal eQTL mutations. **g**, Visualization of Fisher's exact tests showing that gene–mutation combinations were more likely to be eQTLs if they were associated with recurrent phylogenetic genes (genes found to have evidence of phylogenetic signal in at least three tumours) for subclonal mutations, and that this was not significant for clonal mutations. Phylo and Non-phylo indicate whether a gene had evidence of phylogenetic signal in the tumour in which the mutation was present. Two-sided Fisher's exact tests, *P* values not corrected for multiple testing.

Genomes (KEGG) pathway peroxisome proliferator-activated receptor (PPAR) signalling, involved in prostaglandin and fatty acid metabolism[31] was statistically over-represented in this recurrently phylogenetic set of genes (false discovery rate (FDR) = 0.0075, STRINGdb analysis; Fig. 1i). Links between PPAR metabolism and CRC have previously been reported[32,33].

Analogous assessment of phylogenetic signal at the level of gene expression pathways (Fig. 1j and Extended Data Fig. 4; at *P* < 0.05, only cancer C559 showed associations after correction for multiple testing) showed two pathways with recurrent evidence of phylogenetic signal in at least three tumours: (1) fatty acid metabolism, related to the PPAR signalling pathway, which was identified in the gene-level analysis, and (2) MYC_TARGETS_V2 that contains genes regulated by MYC signalling (Fig. 1k). Phylogenetic signal at pathway level was not related to pathway class (as used in Extended Data Fig. 1a,d). Thus, in our dataset, the expression of most pathways was not strongly related to genetic ancestry.

We defined phenotypic plasticity as gene expression changes that occurred independently of evolutionary history, possibly as a consequence of external stimulus from the tumour microenvironment. To examine this, phylogenetic trees and expression-based dendrograms were compared, showing few instances in which genetic history mirrored current levels of gene expression (Extended Data Fig. 5 and Supplementary Fig. 8). Across the cohort, the level of genetic intermixing of clones across tumour spatial regions was uncorrelated with the level of gene expression heterogeneity between regions (Supplementary Fig. 9). To specifically examine the influence of tumour microenvironment, we tested whether gene expression of tumour glands was clustered by tumour region (Supplementary Fig. 10), observing significant clustering in 4 of 11 tumours (FDR < 0.05; Methods and Supplementary Fig. 11). We used CIBERSORTx[34] to quantify immune cell infiltration in our samples and tested for association between the degree of infiltration and overall difference in gene expression, finding a significant but weak association ($R^2$ = 0.21; Methods and Supplementary Fig. 12), with the caveat that there is inherent uncertainty in RNA-seq deconvolution in general. Together, in support of previous research studying how the microenvironment can determine gene expression[35,36], these analyses provided evidence that the tumour microenvironment could influence plastic gene expression programmes in tumour cells irrespective of accrued genetic changes in those cells.

## Genetic determinants of gene expression

Somatic mutations altering gene expression are a potential mechanistic explanation of phylogenetic signal. We used a simple linear regression framework (Methods), inspired by the expression quantitative trait loci (eQTL) used in human population genetics[37], to detect *cis* associations between inter- and intratumour somatic genetic heterogeneity and gene expression.

In total, 5,927 genes had *cis* somatic genetic variation in at least two samples (*n* = 167 samples with matched RNA-seq and WGS data and at least two samples per tumour), comprising *n* = 2,422 non-synonymous genic mutations (mutations were single nucleotide variation (SNVs) or indels), *n* = 20,790 non-genic (enhancer) mutations and extensive somatic copy number alterations (SCNAs). Of these genes, 1,529 (25.8%) had expression significantly correlated with inter- or intratumour somatic genetic variation (including both mutations and copy number alterations; FDR < 0.01, Storey's $\pi$ = 0.1007; Fig. 2a and Supplementary Table 2), which we termed eQTL genes. A higher FDR cut-off of 10% was assessed, but this had only a negligible impact on results.

Somatic copy number alterations contributed to expression changes of 1,163 out 1,529 (76.1%) eQTL genes (Fig. 2b,c and Supplementary Table 2), but the magnitude of the effect on expression was generally small (Fig. 2b; median effect size 0.30 s.d. in expression change per allele copy). A positive correlation between copy number and expression was observed for 1,082 genes but, interestingly, a negative correlation was observed for 81. Positive correlations were enriched at loci with total copy number one and four (Supplementary Fig. 13a,d) whereas negative correlations were disproportionately more common at genes with total copy number two or three (copy number two includes cases with copy-neutral loss of heterozygosity, copy number three includes unbalanced gains; Supplementary Fig. 13b,c). Consequently, we speculate that negative correlations between copy number and expression are due to dominant-negative activity of the amplified allele. We note that this idea is consistent with cell line research which found that single-chromosomal gains can function as tumour suppressors[38].

Mutations, both coding and non-coding, were associated with gene expression variation in 508 eQTL genes (Fig. 2b,d) and, typically, the magnitude of the association was much greater than for SCNAs (mean effect size 1.92 versus 0.30 s.d. for mutation versus single-copy number change; Fig. 2b). For coding somatic mutations, approximately equal numbers of mutations associated with an increase versus decrease

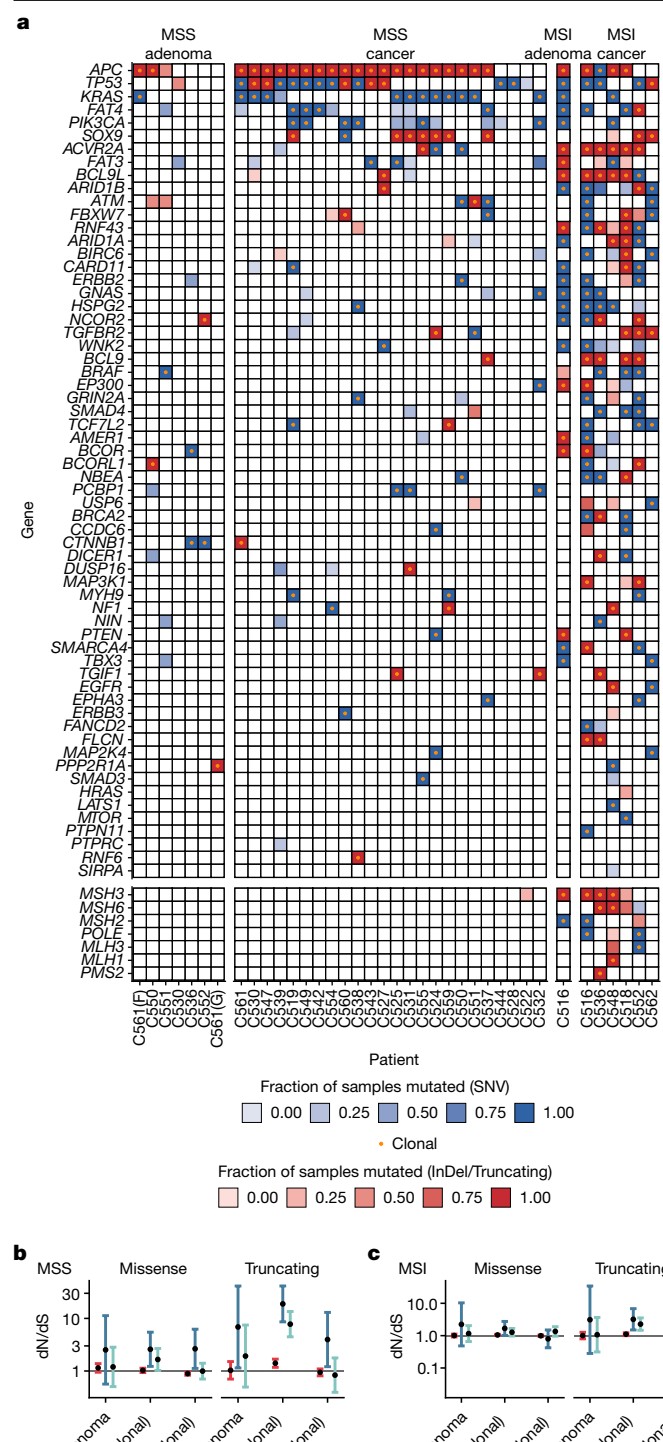

**Fig. 3 | Phylogenetic driver analysis. a**, Non-synonymous somatic mutations and indels in IntOGen CRC driver genes, with clonality status indicated. **b,c**, dN/dS analysis of clonal versus subclonal driver gene mutations, divided between MSS (**b**) (7 adenomas and 24 cancers) and MSI (**c**) (1 advanced adenoma and 6 cancers). Error bars indicate 95% confidence intervals. MMR, mismatch repair (genes).

in expression were observed (33 coding mutations with increasing expression versus 27 with decreasing expression, $P = 0.4$). Non-coding enhancer somatic mutations were associated with the greatest changes in gene expression observed in our cohort, and were more likely to

be associated with increases in expression (486 increases versus 258 decreases, $P = 6.3 \times 10^{-17}$; Fig. 2e). The expression of 175 genes was significantly associated with both SCNAs and mutations, indicating how the combination of somatic mutation and copy number alterations can potentially determine the gene expression phenotype of cancer cells. We used the Hartwig metastatic CRC cohort[39] to validate eQTL results: 22 eQTL mutations had sufficient variation present in the Hartwig cohort to detect associations of the magnitude observed in our cohort and, of these, 9 (41%) were validated (Supplementary Fig. 14). The unexplained gene expression variation for the remaining 13 variants could be due to germline, *trans* or other epigenetic effects. A post hoc power analysis found that we were powered to detect (that is, at least 80% power) effect sizes greater than 0.94 (s.d. in expression change; Supplementary Fig. 15). Assessment of germline SNPs showed some outliers that may have had a small impact on our eQTL analysis, and this could possibly be due to variations in patient genetic ancestry (Methods and Supplementary Fig. 16). With this in mind, and because we did not examine *trans* effects, we emphasize that eQTLs are only associations and not proof of a mechanistic link. In a separate subgroup analysis of mutations in microsatellite stability (MSS) versus microsatellite instability (MSI) cases, mutations in MSS tumours were more frequently associated with large effects on gene expression (Supplementary Fig. 17) whereas the addition of MSI status as a cofactor had minimal impact on tumour eQTL associations (correlation of $R^2$ values between original and MSI-added analysis, $P < 1.1 \times 10^{-16}$, $R^2 = 0.855$; Supplementary Figs. 18 and 19).

Overall, only 2.4% (89 out of 3,705) of subclonal mutations in which eQTL status could be investigated were associated with detectable changes in *cis* gene expression, compared with 3.6% (688 out of 19,256)—many more in absolute numbers—of clonal eQTL variants ($P = 3.7 \times 10^{-4}$; Fig. 2f). Genes associated with subclonal eQTL mutations were enriched for phylogenetic signal (odds ratio (OR) = 3.5, $P = 0.02$; Fig. 2g), and this significant enrichment was absent for genes associated with clonal mutations (OR = 1.7, $P = 0.11$; Fig. 2g). Thus, whereas most somatic mutations did not result in a detectably large direct change in *cis* gene expression, each tumour contained a small number of subclonal genetic variants (median 1) significantly associated with altered gene expression. We emphasize that finding variants associated with gene expression changes does not necessarily imply that those variants underwent selection within the tumour.

## Selection on cancer driver mutations

Cancer genomics studies have established that only a few genes actually contribute directly to cancer evolution, and these genes are termed drivers[40]. We therefore focused on understanding the evolutionary consequences of putative CRC driver mutations on tumour expansion.

We used our extensive single-gland, multi-region WGS data (deep WGS, median depth 35×, between 3 and 15 samples per patient (median, 8) and low-pass WGS (median depth 1.2×, between 1 and 22 samples per patient, median 8) for accurate identification of clonal and subclonal somatic variants (https://doi.org/10.6084/m9.figshare.19849138 from ref. [23]) and to call somatic copy number alterations in each tumour (note that this included additional tumours lacking RNA-seq data). We specifically examined the clonality of 69 genes (excluding *PARP4*, *LRP1B* and *KMT2C*, which we excluded due to a high number of false-positive low-frequency variants in these genes) on the IntOGen list[41] of putative CRC driver genes (Methods and Fig. 3a). The most frequently mutated drivers in colorectal cancer, such as *APC*, *KRAS*, *TP53* and *SOX9*, as well as other known drivers including *PTEN*, *EGFR*, *CCDC6*, *PCBP1*, *ATM* and *CTNNB1*, were invariably clonal in cancers, except for one tumour with a subclonal *KRAS* mutation and another with a subclonal *TP53* mutation. These findings are consistent with previous multi-region sequencing studies[42] but contradict claims of frequent subclonality of these genes in single-sample bulk data[43],

highlighting the need for methods to identify functional intratumour heterogeneity[44].

We used analysis of the ratio of non-synonymous to synonymous substitutions (dN/dS)[45], which quantifies the excess of non-synonymous mutations in a gene, to detect selection across the complete set of cancer drivers (Methods). We found clear evidence of positive selection (dN/dS greater than 1) for clonal missense and truncating mutations in IntOGen driver genes in MSS cancers (Fig. 3b), and dN/dS values were higher for the IntOGen list than for a second, pan-cancer driver list[45], confirming that the IntOGen list was enriched for true CRC drivers. For subclonal variants, we found evidence of subclonal selection of truncating variants and missense mutations with dN/dS higher than 1 for CRC-specific IntOGen variants but not for the pan-cancer driver list[45], suggesting that a subset of putative subclonal CRC driver mutations were under positive selection in growing tumours. For MSI tumours, subclonal selection was less evident from dN/dS, probably due to the higher mutation rate generating a much larger number of neutral mutations in cancer driver genes and thus diluting the dN/dS signal but, nevertheless, selection for clonal missense and truncating mutations was significant in MSI cancers (Fig. 3c). We then examined dN/dS values for each of the IntOGen driver genes in a larger dataset, combining our data with The Cancer Genome Atlas (TCGA) colon and rectal cancer cohorts and additional data[46,47] ($n = 1,253$ CRCs). Most genes in the list showed no evidence of selection, with the majority of the top significant genes being the 'usual suspects' in CRC drivers[42] (Extended Data Fig. 6).

For an orthogonal assessment of driver gene function we turned to the DepMap dataset[48] that assesses the functional consequence of gene knockouts across a large panel of cell lines (Methods). Most CRC candidate drivers showed no evidence of essentiality (a measure of cell viability following gene perturbation) across the CRC cell lines of the DepMap dataset, whereas the two most likely under strong selection in our cohort, KRAS and PIK3CA, were significantly essential in many CRC cell lines and were found to be significantly differentially essential when contrasting mutant versus wild-type (WT) CRC cell lines (Student's $t$-test $P < 10^{-6}$; Supplementary Fig. 20).

Thus, surprisingly, these analyses indicated that even putative driver mutations in CRCs sometimes have limited phenotypic consequence when measured in terms of subclonal selection. The lack of detectable selection on CRC driver mutations is consistent with previous reports of widespread neutral subclonal evolution within CRCs[5,49,50].

## Evolutionary dynamics within tumours

We assessed the evolutionary dynamics of individual driver mutations on a tumour-by-tumour basis through assessment of phylogenetic tree shape and the related clonal structure of the tumour (Fig. 4 and Extended Data Fig. 7; Methods). 'Balanced' trees, in which similar branch lengths are found across tumour samples and regions, are consistent with effectively neutral evolution and were observed for a large proportion of tumours. A clear outlier was tumour C539, in which the tree contained a particularly large clade that spanned multiple geographical regions of the tumour (all A and part of B). This 'unbalanced' tree was suggestive of subclonal selection[51], and indeed, the expanded clade contained a KRAS G12C mutation (Fig. 4h). We used BaseScope, a commercial in situ RNA-based mutation detection technique[52] (Methods), to visualize subclones containing a putative driver alteration. We tested the KRAS G12C subclonal variant in C539 (Fig. 4h and Supplementary Fig. 21) and the PIK3CA E545K subclonal variant in C537 (Fig. 4i and Supplementary Fig. 22). This analysis confirmed the spatial segregation of subclones, showing heterogeneity in a subset of the blocks, whereas we also found complete absence of the clone in a large proportion of other areas of the tumour (Supplementary Table 3). Furthermore, and consistent with our previous reports[4,49], tumours could be split into two groups characterized by subclonal intermixing between spatially

distinct regions (16 out of 28, 57% of tumours) versus strict segregation by geography (Supplementary Fig. 23).

We assessed the functional consequence of 38 subclonal putative driver mutations from the IntOGen list that were detected in MSS cancers. PolyPhen[53] scores showed that 8 out of 38 (21%) mutations were putatively benign mutations (marked in grey in Fig. 4 and Extended Data Fig. 7). Paired RNA-seq showed only wild-type reads for 5 out of 38 (13%) putative driver mutations (also marked in grey). We could not assess mutant transcript expression for 25 out of 38 mutations (66%) because of missing RNA-seq data or lack of reads covering the variant location. Of those, 13 out of 25 (52%) were in genes with dN/dS approximately 1 in the TCGA cohorts; COAD and READ. Six out of 38 (16%) variants were identified as deleterious by PolyPhen and were also found to be expressed in matched RNA-seq (marked in bold).

At the individual tumour and mutation level these analyses showed that, of the large number of putative driver events identified in our cohort (Fig. 3a), many showed no evidence of being under selection: 14 out of 38 (37%) variants were either benign or not expressed in the cancer (although we note that expression could not be assessed for two-thirds of variants), and a further 10 out of 38 (26%) variants were in genes with dN/dS of approximately 1 in the external cohorts. However, positive dN/dS values for pooled cases suggested that some of these subclonal variants were under selection. To identify these, we designed a spatial inference framework able to detect and measure subclonal selection in our dataset.

## Spatial inference of growth dynamics

We decided to further probe for evidence of evolutionary consequence of heritable alterations in individual tumours. Computational models allow the simulation of different types of spatial growth dynamics and have provided insights into tumour evolution and the effect of spatial constraints[8–14]. Here we used computational modelling in combination with approximate Bayesian computation (ABC) to infer subclonal selection and the impact of spatial effects from our spatially resolved WGS data. For this, we extended our previous model based on cell replication, death and mutation[51] to incorporate more realistic spatial growth conditions and branch overdispersion (Extended Data Fig. 8a and Methods). We note that we did not specifically model interactions between subclones. We simulated the genome-wide accrual of somatic mutations in each lineage, including both neutral mutations (Extended Data Fig. 8b–d, bottom) and selected (driver) mutations (Extended Data Fig. 8b–d, top), showing characteristic patterns caused by subclonal selection. Furthermore, distinct clonal patterning was observed for peripheral versus exponential growth (governed by the width of the growing outer rim of cells ($d_{push}$); Extended Data Fig. 8e and Supplementary Fig. 24), in which clonal intermixing was greater in the exponential case.

To compare the model with data, we simulated our empirical spatial sampling scheme (Fig. 4a,c,e, ref. [23] and Supplementary Fig. 1) on our virtual tumours (Extended Data Fig. 8f). This generated realistic whole-genome sequencing synthetic data that we used to reconstruct a (synthetic) phylogenetic tree, thus comparing real data (Fig. 5a) and the corresponding matched simulation (Fig. 5b and Extended Data Fig. 8g). The corresponding spatial patterns of subclonal heterogeneity could be visualized from the simulation (Fig. 5c). Bayesian inference (sequential Monte Carlo, or ABC–SMC[54]) of model parameters was performed on a patient-by-patient basis by matching synthetic and empirically observed trees, making use of regularization with the Akaike information criterion (AIC) for model selection[55] (Fig. 5d and Extended Data Fig. 8h; see Methods and Supplementary Note with https://doi.org/10.6084/m9.figshare.20394369 for details). Specifically, the number of parameters ($k$) is used to regularize the negative log-likelihood (NLL) of the models, calculate AIC and, more importantly to estimate the confidence in model selection, the $\Delta$AIC value (difference in AIC between

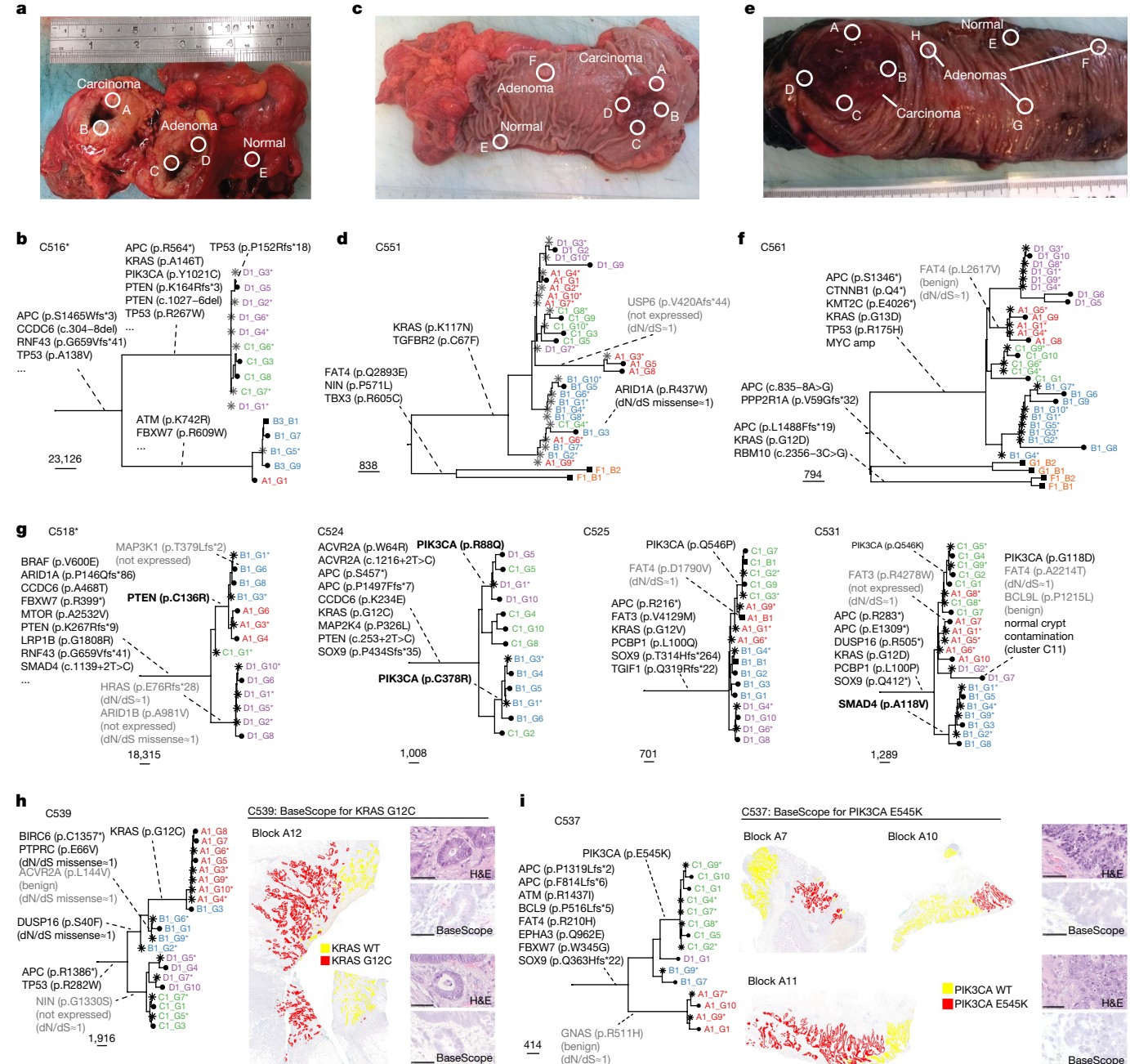

**Fig. 4 | Spatial phylogenomics of colorectal cancer. a**, In this MSI tumour (C516) the cancer (regions A and B) and macroscopically diagnosed advanced adenoma (regions C and D) formed a large mass and were physically adjacent to one another. Photo indicates sampling quadrant, not precise location. **b**, The advanced adenoma shared multiple drivers with the cancer but showed early divergence. **c**, Tumour C551 presented with a cancer and a concomitant adenoma that were very distant, indicating two independent events. **d**, The phylogenetic tree was characterized by clonal intermixing of diverging lineages collocated in the same region (for example, some lineages from regions A, B and C were genetically close). Subclonal drivers of unknown significance were present, including a non-expressed variant in USP6 and

an ARID1A mutation. Early divergence between the cancer and adenoma F was evident, with no shared drivers between the two lesions. **e**, Tumour C561 presented with a large cancer mass and multiple small concomitant adenomas. **f**, Again, there was no notable somatic alteration in common between the different lesions. The cancer showed clonal amplification of MYC and only a benign subclonal mutation in FAT4. **g**, Phylogenetic reconstruction of four further tumours with annotated driver events. **h,i**, Phylogenetic trees with matched in situ mutation detection with BaseScope for the KRAS G12C subclonal variant in C539 (**h**) and the PIK3CA E545K subclonal variant in C537 (**i**). Staining by haematoxylin and eosin (H&E) and BaseScope were each performed once; scale bars, 50 µm.

compared models). ΔAIC greater than 4 is considered to represent strong support for one model over another[55], this was the threshold used to identify strongly preferred models. The relationship between AIC and critical distance of summary statistics between real and simulated trees is reported in Fig. 5e. Generally good agreement between simulated and observed phylogenetic tree structures was observed despite the relative

simplicity of our model, with quantitative assessment of the goodness of fit confirmed by likelihood and posterior predictive *P* value distribution (Fig. 5f). For example, C539 was predicted to contain a selected subclone (Fig. 5a–f) and carried a KRAS G12C mutation that presumably drove the clonal expansion (Fig. 5a). Tumour C548 was inferred to be neutrally evolving (Fig. 5g–l) and thus predicted to carry no strongly selected

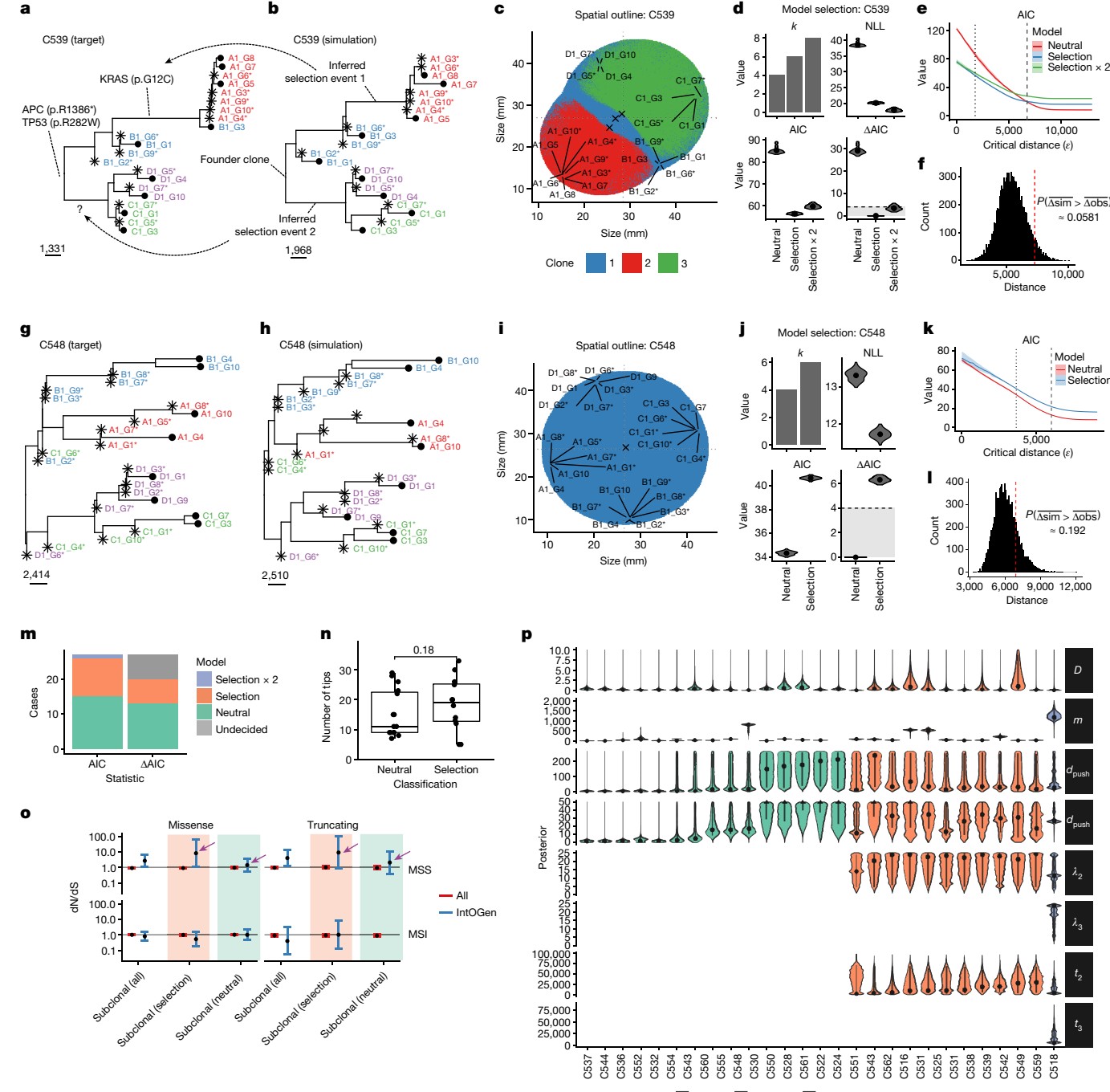

**Fig. 5 | Inference of evolutionary dynamics in individual tumours.**
**a**,**b**, Target tree for C539 (**a**) versus best simulated tree for C539 (**b**). **c**, Spatial patterns and sampling of the simulation. **d**, Model selection considering the number of parameters (*k*) and NLL to calculate AIC. AIC differences (ΔAIC) greater than 4 indicate strong preference of a model. **e**, AIC value with respect to distance from the data (ε) for each of the models. Dotted line indicates final distance of ABC–SMC; dashed line indicates distance of trees with added random uniform noise (0.5–2.0). **f**, Posterior predictive *P* value (one-sided). Dashed line indicates average distance between target and simulated trees. **g**, Target (real) tree for C548. **h**, Simulated tree for C548 identified during the inference. **i**, Spatial patterns of simulation that generated

the data. **j**, Model selection for C548. **k**, AIC value versus ε. **l**, Posterior predictive *P* value (one-sided). **m**, Proportion of instances in which models were selected by model selection. AIC and ΔAIC values are reported, the latter indicating the proportion of tumours that can be explained by both models. **n**, Inference of selection (AIC) was not associated with a higher number of samples per tumour (one-sided bootstrap test, *n* = 15 neutral and *n* = 12 non-neutral). **o**, Subclonal dN/dS values for carcinoma with and without selection (AIC). Numbers of tumours per group: 3 neutral MSI, 3 selected MSI, 12 neutral MSS and 9 selected MSS carcinomas. Error bars indicate 95% confidence intervals. **p**, Marginal posterior distributions of parameters, split by neutral (green), selected (orange) and selected ×2 (purple).

subclonal driver mutations, despite there being subclonal mutations in putative driver genes in this case.

Across the whole cohort (see https://doi.org/10.6084/m9.figshare.20394360 for a supplementary inference result booklet), we found strong

evidence of subclonal selection in 7 out of 27 tumours (ΔAIC greater than 4; Fig. 5m). In four of these seven tumours, a putative subclonal driver mutation was present in the selected clade and the variant was expressed in the RNA (subclone drivers are listed in Supplementary

Table 4 and reported in Figs. 3a and 4 and Extended Data Fig. 7). These included (1) C518, with subclonal selection in A and B driven by PTEN missense mutation C136R; (2) C531, with subclonal selection in B driven by SMAD4 missense mutation A118V; (3) C538, with subclonal selection in D driven by RNF43 nonsense mutation Q153*; and (4) C539, with subclonal selection in A and part of B driven by KRAS missense mutation G12C. In five additional tumours we detected a weak preference for the subclonal selection model. These included (1) C524, in which subclonal selection in B appeared to be driven by a PIK3CA C378R mutation, and (2) C525, in which subclonal selection in C appeared to be driven by a PIK3CA Q546P mutation. The selective advantage of PIK3CA and KRAS mutations agrees with our orthogonal assessment of CRC driver genes using the DepMap database (Supplementary Fig. 20). Evidence of selection in the phylogenetic trees included a significantly longer branch containing the selected event (for example, Fig. 5a, selection event 1), or two distinct regions having a more recent common ancestor with respect to the others (for example, Fig. 5a, selection event 2).

In the remaining 15 out of 27 tumours the preferred subclonal growth model was neutral (Fig. 5m). The number of samples per tumour (that is, more extensive tumour sampling) did not confound model selection (Fig. 5n). Notably, orthogonal dN/dS analysis on the IntOGen driver gene list confirmed the computational modelling results. Specifically, putative subclonal driver gene mutations in tumours predicted to be neutrally evolving showed a dN/dS value of 1 whereas the point estimate was appreciably higher than 1 for driver genes in tumours predicted to experience subclonal selection (Fig. 5o). This also supported the absence of subclonal selection, even in small clades that may not have undergone sufficient expansion to be detectable by our inference method. Aside, these results illustrate that our spatial inference framework could be used for accurate assessment of the evolutionary consequence of putative driver mutations.

Full parameter estimation is reported in Fig. 5p: overdispersion of edge length ($D$), mutation rate per division ($m$), width of the growing outer rim of cells ($d_{push}$), growth rate of the first and second subclones ($\lambda_2$ and $\lambda_3$, respectively) and population size at their introduction ($t_2$ and $t_3$, respectively). The increased growth rate of selected subclones was inferred to be as much as 20 times higher than that of the background clone, and most selected clones originated relatively early during tumour expansion (tumour size fewer than 50,000 cells). Inferred mutation rates were $9.8 \times 10^{-9}$ and $46.6 \times 10^{-9}$ mutations per base pair per division in MSS and MSI tumours, respectively, consistent with previous measurements[56]. Tumours were delineated by either exponentially growing (high $d_{push}$) or growing more slowly at the periphery only (low $d_{push}$). Notably, exponential growth was over-represented in neutrally evolving tumours (Fisher's exact test, $P = 0.022$).

## Epigenome and transcriptome of subclones

Subclone evolution within a cancer is a natural 'competition experiment' between human cells with similar genetic background in the same microenvironment that facilitates delineation of phenotypic differences between subclones and the consequences of driver alterations.

We examined matched ATAC-seq and RNA-seq data from selected subclones versus background clones in six and five, respectively, out of seven tumours with strong selection for which we had sufficient matched 'omics' data. Enrichment analysis of differentially expressed genes between the subclone and background clone highlighted consistent dysregulation of focal adhesion pathways for C531, C542 and C559. The epithelial–mesenchymal transition programme was upregulated in C542 whereas MYC + E2F targets were upregulated in C531 (see Supplementary Fig. 25a for gene-level analysis and Supplementary Fig. 26 for pathway analysis). Analogous analysis of somatic chromatin accessibility alterations showed promoter loss of accessibility of *PPP2R5C*,

a regulator of *TP53* and *ERK* in C542, which had no known genetic driver mutation in the selected clade (Supplementary Fig. 25b).

Finally, we assessed whether heritable changes in gene expression were indicative of subclonal selection. There were eight tumours in which both adequate phylogenetic signal analysis and assessment of subclone selection were possible. There was no association between the number genes with some evidence of phylogenetic signal and the presence of subclone selection (Wilcoxon $P = 0.686$; Supplementary Fig. 27a), nor for spatial segregation versus intermixing of subclones ($P = 0.393$; Supplementary Fig. 27b). Furthermore, the percentage of tested eQTL genes that were significant in each tumour was not associated with neutral evolutionary dynamics ($P = 0.968$; Supplementary Fig. 27c), nor was the magnitude of heritable gene expression changes ($P = 0.195$; Supplementary Fig. 27d). Together this suggests transcriptional variation even within a selected clone. A visual schematic illustrating the main results is shown in Extended Data Fig. 9.

## Discussion

Heterogeneity in gene expression is common, both between and within patients. Leveraging the fact that clone ancestry is encoded by somatic mutations in the genome, here we determined that only a small proportion of the observed subclonal transcriptomic variation shows strong evidence of heritability through tumour evolution (under 1% of expressed genes and under 5% of hallmark pathways). This points towards phenotypic plasticity—the ability of a cancer cell to change phenotype without underlying heritable (epi)genetic change—as a common phenomenon in CRC. We previously considered that the observation of infrequent stringent selection for subclones within CRCs is consistent with the notion that phenotypic plasticity is established within cancer cells at the outset of cancer growth[50]. Here our explicit analysis of transcriptomic variation supports this hypothesis.

Nevertheless, we do find a evidence of heritable changes in gene expression in all CRCs examined. Of 29,949 associations between somatic mutations and gene expression, only 796 (702 clonal) were associated with significant changes in *cis* gene expression and so can be thought of as potentially functional mutations. In any individual tumour we detected a median of 1 (maximum, 34) subclonal mutation that putatively affected gene expression and, notably, the presence of heritable changes in gene expression was not necessarily related to whether the cell lineage with the variant was undergoing subclonal selection. This emphasizes that phenotypic changes do not necessarily correlate with changes in fitness—the newly induced expression of a particular gene may have no relevance to the ability of that cell to survive or grow in its current microenvironment, and indeed across species most genetic 'tinkering' is near neutral or even deleterious[57]. Thus, at least some of the observed tITH is part of the standing phenotypic variation in the tumour but is not selected at the time of the expansion of the primary tumour, even if it is the consequence of the accumulation of mutations during tumour growth. Care should be taken not to conflate transcriptional variation with evidence of important variation in tumour cell biology. We suggest that this variation could partially be a consequence of tumour evolution being 'out of equilibrium', in which an expanding population with high genomic and phenotypic instability generates widespread variation that stabilizing selection has not yet had time to prune. Nevertheless, such variation may be important for future tumour evolution, such as in response to treatment. We emphasize that the limited size of our cohort reduced the power to detect the many small associations between genetics and expression that may occur within tumours, and also means that we were unlikely to observe recurrent events across cancers. Future single cell analyses, rather than the tumour glands used here, are likely to be better powered to reveal DNA–RNA associations[58]. However, we argue that the large effects, which we were generally powered to see, are those most likely to be relevant for tumour biology. We emphasize that our

analysis reports only correlations and is not proof of a mechanistic link, and that there are other potential confounders including patient genetic background, epigenetic effects and unexplored *trans* effects.

Aside from the foregoing, we show that assessment of intratumour heterogeneity can serve as a 'controlled experiment', enabling quantitative measurement of ongoing evolutionary competition within the human body between different lineages with distinct subclonal mutations, providing a platform for function assessment of the 'driverness' of putative driver mutations in vivo in human malignancies. Ongoing collection of associated relapses and metastatic deposits will allow assessment of those subclones and drivers responsible for disease progression.

Our study makes progress in elucidating the role of genetic control and clonal evolution within primary untreated CRC, suggesting that phenotypic plasticity is widespread and underlies pervasive transcriptional heterogeneity.

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

## Methods

### Sample preparation and sequencing

The method of sample collection and processing is described in a companion article (ref. [23]). Sequencing and basic bioinformatic processing of DNA-, RNA- and ATAC-seq data are included there as well.

### Gene expression normalization and filtering

The number of non-ribosomal protein-coding genes on the 23 canonical chromosome pairs used for quality control was 19,671. Raw read counts uniquely assigned to these genes were converted into both transcripts per million (TPM) and variance stabilization transformed (VST) counts via DESeq2 v.1.24.0 (ref. [59]).

A list of expressed genes ($n = 11,667$) was determined by filtering out those for which less than 5% of tumour samples had at least ten TPM. To concentrate on tumour epithelial cell gene expression, genes were further filtered out if they negatively correlated with purity as estimated from matched DNA-seq data (see associated ref. [23] for methodology of purity estimation). Specifically, for the 157 tumour samples that had matched DNA-seq and therefore accurate purity estimates, a linear mixed-effects model of 'expression (VST) ~ purity + (1|patient)' (where '~' represents 'is distributed as') was compared via a chi-squared test to 'expression ~ (1|patient)'. The linear mixed-effects models were built with lmer from the lme4 R package v.1.1-28 (ref. [60]). Genes with a negative coefficient for purity in the first model and FDR-adjusted $P < 0.05$—suggesting that purity had significantly affected expression—were filtered out; this led to a filtered list of 11,401 expressed genes.

### Gene expression clustering

For each tumour with at least five tumour samples ($n = 17$ tumours; note that, except for the large advanced C516 adenoma, adenomas used in ref. [23] did not undergo RNA-seq), mean expression and s.d. of expression were calculated for every filtered expressed gene ($n = 11,401$) using DESeq2 VST normalized counts (inspired by ref. [61]). Euclidean distance matrices of mean expression and s.d. of expression were calculated based on non-MSI tumours. Distance matrices were combined with 'fuse' from the analogue R package v.0.17-6 (ref. [62]) with equal (50/50) weighting, and complete linkage hierarchical clustering was performed. Four gene groups were determined using 'cutree' ($k = 4$) from the dendextend R package v.1.15.2 (ref. [63]). For plotting of Fig. 1a,b, tumours were clustered with the approach described above and both mean expression and s.d. of expression matrices were scaled by columns.

Conversion to entrez gene IDs and gene symbols was carried out in biomaRt v.2.50.3 (ref. [64]) using Ensembl v.90. Where IDs were missing, newer Ensembl versions and manual curation were used (the complete list of gene information is available in Supplementary Table 2).

For the KEGG meta-pathway analysis, pathways and pathway categories were downloaded from https://www.kegg.jp/kegg-bin/show_brite?hsa00001_drug.keg. Enrichment of KEGG pathways for each gene group was determined with enrichKEGG from ClusterProfiler v.4.2.2 (ref. [65]), and pathways enriched at FDR < 0.1 were input into 'enricher' to determine pathway category enrichment (FDR < 0.1). Pathway categories 'Neurodegenerative disease' and 'Infectious disease: bacterial' were removed due to their irrelevance to CRC cell biology.

### Analysis of normal colon scRNA-seq

A scRNA-seq dataset derived from healthy intestine was accessed from Elmentaite et al.[66]. scRNA-seq data for colon gut epithelium were downloaded from https://www.gutcellatlas.org and filtered for cells from the colon in 'Healthy adults'. This left seven donors with a mean of 5,516 cells per donor (range, 1,410–16,828). Expression data were normalized with Seurat v.4.1.0 (ref. [67]) and mean expression within each donor was calculated.

The mean and s.d. of each gene's expression within each donor was calculated. Genes were then filtered and grouped according to the groups identified in Fig. 1a, and plots were produced analogously to Fig. 1b,c.

### Pathway enrichment clustering

Hallmark pathways were downloaded from MSigDB (msigdbr R package v.7.2.1)[24] with unrelated pathways (SPERMATOGENSIS, MYOGENESIS and PANCREAS_BETA_CELLS) removed from analysis, and the COMPLEMENT pathway was renamed COMPLEMENT_INNATE_IMMUNE_SYSTEM. Pathways INTESTINAL_STEM_CELL[68] and WNT_SIGNALING (http://www.gsea-msigdb.org/gsea/msigdb/geneset_page.jsp?geneSetName=WNT_SIGNALING) were added.

For each multi-region tumour ($n = 17$), the TPM expression of protein-coding genes converted to entrez gene IDs ($n = 18,950$) was used as input for single-sample gene set enrichment analysis using the GSVA R package v.1.42.0 (ref. [69]). The mean and s.d. of enrichment were then recorded for each tumour. Because KRAS_SIGNALING_DN had average enrichment below zero it was removed from downstream analysis, leading to a final list of 48 pathways.

Analogously to the genic analysis, mean and s.d. of pathway enrichment were jointly used to determine four groups of pathways whereas tumours were clustered and matrices normalized by column as before. Fisher's exact tests were subsequently performed to determine whether pathway classes[25] were significantly enriched/depleted in particular pathway groups.

CMS and CRIS classifications were determined using the CMScaller R package v.2.0.1 (ref. [70]). As recommended, raw gene counts were used as input with 'RNA-seq=TRUE', meaning that these counts underwent $\log_2$ transformation and quantile normalization. CMS and CRIS were predicted using templates provided in the CMScaller package, and samples were assigned to the subtype with the shortest distance. High-accuracy classifications were determined by running 1,000 permutations, where a classification was considered significant if the FDR-adjusted $P$-value was under 0.05.

### Construction of phylogenetic trees

**Reconstruction of maximum-parsimony trees.** From deep WGS (dWGS) samples, maximum-parsimony trees were reconstructed with the Parsimony Ratchet method[71] implemented in the phangorn R package v.2.8.1 (ref. [72]). Mutations with an estimated cancer cell fraction above 0.25 were considered to be mutated (state 1) and others to be non-mutated (state 0) in a given sample. The ratchet was run for a minimum of 100 and a maximum of $10^6$ iterations, and terminated after 100 rounds without improvement.

The acctran algorithm[72–75] was used to estimate ancestral character states. From these a set of mutations ($M_e$) that were uniquely mutated (that is, state 0 to greater than 1) on each edge $e$ of the phylogeny were obtained.

**Addition of shallow WGS samples to the tree.** For any mutation $i$ the number of reads supporting the variant $y_i$ and the total number of reads covering the locus $n_i$ in a shallow WGS (sWGS) sample were obtained from the bam files.

The mutation data were assumed to follow a binomial (Bin) distribution:

$$y_i \sim \text{Bin}(n_i, p_i),$$

where the success probability $p_i$ is a function of the sample's purity $\rho$, the number of mutated alleles $m_i$ in tumour cells, the total copy number $c_i$ in tumour cells and the copy number in contaminating normal cells, $c_n = 2$, given by

$$p_i = \frac{\rho m_i}{\rho c_i + (1 - \rho_s)c_n} = \frac{\rho m_i}{2 - 2\rho + \rho c_i}.$$

For a set of mutations $M_e$ from a given edge $e$ of a tree $T$, all, none or a fraction $\pi_m$ of mutations might be present in a sample. The marginal likelihood of the observed data $(D_e)$ of the set of mutations is

$$p(D_e|\pi_m) = \prod_{i=0}^{|M_e|} (\pi_m\, p(y_i|n_i, p_i) + (1-\pi_m)\, p(y_i|n_i, p_0)),$$

where $p_0$ is the background noise of the WGS at a unmutated site.

Assuming that mutated sites are not lost at any point in time, for a mutation from the edge $e = (s,t)$ to be mutated in a sample, all variants on the path from the germline node $r$ to the node $s$ of this edge $(r \rightsquigarrow s)$ also have to be mutated (that is, $\pi_m = 1$). All remaining mutations—that is, those that occur in the descendants of $t$ or in different lineages of the tree—must be absent (that is, $\pi_m = 0$). The likelihood of the data $D$ for all mutations that are part of the tree is

$$L(e, \pi_m, p_0, \rho) = p(D_e|\pi_m)\prod_{e' \in \mathrm{Anc}(s)} p(D_{e'}|\pi_m = 1)\prod_{e' \notin \mathrm{Anc}(t)} p(D_{e'}|\pi_m = 0),$$

where $\mathrm{Anc}(s)$ is the set of all ancestral edges on the path from $r$ to $s$. Maximum-likelihood estimates of sample parameters $\hat{e} \in E$, $\pi_m \in [0,1]$, $\hat{p}_0 \in [0,1]$ and $\hat{p} \in [0,1]$ were obtained for each sWGS sample by minimizing $-\log(L)$, and samples were added to location $\hat{x} = (\hat{e}, \hat{\pi}_m)$ of the tree.

**Estimation of copy number multiplicities.** The above analysis was restricted to mutations in regions in which no subclonal SCNA occurred. The multiplicity of mutations $m_{s,i}$ was estimated across the set of all samples $S$ as

$$m_i = \operatorname*{argmin}_{m_{s,j} \in \{1,\dots c_{s,j}\}} \sum_{s \in S} -\log\left(\binom{n_{s,i}}{y_{s,i}} p_{s,i}^{y_{s,i}}(1-p_{s,i})^{n_{s,i}-y_{s,i}}\right)\mathbb{I}_{s,i}$$

with $p_{s,i}$ as defined above and where $\mathbb{I}_{s,i}$ indicates whether the mutation $i$ was detected in sample $s$. Due to potential issues with the accuracy of estimates for large copy numbers, only sites with copy number $0 < c < 4$ were used.

The tool for assignment of sWGS samples to a dWGS tree is available as R package MLLPT at https://github.com/T-Heide/MLLPT.

**Intermixing scores.** To calculate intermixing within tree $T$, each tip $v \in V^1$ was labelled with the region of the tumour from which the corresponding sample was obtained. Intermixing within the tree was then measured as

$$I(T) = \frac{1}{|V^1|}\sum_{v \in V^1}\left(\frac{1}{|D_s|}\sum_{s \in D_s}\mathbb{I}_{m_v \neq m_s}\right), D_s := \{t \in V^1 | t \in \mathrm{desc}(\mathrm{pa}(s))\},$$

where $\mathbb{I}_{m_v \neq m_s}$ is an indicator function that indicates whether $v$ and $s$ had different labels, $\mathrm{pa}(s)$ is the parent of $s$ and $\mathrm{desc}(s)$ is the set of all descendants of $s$.

## Phylogenetic signal analysis

Tumours with fewer than six paired DNA–RNA samples were excluded from this analysis, leaving 114 samples from eight tumours (median 11 samples per tumour, range 6 to 31).

Additional sWGS samples, however, had zero branch length because mutations unique to a sample could not be called with sWGS methodology. To account for these 'missing' unique variants, we inferred the probable number of unique variants from the matched dWGS samples. For each sWGS sample from a particular tumour region, a new tip branch length ('leaf length') was drawn from a Poisson distribution based on the mean number of unique mutations observed in each dWGS sample from the same spatial tumour region. DNA samples that did not have matched RNA-seq samples were then removed from the trees (with drop.tip from ape R package v.5.6-1, ref. [76]). This

process was repeated 100 times for each tumour, leading to a forest of 100 phylogenetic trees with slightly varying branch length for each sWGS sample.

In the genic phylogenetic signal analysis, Pagel's $\lambda$ was calculated for group 1–3 genes ($n = 8,368$) using 'phylosig' from the phytools R package v.1.0-1 (ref. [77]). This returns the maximum-likelihood Pagel's $\lambda$ estimate and a $P$ value for the likelihood ratio test with the null hypothesis of $\lambda = 0$. This analysis was performed for all 100 trees and the median $\lambda$ and $P$ value determined for each tumour, with median $P < 0.05$ indicating evidence of phylogenetic signal for that gene. Genes with recurrent phylogenetic signal were defined as those with evidence of phylogenetic signal in at least three tumours. The STRINGdb R package v.2.6.1 (ref. [78]) was used to determine pathway enrichment of these recurrent phylogenetic genes, and 'string-db.org' was used for plotting of PPAR signalling genes.

To assess how phylogenetic signal is affected by purity, the analysis was rerun with purity-corrected expression. The coefficients of how purity determines gene expression had already been calculated during gene filtering (that is, the coefficient of purity in 'expression ~ purity' regression for all DNA matched samples (Methods) and samples used for phylogenetic analysis had matched DNA samples, allowing the use of accurate purity values. The expression of each gene (first normalized by DESeq2 variance-stabilizing transformation) was then normalized with the following equation:

$$\mathrm{Exp_{pur}} = \mathrm{Exp_{vst}} + (\text{Purity coefficient/Sample purity})$$

Phylogenetic signal analysis was then undertaken with purity-corrected expression (Supplementary Fig. 6).

In pathway phylogenetic signal analysis, pathway enrichment values were used as input for 'phylosig' for the 48 pathways. Evidence of phylogenetic signal was then determined as above. Recurrent phylogenetic pathways were defined as those with evidence of phylogenetic signal in at least two tumours, and Fisher's exact tests were used to determine enrichment/depletion in pathway groups and classes.

To determine the power for each tumour used in phylogenetic signal analysis, gene expression was simulated and $\lambda$ $P$ values estimated. Gene expression was Poisson distributed across nodes and was increased by a factor of 5–100% across every clade of the tree. This was performed over the forest of 100 trees of differing branch length, and this process was then repeated 1,000 times. The power to detect evidence of phylogenetic signal for a particular expression percentage change at a particular clade was therefore inferred by the percentage of simulations that had a median (that is, over the 100 branch-length-variant trees) $P < 0.05$.

## Assessment of phenotypic plasticity

For expression-based sample clustering, we calculated Euclidean distance matrices on genes from groups 1–3 ($n = 8,368$) and performed complete hierarchical clustering for each tumour with at least five RNA-seq samples ($n = 17$). The resulting dendrograms are plotted in Supplementary Fig. 10.

To quantify space–gene expression correlations we constructed a permutation test. For tumours with at least ten samples ($n = 11$), cophenetic distance matrices were extracted from the dendrograms plotted in Supplementary Fig. 10. The sum of all cophenetic distances between samples from the same tumour region was then calculated to acquire a metric of expression correlation with region for each tumour. To determine the significance of this metric, sample names for cophenetic distance matrix were randomly relabelled and the mixing statistic recalculated 10,000 times, followed by evaluation of whether the observed data were more extremely clustered than the random permutations (Supplementary Fig. 11). The intermixing scores used in Supplementary Fig. 9 were calculated as in Intermixing scores.

To assess the impact of tumour microenvironment we used CIBERSORTx[34], specifically with the LM22 signature file comprising 22 immune cell types[79] via the online portal (http://cibersortx.stanford.edu). First, Euclidean distances between the vector of gene expression from pairs of samples in the same tumour were calculated based on the expression of the 8,368 genes used in phylogenetic signal analysis. Euclidean distances were also calculated based on absolute scores from CIBERSORTx (note that CIBERSORTx was run using all genes). These two metrics were then plotted together for sample pairs from the same tumour and the correlation assessed (Supplementary Fig. 12).

## Genetic determinants of gene expression heterogeneity

Tumours with at least two tumour samples were included in this analysis (153 tumour samples from 19 tumours, median four samples per tumour) and only loci mutated in at least two samples and connected to an expressed gene (groups 1–3 from Fig. 1) were analysed (22,961 mutated loci connected to 5,927 expressed genes—29,949 unique gene–mutation combinations).

The following data were used as input for the linear model:
- Exp: a gene × sample matrix of variance-stabilized normalised gene expression of group 1–3 genes, converted to a $z$-score by subtracting the mean expression of all samples and dividing by the s.d. of all samples.
- CNA: a gene × sample matrix of the total copy number of the gene locus. If multiple copy number states were detected for the same gene, the segment overlapping most with the gene's locus was selected.
- Mut: a binary mutation × sample matrix in which mutations (SNVs and indels) were either within the enhancer region of the gene or a non-synonymous mutation within the coding region of the gene itself. Enhancer links to genes were defined using 'double-elite' annotations from GeneHancer tracks[80]. Some enhancer regions overlapped with the gene coding region, and non-synonymous mutations in these regions were annotated as both enhancer and non-synonymous.
- Purity: the purity of each sample as determined from dWGS or sWGS.

In addition, 14 matched normal samples were added and these were assigned WT for all mutations, 2 for total copy number and 0 for purity. For each gene–mutation combination, the following linear model was implemented: Exp $\sim$ Mut + CNA + Purity + Tumour, where 'Tumour' indicates whether the sample was a normal or tumour sample.

A gene–mutation combination was said to be explained if the FDR-adjusted $P$ value of the $F$-statistic for overall significance was less than 0.01. Storey's $\pi$, the estimate of the overall proportion of true null hypotheses, was calculated using the qvalue R package v.2.26.0 (ref. [81]). A gene–mutation combination was significantly affected by a variable (that is, Mut/CNA/Purity/Tumour) if the FDR-adjusted $P$ value for the coefficient of that variable was under 0.05.

For analysis of clonality (Fig. 2f), a mutation was considered 'subclonal' if at least one mutation associated with that gene was not found in all matched DNA–RNA samples for at least one tumour. For combination of eQTLs with phylogenetic analysis and clonality (Fig. 2g), a gene mutation combination was considered an 'eQTL' if it was significant for Mut, 'subclonal' if it was not found in all matched DNA/RNA samples for at least one tumour and considered 'phylogenetic' if the associated gene had significant phylogenetic signal in the tumour in which the mutation was present.

To look for recurrence of eQTL mutations in the Hartwig cohort, mutation loci were first converted to hg19 using liftOver from the rtracklayer R package v.1.54.0 (ref. [82]) and 'hg38Tohg19.over.chain' from http://hgdownload.cse.ucsc.edu/goldenpath/hg38/liftOver. Two out of 22,961 loci could not be converted and were therefore discarded for this analysis. Converted loci were searched for in the CRC Hartwig cohort using the 'purple.somatic.vcf.gz' files. For Hartwig gene expression, 'adjTPM' values were used and converted to a $z$-score whereas tumour purity was extracted from the metadata. For each locus with at least one mutated DNA–RNA Hartwig sample, the linear models of

Exp ~ Mut + Purity and Exp ~ Purity were compared via a likelihood ratio test. An eQTL was said to validate in Hartwig if the $P$ value of the test was under 0.05 and the coefficient of the Mut variable was the same sign as the coefficient in the original eQTL analysis (that is, the mutation increased expression in EPICC and Hartwig or vice versa).

A post hoc power analysis was carried out using the pwr.t2n.test from the pwr R package v.1.3-0 (ref. [83]). For each eQTL, absolute mutation effect size was used as the input effect size with 'power' set to 0.99 and 'n2' set to the number of DNA–RNA Hartwig CRC samples ($n = 394$) minus the number of Hartwig samples with the mutation. The tool then returned the number of samples needed to determine the effect, and this number was multiplied by 1.15 given the non-parametric nature of the data. If absolute input effect size was greater than 3.04, this was set to 3.04 because higher values returned a 'not available' result.

## MSI investigations for eQTL analysis

A PCA analysis of germline SNPs plotted with ggbiplot v.0.55 (ref. [84]) found a lack of bias for germline SNPs, with the top two principal components accounting for only 16.6% of explained variation (Supplementary Fig. 16). Labelling tumours by MSI status also showed that principal component 1 slightly separated MSS from MSI tumours.

To directly assess the effect of MSI on eQTL analysis the analysis was rerun twice, once with only MSS tumour samples ($n = 149$ across 15 tumours) and again using only MSI tumour samples ($n = 18$ across three tumours). Given the large difference in sample size and therefore power, to make the two analyses comparable only mutations with very large (over 1.5) effect sizes were considered. The absolute mutation effect sizes of 73 eQTLs from the MSS analysis were therefore compared with 293 eQTLs from the MSI analysis. A $QQ$-plot comparing these two datasets showed there was a difference in the distribution of effect sizes of significant eQTLs between MSS and MSI analyses (Supplementary Fig. 17). Specifically, there was a higher proportion of MSS eQTLs at very large effect size in comparison with the MSI analysis. This is interesting because it suggests a difference in the genetic control of gene expression between MSS and MSI tumours.

The original eQTL analysis was also rerun with MSI as a cofactor (Supplementary Fig. 18), and this was found to have a minor impact on results. Notably, there was a small decrease in the number of significant eQTL genes (Supplementary Fig. 18a,b), non-coding enhancers were no longer significantly associated with increases in expression ($P = 0.08$; Supplementary Fig. 18e) and subclonal mutations were no longer more likely to be eQTLs ($P = 0.17$; Supplementary Fig. 18f). However, it should be noted that the direction of these effects did not change. Finally, the distribution of $R^2$ values was compared between the original analysis (without MSI as a covariate) and with MSI as a covariate. Supplementary Fig. 19 shows that, for models that were significant in both analyses, $R^2$ values were highly correlated ($P < 1 \times 10^{-16}$, $R^2 = 0.855$). It is worth noting that $R^2$ values tend to be higher for the analysis with MSI, and this was found to be significant (paired Wilcoxon signed rank test, $P = 1.071 \times 10^{-241}$). Therefore, inclusion of MSI as a covariate marginally increased the amount of variance explained by each model but $R^2$ values were very highly correlated with the original analysis

## dN/dS analysis

Per-patient variant calls were obtained from the VCF files and lifted to the hg19 reference genome using the rtracklayer R package v.1.54.0 (ref. [82]). Variants were split into clonal (that is, present in all samples) and subclonal mutations (that is, present in a subset of samples) in cancer, as well as a set of mutations present in any of the adenomas. Patients were further split into MSI and MSS tumours. The dndscv model (dndscv R package v.0.1.0)[45] was fit separately for each of the four mutation sets. For this, default parameters apart from deactivated removal of tumours due to the number of variants were used. In addition to global dN/dS estimates of the fitted models, dN/dS estimates of CRC-specific

driver mutations from IntOGen[41,85] were obtained with the 'genesetdnds' function of dndscv.

## Gene essentiality analysis

Cancer dependency profiles were downloaded from https://depmap.org/broad-sanger/ (version used: CRISPRcleanR_FC.txt) and scaled as previously described[86], making the median essentiality scores of previously known essential and non-essential genes equal to −1 and 0, respectively. The mutational status of selected putative cancer driver genes used to produce the box plots in Supplementary Fig. 20, and to test differential gene essentiality across mutant versus WT cell lines, was obtained from Cell Model Passports[87].

## In situ mutation detection

BaseScope in situ mutation detection was performed as previously described[52], using mutation-specific probes designed and provided by the manufacturer. Data were assessed manually: a tumour gland was denoted as 'mutant' if at least one cell in the gland had detected expression of the mutant transcript, otherwise it was classified as 'wild type' for that mutation.

## Spatial computational inference

Inference of evolutionary dynamics using spatially resolved genomic data was performed by Bayesian fitting of a spatial agent-based model of clonal evolution to the observed molecular data. The model described growth, death, physical dispersion and mutation of individual tumour glands, and was a substantial modification of the framework previously described in ref. [51]. Full details are provided in the Supplementary mathematical note.

## Transcriptomic and epigenetic characterization of selected clones

Differential expression analysis was run using DESeq2 (ref. [59]), comparing RNA samples in inferred selected regions with all other samples from that tumour. Analysis was also rerun with random shuffling of sample labelling to filter for the signal of the selected subclone, and genes found to be differentially expressed in more than 5% of shuffled analyses were excluded. Volcano plots of significant differentially expressed genes were plotted with EnhancedVolcano v.1.12.0 (ref. [88]) (Supplementary Fig. 25a). To perform gene set enrichment analysis[89] all remaining genes were ordered by DESeq2's test statistic, and enrichment of Gene Ontology annotations, KEGG pathways and Hallmark pathways was tested for (FDR < 0.05) using gseGO, gseKEGG and GSEA, respectively, from ClusterProfiler[65]. Significant results are shown in Supplementary Fig. 26.

We also performed differential ATAC-seq peak analysis between selected subclones and background clones. To assess the subclonality of ATAC-seq peaks while controlling for purity, a log-ratio test from DESeq2 was used to compare a 'full model' of '~ purity + clone' to a 'reduced model' of '~ purity'. ATAC-seq peaks were considered to be significantly altered in selected clones when the adjusted P value was below 0.05 (Supplementary Fig. 25b).

## Reporting summary

Further information on research design is available in the Nature Research Reporting Summary linked to this article.

## Data availability

Gene expression data, somatic mutation calls (VCFs from Mutect2+Platypus), copy number calls (Sequenza and QDNAseq), fraction of mutated microsatellites (MSIsensor), ATAC-seq insertion sites and allele counts of somatic SNVs in all sample types are available at Mendeley (https://doi.org/10.17632/7wx3chtsxx.2). Sequence data (processed BAM files) have been deposited at the European Genome-phenome Archive (EGA), which is hosted by the EBI and CRG, under study no. EGAS00001005230. Access to these data is restricted and subject to application. Source data are provided with this paper.

## Code availability

Complete scripts to replicate all bioinformatic analyses and perform simulations and inference are available at https://github.com/sottorivalab/EPICC2021_data_analysis, https://github.com/sottorivalab/EPICC2021_data_analysis_RNA and https://github.com/T-Heide/MLLPT.

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

**Acknowledgements** We thank the ICR's Tumour Profiling Unit, especially N. Matthews, P. Ramigiri, I. Assiotis, K. Fenwick and R. Chauan, for their support in the sequencing efforts. This study was principally supported by funding from the Wellcome Trust (nos. 202734/Z/16/Z to T.A.G. and 202778/B/16/Z to A.S.) and the Medical Research Council (no. MR/P000789/1 to A.S.). A.S. and T.A.G. were also supported by Cancer Research UK (nos. A22909 and A19771, respectively) and the National Institutes of Health (no. NCI U54 CA217376 to D.S., T.A.G. and A.S). This work was also supported by a Wellcome Trust award to the Centre for Evolution and Cancer at the ICR (no. 105104/Z/14/Z). We also acknowledge funding from the Cancer Research UK Accelerator Award (no. A26815 to A.S.). C.B. acknowledges funding from the

Wellcome Trust (no. 209409/Z/17/Z). B.W. is supported by a Barts Charity Lectureship (grant no. MGU045) and a UKRI Future Leaders Fellowship (grant no. MR/V02342X/1). D.R. was partially supported by a Bicocca 2020 Starting Grant and by a Premio Giovani Talenti dell'Università degli Studi di Milano-Bicocca. L.M. is supported by Cancer Research UK (no. A23110).

**Author contributions** J.H. analysed and interpreted the data, with focus on RNA-seq data. T.H. analysed and interpreted data, with focus on ATAC-seq and WGS data. T.H. also designed and implemented the computational inference framework and performed inference analysis. G.D.C. performed copy number analysis. I.S. devised the multiomics protocol, collected samples and generated data. C.K. collected samples and contributed to data generation. L.Z. contributed to dN/dS data analysis. C.L. contributed to ATAC-seq data analysis. M. Mossner, J.F.-M. and A.-M.B. contributed to data generation. B.C. analysed methylation array data. C.J., C.G., E.L., G.C., D.R. and D.N. contributed to data analysis. A.V. performed DepMap analysis. A.B. generated methylation array data. F.I. supervised DepMap analysis and contributed to results interpretation. H.C., M. Mitchinson and M.J. contributed to tissue collection. B.W. contributed to inference interpretation. C.B. supported inference analysis and contributed to results interpretation. D.S. contributed to experimental design and data interpretation. J.B. and M.S. contributed to sample collection coordination. V.G. performed histopathological analysis. M.R.-J. supervised sample collection and performed histopathological analysis. L.M. contributed to results interpretation. T.A.G. and A.S. conceived, designed and supervised the study and wrote the manuscript.

**Competing interests** A.-M.B. has received honoraria from Pfizer and Eisai for non-promotional educational content in the field of genomics. F.I. receives funding from Open Targets, a public–private initiative involving academia and industry, and performs consultancy for the joint CRUK–AstraZeneca Functional Genomics Centre. All other authors declare no competing interests.

## Additional information
**Correspondence and requests for materials** should be addressed to Andrea Sottoriva or Trevor A. Graham.

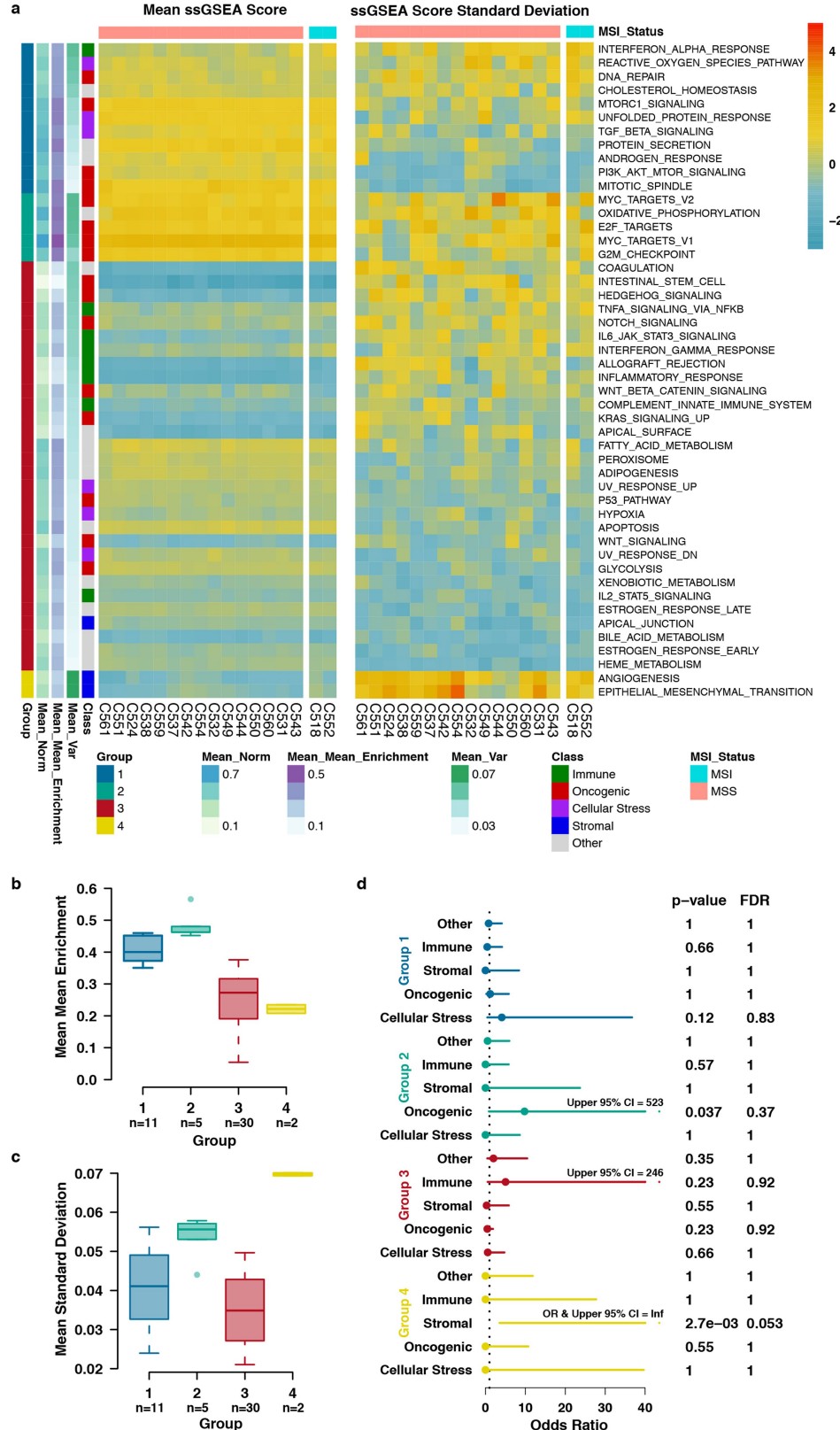

**Extended Data Fig. 1 | Filtering and clustering of pathways based on mean tumour enrichment and intra-tumour heterogeneity of enrichment.** **a**, Heatmaps showing clustering of pathways by enrichment level across tumours and enrichment variation within tumours. Hierarchical clustering revealed four distinct groups, named Group 1–4. Note units are scaled by column in both heatmaps **b**, Summary of mean enrichment level per Class. **c**, Summary of intra-tumour heterogeneity of enrichment per Class, measured by standard deviation. **d**, Fisher's exact test (two-sided) results comparing pathway Groups to classes.

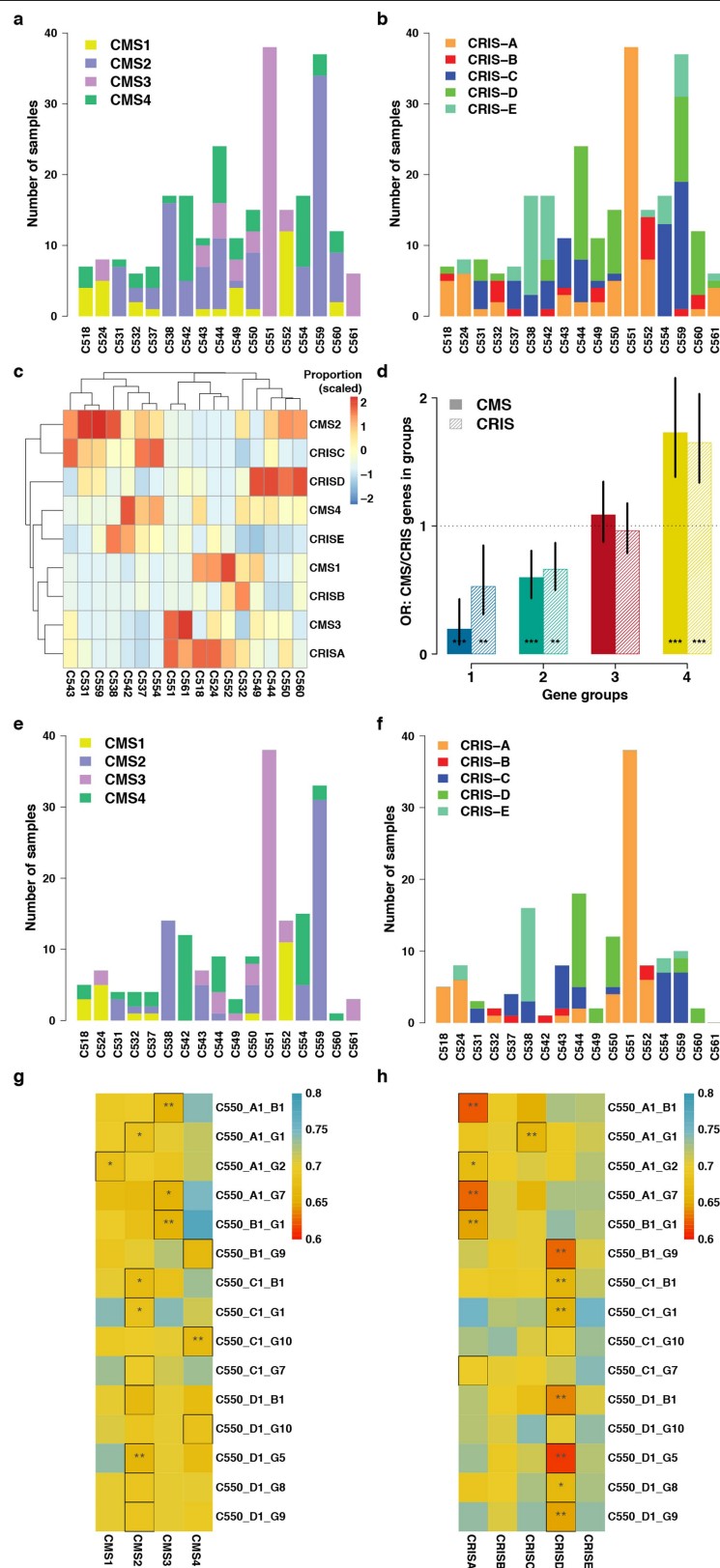

**Extended Data Fig. 2 | CMS and CRIS classification heterogeneity. a**, Stacked bar charts showing per-sample classification of CMS in each tumour. **b**, Stacked bar charts showing classification of CRIS in samples from each tumour. **c**, Heatmap based on data in **(a,b)** where colour indicates proportion of samples of a particular CRIS/CMS class. Associations between CMS and CRIS classes are apparent. **d**, Examination of the enrichment of genes respectively included CMS and CRIS classifications in gene Groups as defined in Fig. 1. Both CRIS and CMS genes are depleted in gene Groups 1&2 but are enriched in Group 4. 11,401 genes used in two-sided Fisher's exact tests. Error bars represent 95% confidence intervals. **e**, CMS assignments per tumour, only samples which could be confidently classified (FDR < 0.05) are shown. **f**, As in **(e)** but for CRIS. **g**, Heatmap of centroid distances of each sample from CMS classes for tumour C550, black squares indicate the minimum (most likely) class for each sample, and stars represent significance of classification. **h**, As in **(g)** but for CRIS.

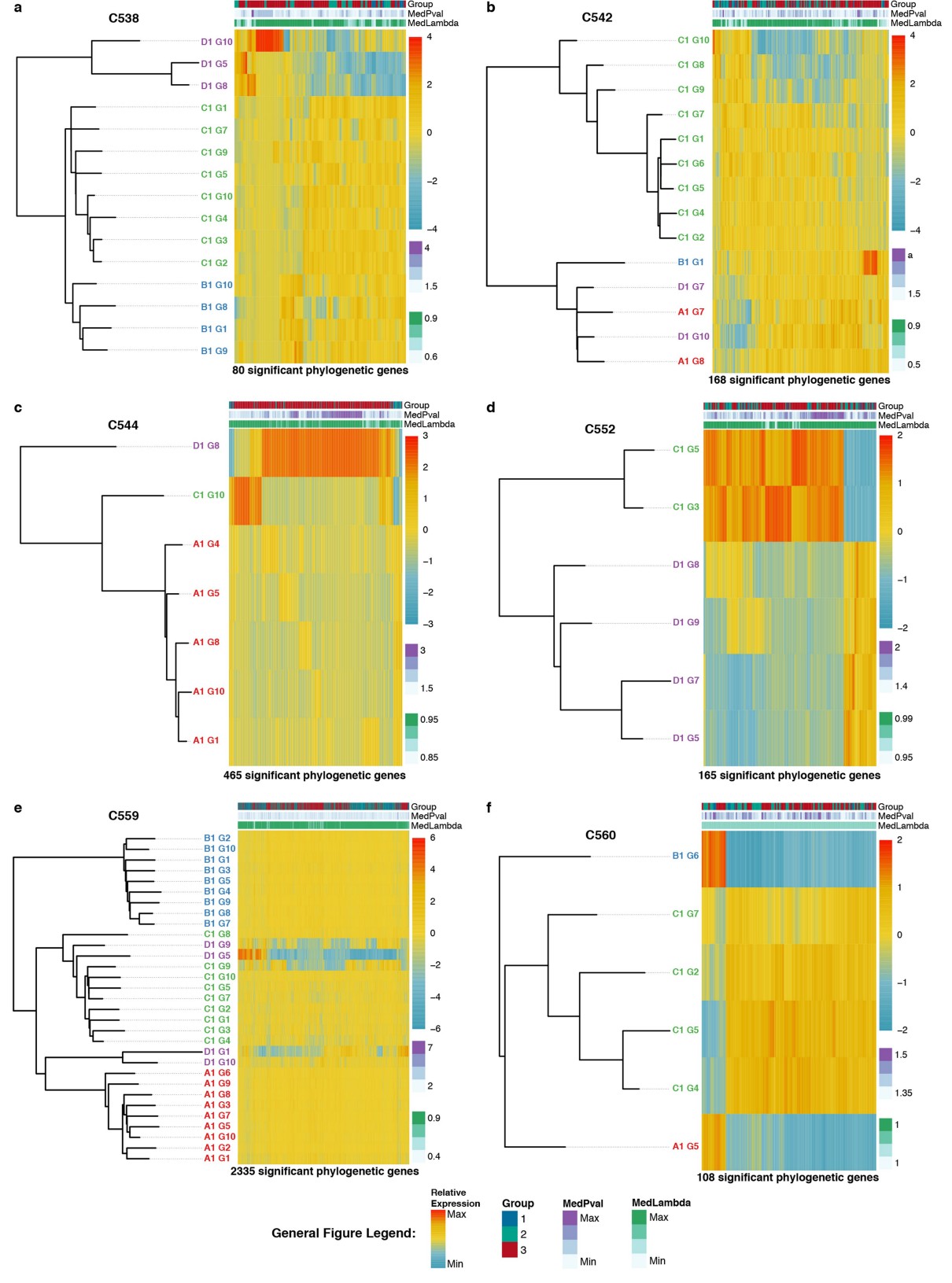

**Extended Data Fig. 3 | Phylogenetic trees (left) annotated with the expression of genes (heatmap, right) which had evidence of phylogenetic signal (*P*<0.05). MedPval = median p-value from forest of 100 trees,** **MedLambda = median lambda value from forest of 100 trees.** Shown by tumour for: **a**, C538. **b**, C542. **c**, C544. **d**, C552. **e**, C559. **f**, C560.

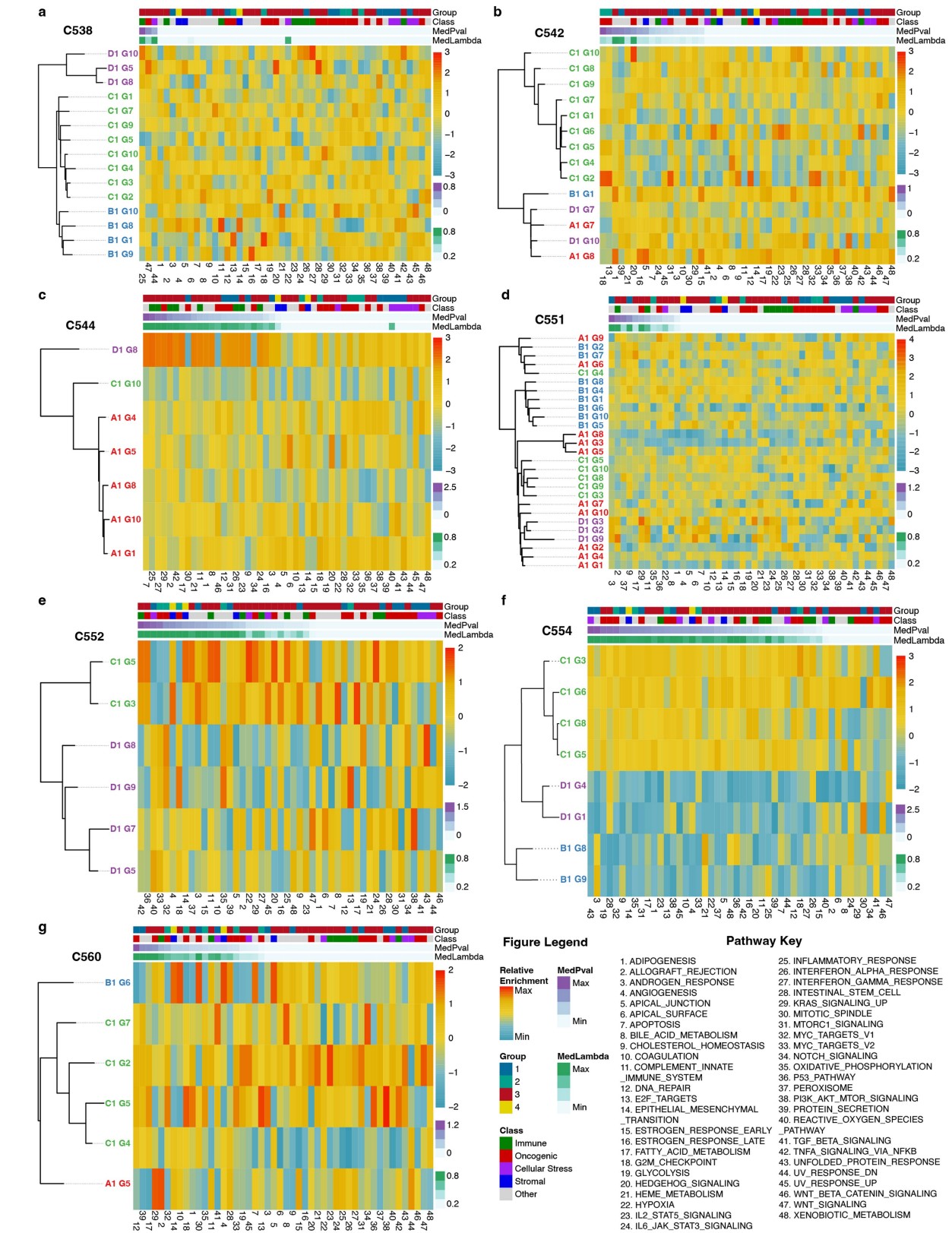

**Extended Data Fig. 4 | Phylogenetic trees (left) annotated with pathway enrichment scores (heatmap, right). MedPval = median p-value from forest of 100 trees, MedLambda = median lambda value from forest of 100 trees.**

Shown by tumour: **a**, C538. **b**, C542. **c**, C544. **d**, C551. **e**, C552. **f**, C554. **g**, C560. Numbers on heatmap x-axis indicate hallmark pathways, refer to Pathway Key.

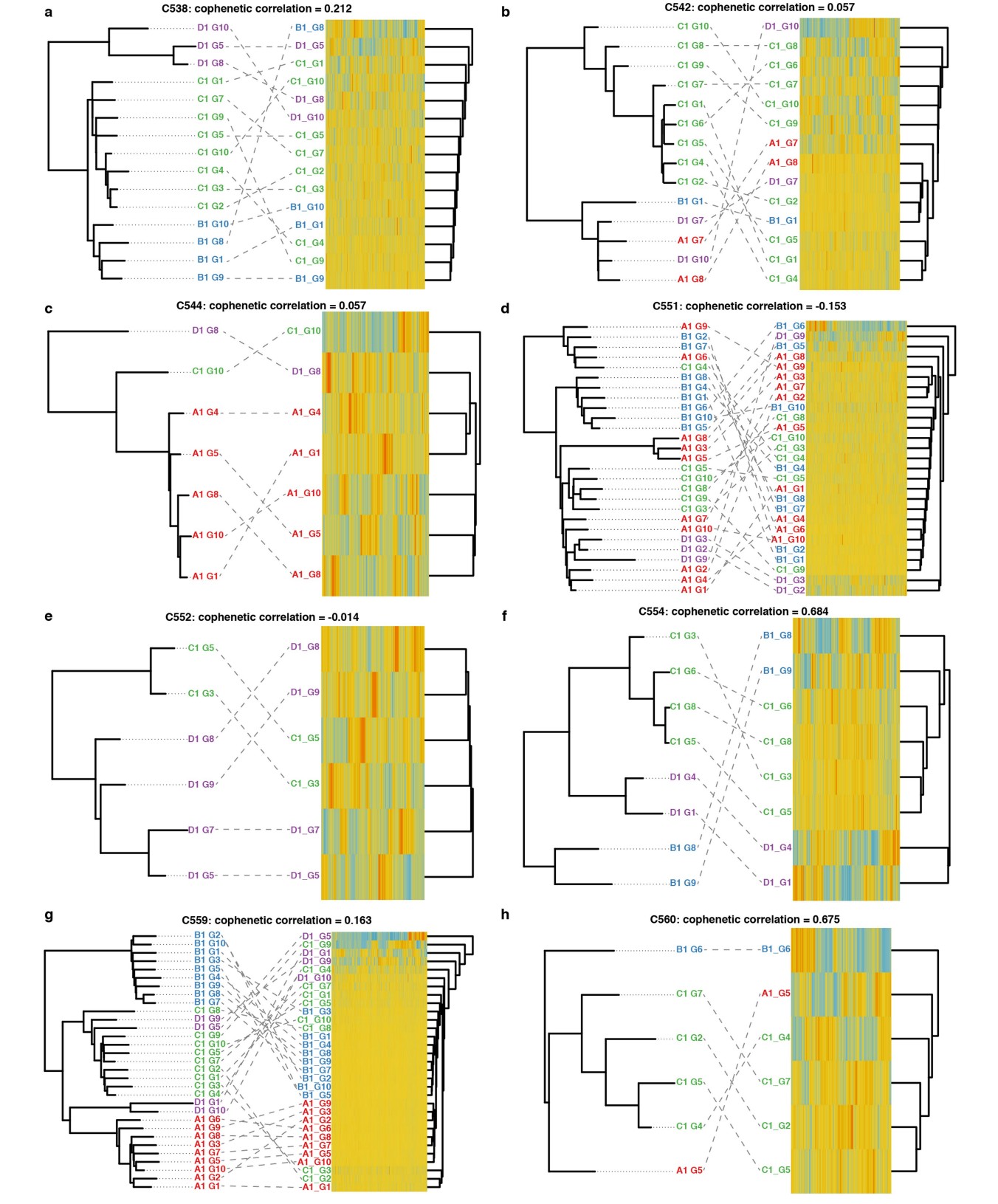

**Extended Data Fig. 5 | Phylogenetic tree versus expression-based clustering.** The dendrogram on the left of each panel is the mutation-based phylogenetic tree, while samples on the right are clustered according to gene expression. Dotted lines show matching samples and samples are coloured according to region-of-origin. **a**, C538. **b**, C542. **c**, C544. **d**, C551. **e**, C552. **f**, C554. **g**, C559. **h**, C560.

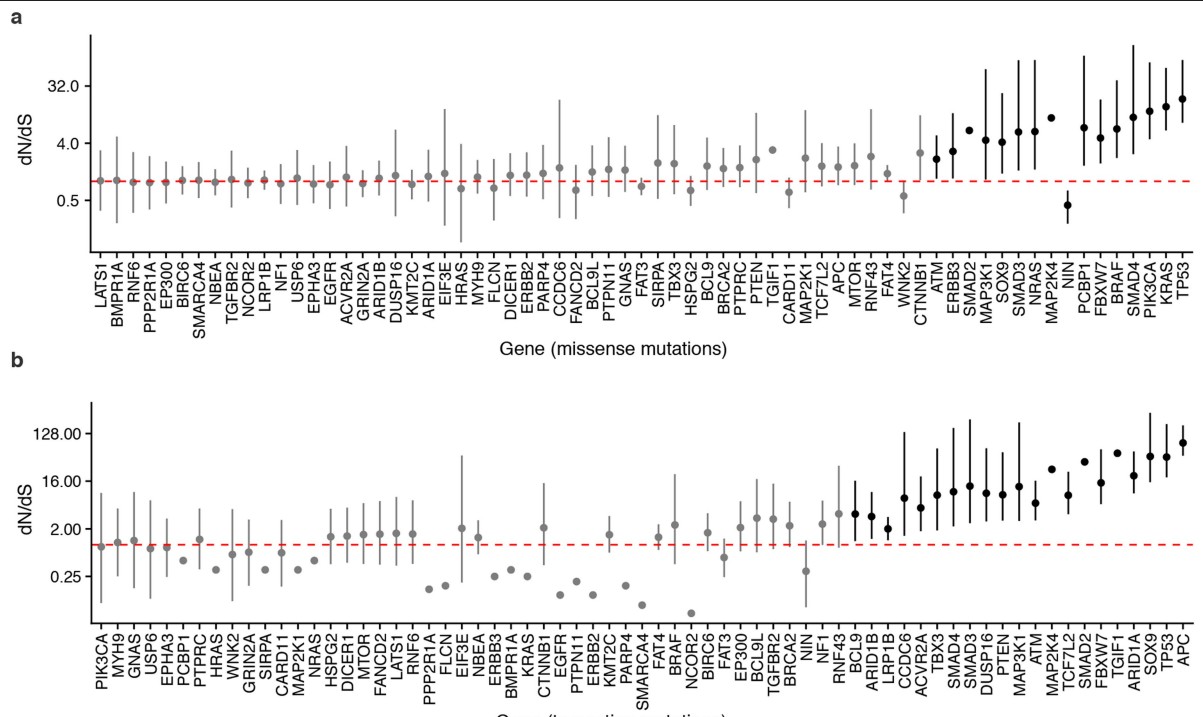

**Extended Data Fig. 6 | Per-gene dN/dS analysis of drivers (n = 1,253 CRCs).**
**a**, Per-gene dN/dS for the 69 IntOGen drivers in TCGA colon and rectal cohorts split into **(a)** missense mutations and **(b)** truncating mutations. Many genes have dN/dS value ≈1 indicating lack of evidence for positive selection. The points show the point estimates and the error bars the 95% CI intervals of the dN/dS.

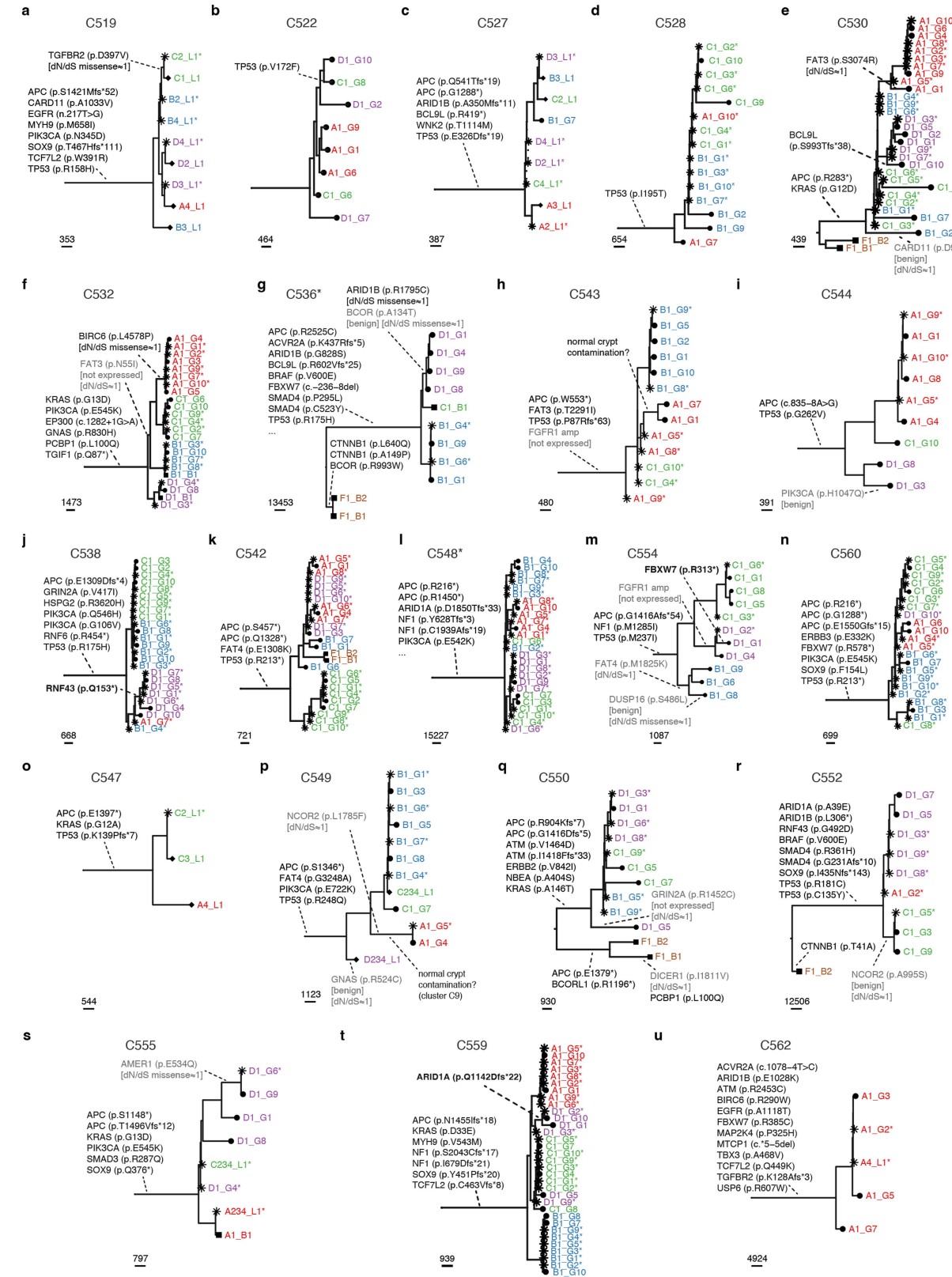

**Extended Data Fig. 7 | Phylogenetic reconstruction of more cancers and adenomas.** Putative cancer driver genes from the IntOGen set are reported in each branch. For MSI tumours we report only a subset of the most relevant genes (see Fig. 3a for a full list). For subclonal drivers, we report whether the variant was expressed (bold), not expressed or benign (grey), and if the per-gene dN/dS value was ≈1. **a**, C519. **b**, C522. **c**, C527. **d**, C528. **e**, C530. **f**, C532. **g**, C536. **h**, C543. **i**, C544. **j**, C538. **k**, C542. **l**, C548. **m**, C554. **n**, C560. **o**, C547. **p**, C549. **q**, C550. **r**, C552. **s**, C555. **t**, C559. **u**, C562.

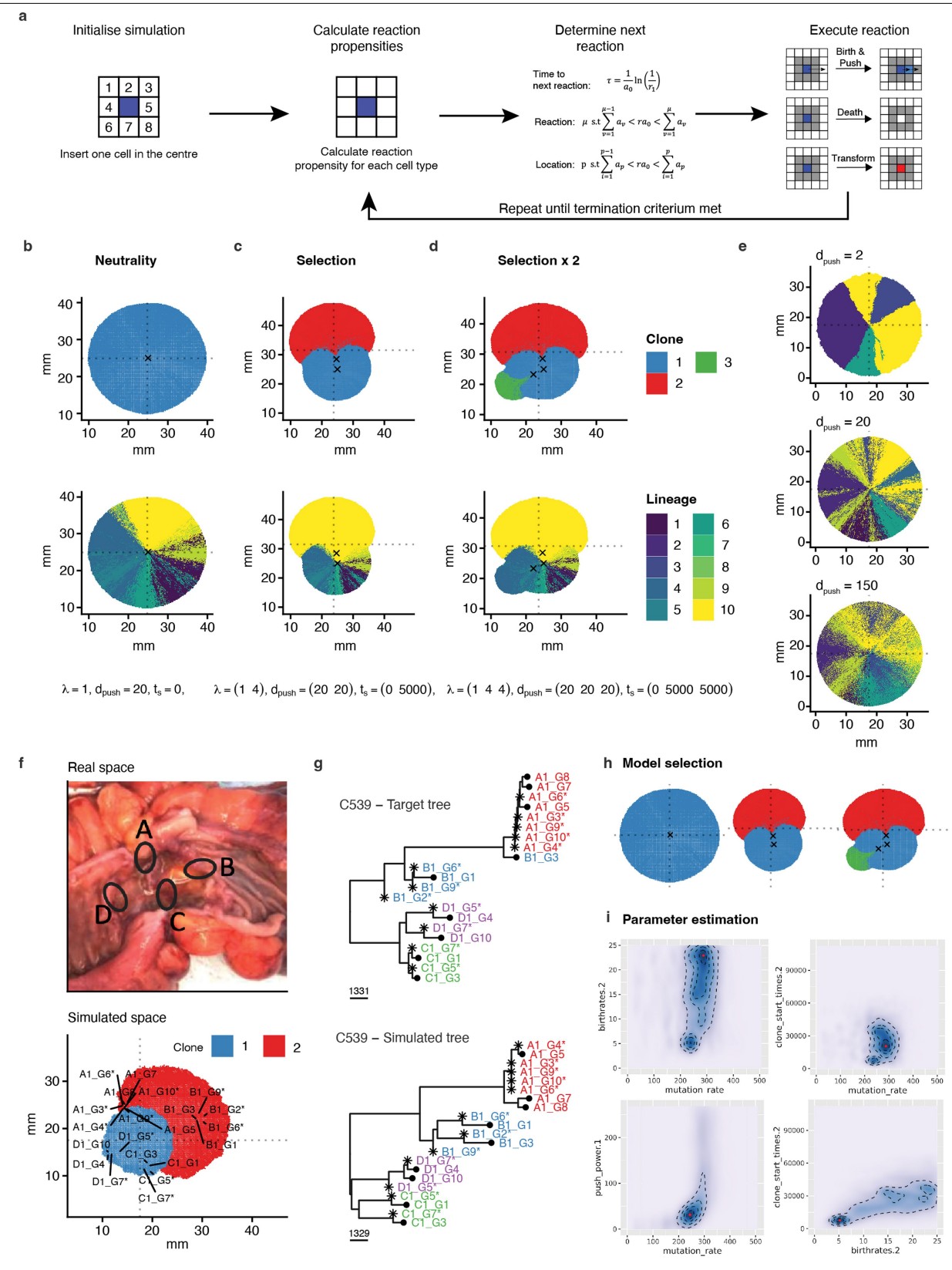

**Extended Data Fig. 8** | See next page for caption.

**Extended Data Fig. 8 | Bayesian inference framework for cancer dynamics in space and time. a**, Schematic representation of the spatial cellular automaton model of tumour growth. **b**, Instance of simulation of a neutrally expanding cancer with a single 'functional' clone (blue, top), and corresponding neutral mutation lineages (bottom). **c**, Simulation of a tumour containing a differentially selected subclone (red, top) and corresponding neutral mutation lineages (bottom). **d**, Simulation with two branching subclonal selection events. **e**, In this neutral simulation we illustrate peripheral versus exponential growth and the effects on lineage mixing. **f**, Spatial sampling annotated during tissue collection for tumour C539 and corresponding simulated spatial sampling. **g**, Real data from patient C539 (top) versus simulated data from an instance selected by the inference framework (bottom). **h**–**i**, Inference framework based on Approximate Bayesian Computation - Sequential Monte Carlo (ABC-SMC) allows for **(h)** model selection and **(i)** posterior parameter estimation given the data. In this case birthrates.2 is the birth rate of the selected subclone, clone_start_times.2 is the time when the subclone arose during the growth of the tumour, push_power.1 is the coefficient of boundary driven growth and mutation_rate is the rate of accumulation of mutations per genome per division.

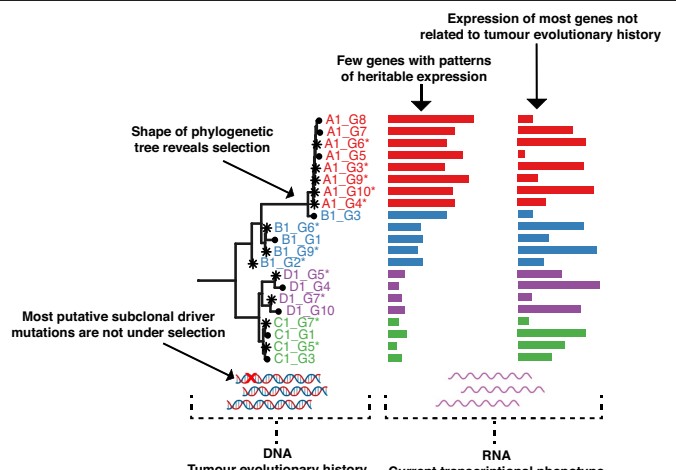

**Extended Data Fig. 9 | Visual schematic illustrating the main results.**
DNA-sequencing of multiple glands from up to four tumour regions allowed
elucidation of tumour evolutionary history and selection inference (left;
phylogenetic tree). Matched RNA-sequencing found few genes with heritable
expression patterns (middle; bars represent expression level), with the
expression of most genes not detectably related to tumour evolutionary
history (right; bars represent expression level). Most putative subclonal driver
mutations were found to not be under selection, while transcriptomic
differences could be found between subclones that were under positive
selection. Sample names and bars are coloured according to region-of-origin.

Andrea Sottoriva

# Reporting Summary

## Statistics

For all statistical analyses, confirm that the following items are present in the figure legend, table legend, main text, or Methods section.

| n/a | Confirmed | |
|---|---|---|
| ☐ | ☒ | The exact sample size ($n$) for each experimental group/condition, given as a discrete number and unit of measurement |
| ☐ | ☒ | A statement on whether measurements were taken from distinct samples or whether the same sample was measured repeatedly |
| ☐ | ☒ | The statistical test(s) used AND whether they are one- or two-sided  *Only common tests should be described solely by name; describe more complex techniques in the Methods section.* |
| ☐ | ☒ | A description of all covariates tested |
| ☐ | ☒ | A description of any assumptions or corrections, such as tests of normality and adjustment for multiple comparisons |
| ☐ | ☒ | A full description of the statistical parameters including central tendency (e.g. means) or other basic estimates (e.g. regression coefficient) AND variation (e.g. standard deviation) or associated estimates of uncertainty (e.g. confidence intervals) |
| ☐ | ☒ | For null hypothesis testing, the test statistic (e.g. $F$, $t$, $r$) with confidence intervals, effect sizes, degrees of freedom and $P$ value noted  *Give P values as exact values whenever suitable.* |
| ☐ | ☒ | For Bayesian analysis, information on the choice of priors and Markov chain Monte Carlo settings |
| ☐ | ☒ | For hierarchical and complex designs, identification of the appropriate level for tests and full reporting of outcomes |
| ☐ | ☒ | Estimates of effect sizes (e.g. Cohen's $d$, Pearson's $r$), indicating how they were calculated |

*Our web collection on statistics for biologists contains articles on many of the points above.*

## Software and code

Policy information about availability of computer code

| Data collection | No software was used for data collection |
|---|---|
| Data analysis | https://github.com/sottorivalab/EPICC2021_data_analysis<br>https://github.com/sottorivalab/EPICC2021_data_analysis_RNA<br>https://github.com/T-Heide/MLLPT |

For manuscripts utilizing custom algorithms or software that are central to the research but not yet described in published literature, software must be made available to editors and reviewers. We strongly encourage code deposition in a community repository (e.g. GitHub). See the Nature Portfolio guidelines for submitting code & software for further information.

## Data

Policy information about availability of data

All manuscripts must include a data availability statement. This statement should provide the following information, where applicable:
- Accession codes, unique identifiers, or web links for publicly available datasets
- A description of any restrictions on data availability
- For clinical datasets or third party data, please ensure that the statement adheres to our policy

This manuscript is a reanalysis of data in associated manuscript Heide, Househam et al. (2022). The data in that associated manuscript are available on Mendeley (DOI 10.17632/7wx3chtsxx.1) and sequence data have been deposited at the European Genome-phenome Archive (EGA), which is hosted by the EBI and the CRG, under accession number EGAS00001005230. Additional supplementary information is available on figshare (DOI 10.6084/m9.figshare.c.6122193).

# Field-specific reporting

Please select the one below that is the best fit for your research. If you are not sure, read the appropriate sections before making your selection.

☒ Life sciences ☐ Behavioural & social sciences ☐ Ecological, evolutionary & environmental sciences

# Life sciences study design

All studies must disclose on these points even when the disclosure is negative.

| | |
|---|---|
| Sample size | No power calculation was performed prior to starting the study. Post hoc power calculations were performed throughout the study and results interpreted in light of these analyses. |
| Data exclusions | Various data exclusions were performed on an analysis-by-analysis basis (usually due to available sample numbers with required multi-omics and/or samples per tumour). These are specified in the manuscript at the point that each analysis is described. |
| Replication | Specific replication on individual samples was not possible due to insufficient material (as described in associated manuscript detailing wet-lab data generation). Repeat sampling of tumours provided a level of psuedo-replication. |
| Randomization | Randomization was not relevant to this study. The explicit analysis clinical-pathological of covariates, such as mismatch repair status, is reported in the manuscript. Our study was an observational cohort study so randomization was not possible. No differential outcomes were examined so randomization was not needed. |
| Blinding | Researchers were blinded to clinical-pathological data. |

# Reporting for specific materials, systems and methods

We require information from authors about some types of materials, experimental systems and methods used in many studies. Here, indicate whether each material, system or method listed is relevant to your study. If you are not sure if a list item applies to your research, read the appropriate section before selecting a response.

## Materials & experimental systems

| n/a | Involved in the study |
|---|---|
| ☒ | ☐ Antibodies |
| ☒ | ☐ Eukaryotic cell lines |
| ☒ | ☐ Palaeontology and archaeology |
| ☒ | ☐ Animals and other organisms |
| ☒ | ☐ Human research participants |
| ☒ | ☐ Clinical data |
| ☒ | ☐ Dual use research of concern |

## Methods

| n/a | Involved in the study |
|---|---|
| ☒ | ☐ ChIP-seq |
| ☒ | ☐ Flow cytometry |
| ☒ | ☐ MRI-based neuroimaging |

