## [Peer Review File · Nature]

Manuscript Title: Phenotypic plasticity and genetic control in colorectal cancer evolution

Reviewer Comments & Author Rebuttals

Reviewer Reports on the Initial Version:

Referees' comments:

Referee #1 (Remarks to the Author):

Househam, Heide et al. submit here a manuscript exploring the impact of genetic variation on transcriptomic intra-tumoral heterogeneity (tITH). The group use matched multi-regional WGS, RNAseq and ATACseq from 17 invasive carcinoma to understand the control of tITH. The manuscript forms part of a broader study using the same (and additional adenoma and normal) samples that provides greater granular detail and explanations of the 'big bang' theory previously published. In this submission, the authors initially describe the degree of tITH and cluster this according to both average expression and degree of variance. Leveraging matched WGS data the authors conclude that RNA expression is neither strongly heritable nor related to genetic ancestry. Next, the authors seek to understand the phenotypic consequence of somatic driver mutations through dN/dS analysis of their multi-regional WGS data. Interestingly, they conclude that most driver mutations have limited 'phenotypic consequence' and demonstrate limited selection. To further explore these observations the authors next apply complex computational modelling of tumour spatial growth and sub-clonal expansion. They find strong correlations between the model and their real-world data which allows them to accurately infer the evolutionary consequence of somatic driver mutations. Strikingly, neutral evolution appears dominant with only a third of tumours showing strong evidence of subclonal selection. Finally, the authors use samples with matched ATACseq and RNAseq data in tumours displaying sub-clonal selection and highlight recurrent alterations in focal adhesion pathways.

This is an impressive dataset, that for the first-time permits dissection of some truly fundamental aspects of tumour evolution and phenotype. The authors should be applauded for the scope of this study. The manuscript is well written and figures are of a reasonable quality however some suffer from too much detail and small font size making it difficult to read. Data generated appear high quality and the authors seem to have taken much care in quality control. The highly complex narrative flow is decent through most of the manuscript although the latter section using ATACseq seems somewhat disconnected and limited in depth. Crucially, whilst this reviewer finds these data and analyses to be reliable, I find some of the conclusions and even some premises difficult to fully accept at present. This is perhaps reflective of the innovative focus of this study which has never been addressed before in this depth and for which terminology may be early in acceptance.

Major comments:

1. I think there is a fundamental question surrounding plasticity and variance that needs to be addressed. The authors state in their abstract "Using spatially-resolved paired whole genome and

transcriptome sequencing, we find that the majority of intra-tumour variation in gene expression is not strongly heritable and therefore is plastic.” I am not fully convinced that that just because a transcriptome is not heritable it has to be plastic. This reviewer would interpret ‘plastic’ as being variant and capable of change/reversion. Surely, different ‘normal’ tissues have varying degrees of plasticity – for example the gut most would regard as plastic and the CNS less so, both however have transcriptional heterogeneity (variance). Both also have non-heritable gene expression. Further, the analyses the authors are performing are based on a single time point so how can they be confident that these read outs do or don’t change over time and if they can what is the sensitivity of this? Whilst this may well be semantics, I think this issue requires clarity.

2. Similarly, in the discussion the authors state “This points towards phenotypic plasticity the ability of a cancer cell to change phenotype without underlying heritable (epi)genetic change to be a common phenomenon in CRC.”. The authors seem to be linking transcriptional variance with phenotypic plasticity in this paragraph. Is this not conflation? There is therefore an important definition that this reviewer is unclear on – are the authors defining phenotype as the degree of selection advantage within the tumour or the transcriptional pattern or both? Is it truly possible to infer such a highly dynamic process from these forms of analysis that rely on evolutionary inference based on MRS?

3. In a related point to #1 I think it would be useful to provide context to the transcriptomic analyses. There is little description of the degree of variance in normal tissue. Given how much detail is known about phenotype and plasticity in normal colonic tissue is it not possible to link the tumour findings back to normal to provide both reassurance as to the authors’ conclusions and contextual framework.

4. The ATACseq portion of the manuscript appears incomplete in analysis and doesn’t integrate well into the manuscript. The authors have understandably mainly chosen to focus on those tumour subclones showing a selection advantage but what are seen in those that do not? Indeed, more generally what is the epigenetic contribution to tITH, at least in regards to chromatin accessibility? Surely, this could be addressed using data the authors have generated?

Minor comments:

1. CMS and CRIS calling p3. These are quite striking results that call into question the entire validity of these sub-types. If this is true then how is it that these subtypes are capable of defining clinically relevant groups based on single biopsies? Also, why do the authors feel that CRIS and CMS don’t correlate well when this has previously been shown not to be the case?

2. p6. The negative correlations of SCNAs with expression change (10%) is interesting. The authors speculate that this may be related to dominant-negative activity. This requires further attention and confirmation to ensure that this is not a technical error and better understand this result.

3. The DepMap analysis (p8) is quite superficial and this reviewer is not certain that linking this quite artificial cell line derived dataset to their real world data is helpful.

4. I cannot fully review the validity of the computational model used in this study as it is outside of my area of expertise however, I would be interested to know why a ΔAIC of >4 was used? Further how does the model take into account cooperative as well as more accepted competitive interactions between subclones?
5. I am uncertain why the adenoma samples present in the wider EPICC study have not been included here. Surely it is important to define whether the evolutionary principles proposed in this submission are conserved (or not) at an earlier time in tumour development?
6. Does cytosol specific RNAseq bias towards specific transcripts compared to whole cell RNAseq?
7. Does the sampling of purely glandular structures bias the transcriptomic analysis? How reflective are the glandular samples of more traditionally sampled bulks?
8. Some of the supplementary figures have overlapping gene annotations which should be tidied up.

Referee #2 (Remarks to the Author):

A. Summary of key results

The manuscript leverages the single-gland multiregion profiling of colorectal cancer by whole genome sequencing and RNAseq (generated for an accompanying manuscript) to infer the relationships between genetic and transcriptomic heterogeneity. The authors reconstruct phylogenetic trees to identify genetic ancestry and find that gene expression patterns do not strictly follow the genetic clustering of samples. Aiming to identify the true genetic drivers of CRC, the authors perform eQTL analysis and confirm that most mutations do not alter gene expression, while some of the subclonal genetic variants can affect gene expression. Interestingly, driver mutations identified in this study are predominantly clonal and very few cases of positive selection are identified. This can be attributed to higher-than-before depth of the data (single-gland analysis and multiregional sampling). CRCs analyzed in this study are either under “neutral” evolution or show signs of clonal selection. The Authors generate a mathematical model recapitulating the observed phylogenetic trees, which allows to capture evolutionary consequences of subclonal driver mutations.

B. Originality and significance

The manuscript presents original data and a novel mathematical model for identification of the impact of subclonal driver mutations.

C. Data and methodology

The impressive depth of data generated for this manuscript allows for a more detailed characterization of clonal events in CRC. The quality of data and presentation is high and meets the standards of the journal.

D. Appropriate use of statistics

Yes

E. Conclusions: robustness, validity and reliability

Overall, the conclusions are well supported by the presented analyses. Several points listed below

could be improved.

F. Suggested improvements

1. The main conclusion of the study is that in the majority of the tumors there is no clonal selection for genetic traits. Is there evidence for punctuated copy number evolution and thus clonal diversification early in tumor evolution?
2. Another major conclusion of this study is that phenotypic heterogeneity in CRC is mostly driven by non-hereditary changes. Is there a pattern of chromatin accessibility associated with distinct clades in the analyzed tumors that would support this conclusion? Fig. S18B shows only selected comparisons for 4 tumors. These data could be helpful in supporting the epigenetic plasticity as a source of transcriptomic heterogeneity.
3. On a similar note, are the differences in gene expression between distinct glands/locations explained by differential infiltration by immune cells or presence of other stromal cells? CIBERSORTx could help infer cell types present in each sample.

Minor points:

1. Page 3: "Pathway group 4 (low average pathway enrichment and high heterogeneity) contained two pathways, epithelial- mesenchymal transition (EMT) and angiogenesis, and was enriched for "stromal" acting pathways." Does this mean that those two pathways were classified into 4 pathway groups? If that is the case, it would be better to show more detailed pathway analysis, without the grouping.
2. Fig. S2C – not sure what this shows – should one expect a correlation? Why is there proportion instead of correlation measure? A short discussion of the differences between these classifiers would be helpful.
3. Fig. 1 Some labels (especially legends for G,H, L, M) are too small to read.
4. Phylogenetic signal – since the Authors construct this on Groups 1-3 genes and omit group 4, which is less variable in expression. Would including these genes significantly alter the association with genetic ancestry?
5. How is tumor cell purity affecting phylogenetic signal and genetic ancestry reconstruction?
6. Fig. S3 shows Phylogenetic tree and associated gene expression pattern. It looks like in 6/8 tumors the branching on the left corresponds to patterns in gene expression. Thus, the conclusion that "the expression level of most genes was not strongly related to genetic ancestry within the tumour" should be expanded to include the fact that in 6/8 tumors phylogenetic pattern is associated with expression of selected genes.
7. The fact that most CRC candidate drivers show no essentiality in DepMap dataset is very interesting. Could the authors provide a discussion of why that might be? Could it be that initiating mutations are not essential for full blown CRC? Is there any evidence of selection for genes identified as essential in DepMap within Authors' data?
8. Fig. 3H&I: The resolution of BaseScope images is very low – please provide insets of higher magnification. It would also be helpful to see matching H&E staining and example of raw image that was used to score the glands as wt or mut.

G. References

Appropriate

H. Clarity

The manuscript is well written and apart from minor points mentioned above, it is clearly presented.

Referee #3 (Remarks to the Author):

Summary:

Housham et al. conducted a spatially resolved analysis of 27 CRCs assayed for RNA-seq and WGS focusing on tumor evolution and intra-tumor heterogeneity. WGS data was used to construct sample phylogenies for samples from each tumor. Interestingly, the expression of most genes did not correlate with the phylogeny. Next, the authors tested for association of gene expression with somatic alterations in an eQTL-like framework. ~1,400 genes had expression associated with somatic variation, which was largely driven by copy number. There was little evidence that somatic SNVs drive gene expression (though it was not clear whether this was due to low power to detect such events). A dN/dS analysis was carried out and revealed little sub-clonal selection. Likewise, individual putative sub-clonal drivers were largely consistent with benign/neutral effects. Simulations were used to test for overall evidence of subclonal selection in the spatial data, identifying 8/26 such instances.

Overall, this is a very interesting study of a cutting-edge dataset, which uses multiple different approaches to evaluate important questions about selection and plasticity. The findings of little phylogenetic signal in the gene expression and little evidence of sub-clonal selection, demonstrated through multiple different analyses, are very compelling and work to the strengths of this dataset. The statistical analyses are generally rigorous and well described. Given the key findings are null/neutral (but important!), my primary concerns are whether these results are actually indicative of a null/neutral architecture or are just underpowered, as well as some lack of clarity on the various selection analyses.

Comments:

* It is interesting that CMS/CRIS classifications were generally not homogenous, indicative of ITH of molecular subtypes. However it is not clear how much of this is due to (a) biological heterogeneity versus (b) uncertainty in the classification itself. Can it be shown that the findings are consistent with (a) vs (b), or otherwise comment on the plausibility of these two explanations. What would the molecular classifications look like for a "pseudo-bulk" analysis where all measure from a tumor are merged and then classified. Finally, assuming this molecular subtype ITH is really biological, how does this fit in with the broader observation of that gene expression appears to be largely neutral and carry little phylogenetic signal? Should molecular subtype ITH also be interpreted as a largely neutral process with subtypes not conferring selective advantage?

* Conventional eQTL analyses typically regress out many latent factors of expression (Principal Components or PEER factors, see PMC3398141 and GTEx Consortium workflow for example) which accounts for a large fraction of variance and generally increases power. However, the analysis in this manuscript only included covariates for Purity and Tumor/Normal. It is thus important to investigate whether adding latent factors as covariates or regressing them out of the expression has an impact on the results. It was also not clear if MSI cases were excluded from this analysis or if a covariate for

MSI status should be added.

* There are several claims that somatic SNVs do not influence gene expression (e.g. "Thus, while most somatic SNVs do not cause a direct change in cis gene expression") but it seems just as likely that the study was underpowered to detect all such changes. Can the authors evaluate the statistical power to detect SNV → eQTL effects and either confirm that the study is sufficiently well powered to rule out such effects or clarify what effect size range they are well powered to rule out.

* It is surprising and interesting that non-coding somatic mutations had generally stronger effect sizes than coding variants. Here again, is this a consequence of different statistical power; for example, fewer non-coding somatic events in a typical enhancer than a typical gene means the effect sizes in the former must be higher to be detectable.

* It's somewhat unclear how to reconcile the dN/dS analysis finding no significant subclonal selection ("For subclonal variants, we found no evidence of subclonal selection for truncating variants ($dN/dS \sim 1$) and missense mutations had dN/dS slightly higher than 1 although the difference was non-significant, suggesting that only a small subset of putative CRC driver mutations were actually under positive selection in the growing tumours") with the simulation results that 8/26 tumors had strong evidence of subclonal selection and an additional 4 had some evidence of subclonal selection. This is made more confusing by the claim that "dN/dS analysis on the IntOGen driver gene list in neutral versus selected tumours confirmed the computational modelling results" which seems to contradict the previously quoted statement (although none of the blue lines in Figure 4O appear statistically significant, so unclear which claim is right) There's a big difference between no significant selection and 12/26 tumors having evidence of selection. Are the simulation-based results more sensitive than dN/dS or could they possibly be inflated? Are they diverging model results evidence of widespread subclonal selection on non-coding variants (which dN/dS cannot evaluate) or the presence of weak/mini-drivers?

* It would be interesting to see if the coding eQTL genes also showed evidence of selection by dN/dS analysis. As the authors note, eQTL signal alone is not evidence of selection.

* Pagel's lambda was used to quantify the phylogenetic signal, but I believe this statistic can be strongly affected by branch lengths even when the overall tree structure is similar. Can it be ruled out that lack of phylogenetic signal is not due to noise in the RNA/WGS data and is robust to the choice of scaling/normalizing the RNA-seq data?

* In addition to testing for neutrality using dN/dS, there are orthogonal approaches that evaluate the Variant Allele Fraction spectrum (see: Williams et al. Nat Genet PMID 29808029 or Salichos et al. Nat Comms PMID 32024824). Given the focus of this manuscript on apparent neutrality, it would be helpful to confirm using one of these orthogonal statistical approaches.

* It's not clear what the ATAC-seq analysis is showing and it is presented very tersely. ATAC-seq is introduced, but then the focus is on differentially expressed genes; then SCAAs are mentioned in one sentence (though the term is never actually defined) but what "analogous analysis" was actually

carried out is not explained and the findings are not explained.

* Given the overall observations of neutrality, can the authors comment on whether there is any utility to collecting additional spatial data in untreated primary CRCs?

* A 1% FDR cutoff was selected to call eQTL genes which is reasonable, but it would be helpful to also report the estimated fraction of non-null effects (i.e. the Storey π_0 statistic). It would also be of interest to know how many eQTL genes are detectable at a 10% FDR, which is the more liberal threshold typically used in conventional eQTL scans.

* The abstract mention of "eQTLs" reads somewhat confusingly as if a conventional germline eQTL analysis was carried out, I recommend modifying this to something like "somatic eQTLs" to make the distinction clear.

Author Rebuttals to Initial Comments:

Referee #1 (Remarks to the Author):

Househam, Heide et al. submit here a manuscript exploring the impact of genetic variation on transcriptomic intra-tumoral heterogeneity (tITH). The group use matched multi-regional WGS, RNAseq and ATACseq from 17 invasive carcinoma to understand the control of tITH. The manuscript forms part of a broader study using the same (and additional adenoma and normal) samples that provides greater granular detail and explanations of the 'big bang' theory previously published. In this submission, the authors initially describe the degree of tITH and cluster this according to both average expression and degree of variance. Leveraging matched WGS data the authors conclude that RNA expression is neither strongly heritable nor related to genetic ancestry. Next, the authors seek to understand the phenotypic consequence of somatic driver mutations through dN/dS analysis of their multi-regional WGS data. Interestingly, they conclude that most driver mutations have limited 'phenotypic consequence' and demonstrate limited selection. To further explore these observations the authors next apply complex computational modelling of tumour spatial growth and sub-clonal expansion. They find strong correlations between the model and their real-world data which allows them to accurately infer the evolutionary consequence of somatic driver mutations. Strikingly, neutral evolution appears dominant with only a third of tumours showing strong evidence of subclonal selection. Finally, the authors use samples with matched ATACseq and RNAseq data in tumours displaying sub-clonal selection and highlight recurrent alterations in focal adhesion pathways.

This is an impressive dataset, that for the first-time permits dissection of some truly fundamental aspects of tumour evolution and phenotype. The authors should be applauded for the scope of this study.

We thank the reviewer for their accurate precis of our study and for their exceptionally positive assessment of our work.

The manuscript is well written and figures are of a reasonable quality however some suffer from too much detail and small font size making it difficult to read. Data generated appear high quality and the authors seem to have taken much care in quality control. The highly complex narrative flow is decent through most of the manuscript although the latter section using ATACseq seems somewhat disconnected and limited in depth. Crucially, whilst this reviewer finds these data and analyses to be reliable, I find some of the conclusions and even some premises difficult to fully accept at present. This is perhaps reflective of the innovative focus of this study which has never been addressed before in this depth and for which terminology may be early in acceptance.

We appreciate this feedback on the presentation and have taken care to improve the presentation in this revision. Specifically, we have attempted clearer and more consistent use of terminology and made a clear link to our sister paper, under review at the same journal, which is focused on analysis of the ATAC-seq data (page 3, comment 1.0, text in red). We have edited the figures to improve legibility.

The discussion of evolutionary concepts, and the choice of words to describe them, is discussed in response to the reviewer's specific comments below. We are grateful for the

input about how best to communicate the ideas.

Major comments:

1. I think there is a fundamental question surrounding plasticity and variance that needs to be addressed. The authors state in their abstract “Using spatially-resolved paired whole genome and transcriptome sequencing, we find that the majority of intra-tumour variation in gene expression is not strongly heritable and therefore is plastic.” I am not fully convinced that just because a transcriptome is not heritable it has to be plastic. This reviewer would interpret ‘plastic’ as being variant and capable of change/reversion. Surely, different ‘normal’ tissues have varying degrees of plasticity – for example the gut most would regard as plastic and the CNS less so, both however have transcriptional heterogeneity (variance). Both also have non-heritable gene expression. Further, the analyses the authors are performing are based on a single time point so how can they be confident that these read outs do or don’t change over time and if they can what is the sensitivity of this? Whilst this may well be semantics, I think this issue requires clarity.

2. Similarly, in the discussion the authors state “This points towards phenotypic plasticity the ability of a cancer cell to change phenotype without underlying heritable (epi)genetic change to be a common phenomenon in CRC.”. The authors seem to be linking transcriptional variance with phenotypic plasticity in this paragraph. Is this not conflation? There is therefore an important definition that this reviewer is unclear on – are the authors defining phenotype as the degree of selection advantage within the tumour or the transcriptional pattern or both? Is it truly possible to infer such a highly dynamic process from these forms of analysis that rely on evolutionary inference based on MRS?

3. In a related point to #1 I think it would be useful to provide context to the transcriptomic analyses. There is little description of the degree of variance in normal tissue. Given how much detail is known about phenotype and plasticity in normal colonic tissue is it not possible to link the tumour findings back to normal to provide both reassurance as to the authors’ conclusions and contextual framework.

These are three thoughtful and important points that get to the core of the definition of ‘plasticity’ and how we attempt to quantitatively measure it in our dataset. We address them together below.

As the reviewer points out, we should aim to differentiate between gene expression variability or ‘transcriptional noise’ and actual plastic transcriptional programmes that can be accessed by the cell without the need to wait for a genetic or epigenetic alteration to occur.

The innovation in our study is to link the “temporal” information encoded in the evolutionary history (measured with whole genome sequencing and phylogenetics) with the current phenotype (measured foremost through gene expression). Thus, the definition of a plastic trait that we use in our study is “a phenotypic trait that varies independently of evolutionary history”. We do not include clonal selection in our definition of plasticity, but towards the end of our manuscript (see Results section “In vivo characterisation of the

epigenome and transcriptome of selected subclones”) we did analyse the transcriptional programmes active in positively selected clones (now Figures S35 & S36), and assessed whether positively selected clones are enriched for heritable gene expression programmes (now Figure S37).

We think that the reviewer’s comments refer to a definition of trait plasticity in which an organism (or cell) can change the trait within its lifetime, usually in response to some external stimulus. We could not, of course, perform direct temporal measurements of cell phenotypes in our patient samples so we could not directly assess plasticity as defined in this way.

Nevertheless, whilst the two definitions are somewhat distinct, we consider that they are also in parts interrelated – for example a particular genetic mutation could “lock in” in a particular phenotypic trait, making that trait/phenotype then fixed over time and across generations even when the external stimuli vary. Our analysis is designed to detect these “locked in” events, and we were surprised to detect so few examples of them.

In the revised manuscript we have now provided extra discussion around the definition of plasticity and what our analysis is capable of detecting. Specifically, we explicitly define phenotypic plasticity as gene expression changes that occurred independently of evolutionary history, potentially as a consequence of external stimulus from the tumour microenvironment. We added a short narrative about non-plastic traits being fixed through tumour evolution (comment 1.1, page 3, text in red).

Further, to address the reviewer’s thoughtful comments, we now perform two additional analyses: (1) we explore gene expression variability in healthy colon to assess if plasticity is specific to cancer tissues, and (2) with respect to the reviewer’s definition of plasticity of gene expression changes within a cell lifetime, we attempt to assess how the microenvironment determines gene expression programmes.

(1) Following the suggestion of the reviewer, in a new analysis we now compare the variability we observed in the tumour with the variation in expression in normal colon tissues.

We accessed a single cell RNA-seq dataset derived from healthy intestine from Elmentaite et al. 2021 (PMID: 34497389). scRNA-seq data for colon gut epithelium was downloaded from <https://www.gutcellatlas.org> and filtered for cells from the colon in ‘Healthy adults’. This left 7 donors with a mean of 5,516 cells per donor (range 1,410-16,828). Expression data was normalised with Seurat and the mean expression within each donor was calculated. First, we assessed the set of genes identified as expressed in our cohort (new methods section 2.1). Figure R1 below (new Figure S1 in the manuscript) shows that genes which were moderately/highly expressed in our cancer glands have higher average expression in the normal single cell dataset compared to all other genes. Thus, as expected, Figure R1 confirms that most genes which are expressed in tumours are also expressed in normal colon.

Figure R1. Violin plot showing that genes identified as moderately/highly expressed in colon cancer glands are more highly expressed in normal colon cells from healthy adults than all other genes (Wilcoxon $p < 1e^{-6}$).

Next, we examined the variability of expression of these genes across patients (to assess the degree of transcriptional heterogeneity), analogous to our analysis in Figure 1 of our original submission. The mean and standard deviation of each gene's expression within each donor was calculated. Then, genes were filtered and grouped according to the groups identified in Figure 1A of the original manuscript and plots were produced analogous to Figures 1B&C (see Figure R2 below and new Figure S2).

Figure R2: Mean mean expression and mean standard deviation of genes in colonic single cells of healthy adults, split by gene group.

The original analysis in Figure 1 identified four groups, the most notable of which was Group 2 which had the highest intra-tumour variability of gene expression in our cancers and was enriched with genes involved in cancer-related pathways (Figure 1B-D), making it a set of candidate plastic genes. In contrast (as shown in Figure R2B) normal colonic cells appear to

have significantly less variability in the expression of these cancer-related Group 2 genes. This indicates that there is a set of genes that show increased transcriptional diversity in tumours but it is not normally heterogeneous in normal tissues, and hence their variability in cancer does not derive from intrinsic transcriptional noise. This is suggestive that these group 2 genes could represent gene expression programmes that enable plasticity in cancers.

Contrastingly, in tumours, we observed group 1 genes showed less diversity of expression than group 2 genes, whereas in normal cells we observed the reverse. Group 1 genes are enriched for phylogenetic signal in tumours (heritable patterns of gene expression), providing examples of how tumour evolution can also simultaneously act to restrict gene expression heterogeneity.

We now discuss these data in the main text (comment 1.1-1.3 – First Part, page 3, text in red).

(2) We assessed the impact of tumour microenvironment as an external stimulus that drives gene expression (i.e. as a stimulus that potentially drives plastic switching of gene expression programmes). There were two parts to this new analysis:

First, we noted that our samples were collected from four spatially-distinct regions of the tumour (“around the clock face”), and made the assumption that the tumour microenvironment was different in each of the four tumour regions. We then tested the degree to which gene expression in our tumour glands was a consequence of microenvironment by assessing the correlation between gene expression and tumour region.

We calculated Euclidean distance matrices on the genes from gene groups 1-3 (n=8368) and performed complete hierarchical clustering for each tumour with at least 5 RNA-seq samples (n=17). The resulting dendrograms are plotted in Figure R3 (new Figure S14), with sample names, nodes and branches coloured according to region-of-origin, alongside the associated expression heatmaps. In Figure R3, some tumours were found to have perfect correlation between tumour region and gene expression (e.g. tumours C518 and C554) while others demonstrated a complete lack of association (e.g. tumours C542 and C560). The majority of tumours lay between these two extremes; some clustering of samples by region to varying degrees.

Figure R3. Hierarchically clustered dendrograms of 17 tumours based on the expression of genes from Groups 1-3 ($n=8368$), with matching heatmaps. Colours of the sample names, nodes and branches of the dendrograms correspond to the tumour region; A = red, B = blue, C = Green, D = purple.

In order to quantify the space-gene expression correlations we constructed a permutation test. For tumours with at least 10 samples ($n=11$) cophenetic distance matrices were extracted from the dendrograms plotted in Figure R3. The sum of all cophenetic distances between samples from the same tumour region was then calculated to get a metric of expression correlation with region for each tumour. To determine the significance of this metric, sample names for the cophenetic distance matrix were randomly relabelled and the mixing statistic recalculated 10,000 times, and we evaluated if the observed data was more extremely clustered than the random permutations. 4/11 tumours had significant associations between sample gene expression and region-of-origin (see Figure R4 and new Figure S15). These analyses were consistent with the notion that (unmeasured) variations in

the microenvironment between tumour regions were a determinant of gene expression heterogeneity in some, but not all, tumours.

Figure R4. Permutation test for correlations between gene expression and sample region-of-origin. Red dashed lines show the empirically observed value of the statistic, and the blue histogram the computed null distribution. P-values are FDR-adjusted.

Supporting this analysis, we next explored whether immune/stromal contamination differences explained gene expression differences between regions. The analysis is detailed in response to Reviewer 2, comment 3. We found that immune infiltration differences between samples are significantly associated with expression differences, but that only a small component of the gene expression heterogeneity is explained by non-tumour contamination ($R^2=0.219$).

From these analyses we conclude that the microenvironment does often influence gene expression.

Second, we examined whether or not glands with similar gene expression phenotypes tended to be closely genetically related. We exploited the fact that we observed physical intermixing of clones across the tumour regions, meaning that two glands that were close together in physical space were not necessarily close genetic relations. In a tumour where clonal intermixing was prevalent, if gene expression were plastic and determined by the microenvironment, then we would expect little relationship between genetic ancestry and overall gene expression patterns.

Phylogenetic relatedness was compared to expression-based clustering in the 8 tumours used for the phylogenetic signal analysis. For each tumour using matched DNA-RNA samples only, Euclidean distances were calculated based on the expression of 8368 genes in gene groups 1-3, and samples underwent complete hierarchical clustering. The resulting dendrograms (and associated expression heatmaps) were plotted against the corresponding phylogenetic trees (Figure R5 and new Figure S11).

Figure R5. Phylogenetic tree versus expression-based clustering. The dendrogram on the left of each panel is the mutation-based phylogenetic tree, while samples on the right are clustered according to gene expression. Dotted lines show matching samples and samples are coloured according to region-of-origin.

Figure R5 shows that close genetic relations do not necessarily have similar gene expression-based profiles. There are some instances when a phylogenetic clade almost exactly matches expression (e.g., region C of tumour C560) but many examples of samples clustering closely according to expression that are from distinct regions and phylogenetic clades.

There was one example of clonal intermixing in the phylogenetic tree corresponding to gene expression clustering by spatial region (region D in tumour C559 where two samples are in region A's clade, two samples are in region B's clade, and all four samples cluster closely on the dendrogram). We note that our ability to detect these arguably clear examples of gene expression plasticity in response to microenvironmental differences was limited by the number of samples where we had been able to successfully obtain high quality matched DNA and RNA.

We then performed a second analysis that compared the cohort-wide average correlation between genetic ancestry and gene expression relatedness. Specifically, we assessed dendrograms of clustered gene expression data alongside genetic phylogenies and calculated the frequency at which descendent samples on dendrograms and phylogenies respectively were found in different tumour regions. This was quantified by our “intermixing score”, previously only for computed for DNA-based phylogenetic trees (see Figure S13 of the original manuscript, now S31 in the revised manuscript, and “Intermixing scores” in methods section 4), and now also computed on the gene expression dendrograms (shown above in Figure R3). We then assessed the correlation between these expression intermixing scores and the previously calculated genetic intermixing scores, finding that the two measures were not significantly correlated (Figure R6 and new Figure S13). Thus, on average gene expression heterogeneity is not well explained by genetic ancestry.

Figure R6. Scatter plot comparing intermixing scores calculated using the WGS-based phylogenetics trees versus gene expression-based hierarchically clustered dendrograms.

Taken together, these two analyses further demonstrate a general lack of genetic control of gene expression in colorectal cancers.

In summary then, these new analyses indicate a frequent influence of the tumour microenvironment on gene expression patterns, and provide further demonstration that genetic ancestry is only infrequently associated with gene expression. We think these data provide good support to our conclusion that gene expression programmes are frequently plastic in CRC, and enabling adaption to heterogeneous microenvironments.

We now present the new analysis in the revised version of the manuscript and clarify the distinction between plasticity and transcriptional noise throughout the paper (page 3 & 8, comment 1.1-1.3, text in red and new Figures S1, S2, S11 and S13-15). In addition to the above analyses, we now also provide a plot showing the WGS-based phylogenetic trees and expression-based hierarchically clustered dendrograms side-by-side for all tumours with at least 5 samples (n=17). See Figure R7 and new Figure S12.

Figure R7. WGS-based phylogenetic trees plotted side-by-side with expression-based hierarchically clustered dendrograms for the 17 tumours with at least 5 RNA-seq samples. Dotted lines show matching samples and samples are coloured according to region-of-origin.

4. The ATACseq portion of the manuscript appears incomplete in analysis and doesn't integrate well into the manuscript. The authors have understandably mainly chosen to focus on those tumour subclones showing a selection advantage but what are seen in those that do not? Indeed, more generally what is the epigenetic contribution to tITH, at least in regards to chromatin accessibility? Surely, this could be addressed using data the authors

have generated?

We do appreciate the point raised by this reviewer. The reason why we did not focus more on the ATACseq data is because that is the central topic of our sister paper, which is under review in the same journal (Heide, Househam et al.). We now highlight this in the revised manuscript – see page 3, comment 1.4, text in red.

Minor comments:

1. CMS and CRIS calling p3. These are quite striking results that call into question the entire validity of these sub-types. If this is true then how is it that these subtypes are capable of defining clinically relevant groups based on single biopsies? Also, why do the authors feel that CRIS and CMS don't correlate well when this has previously been shown not to be the case?

We note that intra-tumour heterogeneity of CMS and CRIS subtypes has previously been reported in the literature (Dunne et al. 2016; PMID: 27151745, Sirinukunwattana et al. 2021; PMID: 32690604), as has the discordance between CRIS and CMS classifications (Dunne et al. 2017; PMID: 28561046, Isella et al. 2017; PMID: 28561063). We have extended our discussion about this point in the revised version of the manuscript (page 4, comment 1.5, text in red).

In terms of defining clinically-relevant groups, we note that transcriptional subtypes correlate strongly with information provided by existing molecular features already in the clinic, such as immune infiltration, RAS/RAF pathway mutations and microsatellite instability (Dientsmann et al. 2017; PMID: 28050011). We suspect such features are typically "clonal" in a tumour (certainly *KRAS* mutations and MSI status are almost always clonal), and so the gene expression subtypes can be useful for these features.

The most obvious mechanistic explanation of subtype heterogeneity and disagreement between classification schemes is varying levels of stromal/immune cell contamination. It is well known that a strong transcriptional determinant of the subtype is the stromal cells, which is variable between tumour regions (quantified in our data in response to reviewer 2 point 3). While CRIS subtyping is based only on the cancer transcriptional component (Isella et al. 2017; PMID: 28561063), we suggest that the expression of these genes is also influenced by microenvironmental cells. Indeed, we note that the analysis shown in original Figure S2C (now Figure S4C) does reveal correspondence between CMS and CRIS but, as reported, this correspondence was weak. Previous studies that compared CMS to CRIS also found limited overlap, driven by strong CMS dependence on stromal infiltration (see Figure 4G of Isella et al. 2017; PMID: 28561063 and Dunne et al. 2017; PMID: 28561046).

2. p6. The negative correlations of SCNAs with expression change (10%) is interesting. The authors speculate that this may be related to dominant-negative activity. This requires further attention and confirmation to ensure that this is not a technical error and better understand this result.

We agree that this was an interesting and unexpected finding. We observed a significant negative correlation between copy number and gene expression in our multivariate analysis for 81 genes (81/1163 genes with a significant association between expression and copy number; 6.96%). To check for a technical error, we ran a new univariate analysis (i.e., Expression \sim CNA) on the 81 negative CNA association genes. The CNA coefficient was found to always be below zero and a significant FDR-adjusted p-value was found for 63/81 genes (77.8%).

A possible mechanism here could be that these genes act as transcriptional repressors of themselves in a negative feedback loop. Unfortunately, no gene list of transcriptional repressors has been curated and so this hypothesis could not be tested. We note that this idea is consistent with research in cell lines which found that single-chromosomal gains can function as tumour suppressors (Sheltzer et al. 2017; PMID: 28089890).

We appreciate our analysis here is fairly superficial. We would prefer to leave the data in the manuscript, but could remove it if the reviewer feels strongly.

3. The DepMap analysis (p8) is quite superficial and this reviewer is not certain that linking this quite artificial cell line derived dataset to their real world data is helpful.

We agree that the field should not over-interpret the link between cell line data and patient sample biology. Nevertheless, we do think that DepMap provides a rare example of functional data about the consequence of cancer gene manipulation, which makes a useful reference point for our assessment of driver mutation “driverness”.

It is worth noting that the CRISPR screens conducted by DepMap were designed to identify negatively selected or lethal loss of function (LOF) alterations. Specifically due to the short expansion period of cells (approx. 10 days) the power to detect positive selection of tumour suppressor gene (TSG) LOF is lower compared to negatively selected mutations. Further, most of the screened cell lines underlying the considered DepMap dataset have mutated/non-functional TSGs, whose knock-down does not provide further selective advantage. In addition, heterogenous on-target efficacy of gene targeting single-guide RNAs might also have (cryptically) influenced results. Nevertheless, it remains the case that the most essential of oncogenes in DepMap, namely KRAS and PIK3CA, corresponded to subclonal expansions in our cohort, with KRAS in tumour C539 as the clearest example.

We further expand on this point in the revised version of the manuscript (page 14, comment 1.7, text in red).

4. I cannot fully review the validity of the computational model used in this study as it is outside of my area of expertise however, I would be interested to know why a ΔAIC_i of >4 was used? Further how does the model take into account cooperative as well as more accepted competitive interactions between subclones?

The value of ΔAIC_i used is consistent with accepted interpretation: conventionally, models with a $\Delta AIC_i < 2$ are considered to have substantial support, and those with a $\Delta AIC_i < 4$ to have strong support (Burnham & Anderson 2004). In line with this we chose a $\Delta AIC_i < 4$ to

identify cases in which data supported more than one model. We now better clarify the use of $\Delta AIC_i < 4$ in the revised version of the manuscript (page 12, comment 1.8a, text in red).

Our model does not explicitly consider cooperative interactions between subclones. While extremely interesting, such interactions are significantly more complex to model than the (relatively simple) framework we have presented. We note that our current model (of competitive evolution) can reproduce the observed data with surprisingly high accuracy (e.g., Figure 4A-L). We note that our use of the AIC regularization scheme, part of the statistical model fitting procedure, would strongly penalize against more complex models and prefer simpler models (unless the complex model explained the data enormously better). In other words, it is very unlikely that our phylogenetic tree-based inference scheme can access sufficient information in our data to discern cooperative effects between subclones. We describe these modelling limitations in the revised version of the manuscript (page 12, comment 1.8b, text in red).

5. I am uncertain why the adenoma samples present in the wider EPICC study have not been included here. Surely it is important to define whether the evolutionary principles proposed in this submission are conserved (or not) at an earlier time in tumour development?

We agree that an equivalent analysis of adenomas would be interesting but unfortunately (with the exception of the one patient with the large, advanced adenoma – C516) RNA-sequencing was not carried out on the adenoma samples. We clarify this in the revised version of the manuscript (page 16, comment 1.9, text in red). In future work we hope to be able to collect a large cohort of adenomas and then to explore genotype-phenotype in premalignancy.

6. Does cytosol specific RNAseq bias towards specific transcripts compared to whole cell RNAseq?

To specifically address this comment in our data, we compared the expression values of normal colorectal tissues generated as part of the TCGA project (whole cell-seq) with those generated by our single crypt isolation protocol (cytosol-seq). As shown in Figure R8 below, the correlation of expression values between the two cohorts is very high ($R > 0.85$).

Notably for genes with lower expression (i.e., $CPM < 2$) in the TCGA cohort some deviations appear to exist. We argue this most likely stems from the reduced amount of stromal contamination in our data: these genes are lowly expressed in the TCGA cohort but not expressed at all in our data and so are largely derived from non-epithelial cells.

One could argue that the gene expression quantification from our study is a more relevant measure of gene expression than whole-cell data, as translation takes place outside the nucleus. We note that the normal tissue reference we used was generated from the same patients we collected tumours from, and was profiled with the same exact protocol, so there is no risk of technical bias here. Nevertheless, we note that whole single cell sequencing and nucleus-only sequencing produces highly comparable data, suggesting that there is little quantitative difference between cytosolic and nuclear RNA transcripts (Ding et al. 2020; PMID: 32341560).

Figure R8. Correlation between gene expression in TCGA normal colon samples vs our normal samples.

This analysis and associated figure are included in our related paper under review in the same journal, which discusses the methodology in more detail (“Single gland multi-omics” and Figure S2 in Heide, Househam et al. 2022).

7. Does the sampling of purely glandular structures bias the transcriptomic analysis? How reflective are the glandular samples of more traditionally sampled bulks?

Please see the answer to the preceding comment, which addresses the issues of both cytosol-specific and gland-based RNA-seq data.

8. Some of the supplementary figures have overlapping gene annotations which should be tidied up.

We thank for the reviewer for pointing out this formatting error (in original Figure S10). We have now corrected this in new Figure S24.

Referee #2 (Remarks to the Author):

A. Summary of key results

The manuscript leverages the single-gland multiregion profiling of colorectal cancer by whole genome sequencing and RNAseq (generated for an accompanying manuscript) to infer the relationships between genetic and transcriptomic heterogeneity. The authors reconstruct phylogenetic trees to identify genetic ancestry and find that gene expression patterns do not strictly follow the genetic clustering of samples. Aiming to identify the true genetic drivers of CRC, the authors perform eQTL analysis and confirm that most mutations do not alter gene expression, while some of the subclonal genetic variants can affect gene expression. Interestingly, driver mutations identified in this study are predominantly clonal and very few cases of positive selection are identified. This can be attributed to higher-than-before depth of the data (single-gland analysis and multiregional sampling). CRCs analyzed in this study are either under “neutral” evolution or show signs of clonal selection. The Authors generate a mathematical model recapitulating the observed phylogenetic trees, which allows to capture evolutionary consequences of subclonal driver mutations.

B. Originality and significance

The manuscript presents original data and a novel mathematical model for identification of the impact of subclonal driver mutations.

C. Data and methodology

The impressive depth of data generated for this manuscript allows for a more detailed characterization of clonal events in CRC. The quality of data and presentation is high and meets the standards of the journal.

D. Appropriate use of statistics

Yes

E. Conclusions: robustness, validity and reliability

Overall, the conclusions are well supported by the presented analyses. Several points listed below could be improved.

We thank the reviewer for their positive opinion of our work and are glad that they see value and novelty in our dataset and analyses.

F. Suggested improvements

1. The main conclusion of the study is that in the majority of the tumors there is no clonal selection for genetic traits. Is there evidence for punctuated copy number evolution and thus clonal diversification early in tumor evolution?

This is an interesting aspect of tumour evolution that has previously been studied in colorectal cancer (Sottoriva et al. 2015; PMID: 25665006, Cross et al. 2018; PMID: 30177804). In the sister manuscript where we report the copy number profiles for each sample (Supplementary Figure S3 in Heide, Househam et al.), we do observe the same patterns of copy number as in previous analyses, with cancer genomes characterised by high aneuploidy, in stark contrast with adenoma genomes that showed much fewer copy number alterations. At the same time, the per-gland copy number profiles were not as heterogeneous, supporting a Big Bang model of tumour growth characterised by an early punctuated transformation, possibly at the transition of adenoma to carcinoma, followed by

less radical chromosomal instability. We have explored copy number alteration evolution in CRC extensively in another study (preprint, doi:10.1101/2020.03.26.007138) and so decided not to revisit this topic in this manuscript.

2. Another major conclusion of this study is that phenotypic heterogeneity in CRC is mostly driven by non-hereditary changes. Is there a pattern of chromatin accessibility associated with distinct clades in the analyzed tumors that would support this conclusion? Fig. S18B shows only selected comparisons for 4 tumors. These data could be helpful in supporting the epigenetic plasticity as a source of transcriptomic heterogeneity.

This is a great point. In the sister manuscript in revision in the same journal we have investigated the patterns of chromatin variability in the same set of cancers and found that most recurrent chromatin accessibility alterations were clonal in the tumour (Heide, Househam et al., Figure 3). However, we did find a correlation between genetic and epigenetic divergence, indicating that chromatin accessibility is, at least in part, heritable and so likely evolving in CRCs (Heide, Househam et al., Figure S33 and S34). Some of the SNVs identified in the eQTL analysis in this manuscript were indeed associated with a change in chromatin accessibility at the locus (Heide, Househam et al., Figure S25). To avoid replication, we refer to the sister paper to for these detailed analyses.

On a more philosophical note, we view chromatin changes as more likely to be *permissive* of gene expression change, rather than directly *causative* of the change. This is because chromatin has to be accessible in order for RNA to be transcribed from DNA, but transcription only happens if the DNA is unmethylated, if transcription factors bind appropriately, and if the replication machinery is in-place. These latter three factors are all regulated by a multitude of genetic, transcriptional and post-transcriptional factors. Consequently gene expression does not necessarily follow when a somatic chromatin accessibility alteration (SCAA) occurs, and indeed in (Heide, Househam et al.) we do not observe a perfect correlation between SCAAs and gene expression.

3. On a similar note, are the differences in gene expression between distinct glands/locations explained by differential infiltration by immune cells or presence of other stromal cells? CIBERSORTx could help infer cell types present in each sample.

The gene filtering procedure (as described in new methods section 2.1) removes genes that were negatively correlated with purity to focus on epithelial cell intrinsic gene expression that could be determined by genetic changes in those same cells.

However, it is still an interesting question whether immune/stromal infiltration explains expression differences between regions. As CIBERSORTx (Steen et al. 2020; PMID: 31960376) was suggested by the reviewer, we used it to assess how immune infiltration impacted differences in expression between samples, specifically using the LM22 signature file comprising of 22 immune cell types (Newman et al. 2015; PMID: 25822800). Firstly, Euclidean distances between the vector of gene expression from pairs of samples in the same tumour were calculated based on the expression of the 8,368 genes used in the phylogenetic signal analysis. Euclidean distances were also calculated based on the absolute scores from CIBERSORTx (note that CIBERSORTx was run using all genes, i.e. including

stromally-derived genes). The Euclidean distance between two samples based on overall gene expression was found to significantly correlate with the distance based on CIBERSORTx estimates (Figure R9A; $p=3.46E^{-140}$, $R^2=0.219$). Therefore there is a significant but weak association between immune and overall expression differences: almost 80% of the variance in gene expression based sample pairwise expression comparisons is not explained by CIBERSORTx-measured stromal/immune cell composition.

Divergence in gene expression tended to be higher for between tumour-region comparisons as opposed to within-region comparisons for both epithelial genes and CIBERSORTX data: in other words the tumour microenvironment was more similar within than between tumour pieces. This trend was found to be significant for both metrics but was stronger for genes expressed in the epithelium (Figure R9B vs R9C).

Figure R9. Analysis of the impact of immune infiltration on expression differences. (A) Scatter plot showing the correlation of Euclidean distances between samples when

calculated from the expression of genes and CIBERSORTx estimates respectively. Each dot is a within-tumour sample pair, dots are coloured by pair type (i.e., within-region/between-region). **(B)** Violin plot showing the Euclidean distance of pairwise samples based on gene expression, split by pair type **(C)** Violin plot showing Euclidean distances of pairwise samples based on CIBERSORTx estimates, split by pair type.

This analysis is now shown in Figure S16 and discussed in the revised version of the manuscript (page 7, comment 2.3, text in red). This, together with the analysis detailed in Figures R3-7 demonstrate how expression heterogeneity is not stringently defined by either genetic ancestry or microenvironmental contexts.

A caveat to consider for these analyses is the accuracy of immune/stromal deconvolution. To explore this, we ran multiple deconvolutions tools on this cohort (note all genes were used for this analysis) and compared their overall contamination/purity/infiltration values (Figure R10).

Figure R10. Pairs plot demonstrating correlation between different deconvolution tools and DNA-based purity estimated by Sequenza (Favero et al., 2015; PMID: 25319062) for matched samples (n=157). Bottom left panels show spearman correlation coefficients and associated significance values (adjusted for multiple testing). The scores used from each tool were as followed; EPIC (Gfeller et al. 2020; PMID: 32124324) = “otherCells”, quanTlseq (Plattner et al. 2020; PMID: 32178821) = “Other”, CIBERSORTx (Steen et al. 2020; PMID: 31960376) = Absolute summary score, xCell (Aran et al. 2017; PMID: 29141660) = “MicroenvironmentScore”, ESTIMATE (Yoshihara et al. 2013; PMID: 24113773) = “ESTIMATE” score.

While most tools showed a significant correlation between their summary scores, the strength of this correlation was relatively low. Additionally, only a few tools displayed a significant correlation with ground truth purity for matched DNA-seq samples (Figure R10; bottom row). The low strengths of these correlations seem particularly surprising considering that this is just based on the summary score for each tool and the individual immune/stromal cell type estimations are not considered. Figure R10 therefore reveals the inherent uncertainty in RNAseq deconvolution, which we briefly emphasise when discussing Figure R9 (i.e. Figure S16) in the main manuscript (comment 2.3, text in red, page 7).

Minor points:

1. Page 3: “Pathway group 4 (low average pathway enrichment and high heterogeneity) contained two pathways, epithelial- mesenchymal transition (EMT) and angiogenesis, and was enriched for “stromal” acting pathways.” Does this mean that those two pathways were classified into 4 pathway groups? If that is the case, it would be better to show more detailed pathway analysis, without the grouping.

We apologise for the confusing language of this sentence. The pathways used in this analysis can be put into 5 distinct classes according to function (Immune/Oncogenic/Cellular Stress/Stromal/Other) – as originally implemented in Jiménez-Sánchez et al. 2020 (PMID: 32483290). Three pathways (ANGIOGENESIS, APICAL_JUNCTION and EPITHELIAL_MESENCHYMAL_TRANSITION) comprise the “Stromal” class. In our analysis of pathway clustering based on mean enrichment and variation in enrichment we identified four groups, where “Group4” comprises of just EPITHELIAL_MESENCHYMAL_TRANSITION and ANGIOGENESIS pathways. Via a Fisher’s exact test, Group4 pathways are enriched for Stromal pathways ($p=2.7e^{-03}$). We have re-worded this sentence to be clearer (page 4, comment 2.4, text in red).

2. Fig. S2C – not sure what this shows – should one expect a correlation? Why is there proportion instead of correlation measure? A short discussion of the differences between these classifiers would be helpful.

We examined the relationship between CRIS and CMS classifications and recapitulated the findings of Isella et al. 2017 (PMID: 28561063, specifically Figure 4G) and others (Dunne et al. 2017; PMID: 28561046) where limited associations were found. Figure S2C (now Figure S4C in the revised manuscript) specifically shows the proportion of samples from each tumour that were classified as each subtype. The language of “correlation” in the main text is therefore mis-leading and has been changed (page 4, comment 2.5, text in red).

A strong transcriptional determinant of the subtype in CMS is the stromal cells, which is variable between tumour regions (quantified in our data in comment 2.3 above). While CRIS subtyping is based only on the cancer transcriptional component (Isella et al. 2017; PMID: 28561063), we suggest that the expression of these genes is also influenced by microenvironmental cells.

3. Fig. 1 Some labels (especially legends for G,H, L, M) are too small to read.

We thank the reviewer for pointing this out. Label sizes have been increased in the updated Figure 1.

4. Phylogenetic signal – since the Authors construct this on Groups 1-3 genes and omit group 4, which is less variable in expression. Would including these genes significantly alter the association with genetic ancestry?

Group 4 genes had a low mean expression and relatively low intra-tumour heterogeneity of expression. Group 4 was also the only group to not be significantly enriched for a relevant meta-pathway (see Figure 1D). To avoid the identification of spurious associations (and associated decrease in power for the rest of our analysis due to multiple testing) we therefore excluded these genes from the analysis.

Inclusion of Group 4 genes has a negligible impact on the proportion of phylogenetic genes per tumour (a median of 56 genes in group 4 have significant phylogenetic signal). We clarify this point in the revised version of the manuscript (page 6, comment 2.7, text in red).

5. How is tumor cell purity affecting phylogenetic signal and genetic ancestry reconstruction?

This is an important point. Purity is accounted for during genetic ancestry reconstruction. The reconstruction of the ‘core’ WGS based phylogeny was based on the presence/absence of called mutations in glands, where a mutation was called if the cancer cell fraction (which depends on the purity) was above 0.25 (see methods section 4 for details). The ML method we used to place low-pass WGS samples on the reconstructed tree also takes the purity of samples into account and estimates it from the SNV data. We found the estimated purity to be strongly correlated with those we derived from orthogonal analysis of copy-number alterations (Figure R11A). Leave-one-out validation using subsampled deep WGS samples also confirmed the ability to derive correct estimates across a wide range of purities (Figure R11B).

Figure R11. Purity estimation of LP-WGS samples during tree reconstruction. A) MLE purity estimates vs CNA derived estimates of LP-WGS sample purity show a strong correlation between both measures. B) Leave-one-out validation with subsampled deep WGS samples demonstrates the ability of the MLE LP-WGS addition method to recover the sample purity from LP-WGS SNV data across a wide range of purity and coverage values.

To assess how phylogenetic signal is affected by purity, the analysis was rerun with purity-corrected expression. Briefly, the coefficients of how purity determines gene expression had already been calculated during gene filtering (i.e., the coefficient of Purity in $Exp \sim Purity$ regression for all DNA matched samples; see new methods section 2.1) and samples used for phylogenetic analysis had matched DNA samples, allowing the use of accurate purity values. Each gene's expression (first normalised by DESeq2's variance stabilising transformation) was then normalised with the following equation (1).

$$Exp_{pur} = Exp_{vst} + \frac{Purity\ coefficient}{Sample\ purity} \quad (1)$$

Phylogenetic signal analysis was then undertaken with purity-corrected expression, the results of which are shown in Figure R12.

Figure R12. Phylogenetic signal in colorectal cancer with purity-adjusted expression. (A) and (B) Phylogenetic trees and heatmaps of genes with significantly high phylogenetic signal for tumours C552 and C554 respectively. **(C)** Genes with recurrent phylogenetic signal across tumours, genes shown were found to have significantly high phylogenetic signal in at least three tumours **(D)** Results of chi-squared test showing whether gene groups were enriched for phylogenetic genes (genes with phylogenetic signal in at least one tumour). **(E)** Bar chart showing the impact of RNA-seq normalisation method on the percent of genes that were

found to be phylogenetic per tumour. (F) Clustered bar chart showing in each tumour, of the genes found to be phylogenetic in at least one analysis, the percentage that were found in both, only Original or only Purity respectively.

When comparing Figures R12A-D to Figure 1G-J the impact of purity appears to be minimal. For instance, the heatmaps of Fig R12A&B display similar patterns and clustering to their Figure 1 counterparts, and the number of recurrent phylogenetic genes is only slightly increased (n=84 vs 61 genes). Additionally, Figure R12E shows that the number of phylogenetic genes per tumour does not greatly differ between the two analyses in that some tumours, such as C544, C551 and C554, are increased for the purity analyses while others are decreased. Interestingly, the proportion of genes found to be phylogenetic is relatively high for most tumours (Fig R12F) indicating that correcting for purity has only a small impact on phylogenetic signal analysis. We report this analysis in new Figure S8 and clarify these comparisons in the revised version of the manuscript (page 6, comment 2.8, text in red).

6. Fig. S3 shows Phylogenetic tree and associated gene expression pattern. It looks like in 6/8 tumors the branching on the left corresponds to patterns in gene expression. Thus, the conclusion that “the expression level of most genes was not strongly related to genetic ancestry within the tumour” should be expanded to include the fact that in 6/8 tumors phylogenetic pattern is associated with expression of selected genes.

We apologise for the lack of clarity on this point. This is now Figure S5 in the revised version of the manuscript and it shows, for each tumour, only those genes that were found to have phylogenetic signal. Therefore, all tumours (rather than just 6/8) show a phylogenetic pattern in “selected genes”. Our argument is that the number of genes which show significant phylogenetic signal in each tumour is a small proportion of the expressed genes analysed (median 2%, range 0.8-27.9%) or 166 genes (range 67-2335 genes) out of 8368 genes analysed.

X-axis labels have been added to the heatmaps of Fig S5 (as well as Fig 1G&H) to make clear how many genes are plotted for each tumour (corresponding to the number of genes that were significantly phylogenetic for each tumour).

7. The fact that most CRC candidate drivers show no essentiality in DepMap dataset is very interesting. Could the authors provide a discussion of why that might be? Could it be that initiating mutations are not essential for full blown CRC? Is there any evidence of selection for genes identified as essential in DepMap within Authors’ data?

This is indeed something that interested us too. On the divergence of reported driver genes versus those that validate in functional assays we can only speculate. However, we can’t exclude that many reported putative driver genes are either spurious (e.g., they are highly mutable and hence recurrent but not functional) or are drivers only in specific contexts and neutral in others or become no longer necessary once the tumour is established. Relatedly, some of these drivers may have occurred and been selected in normal tissue before tumourigenesis (Martincorena et al. 2019; PMID: 31138277). Finally, and perhaps most importantly, the CRISPR screens conducted by DepMap were designed to identify negatively

selected or lethal LOF alterations, not positively selected events. Specifically due to the short expansion period of cells (approx. 10 days) the power to detect positive selection of TSG LOF is lower compared to negatively selected mutations.

However, it is indeed the case that the most essential of oncogenes in DepMap, namely KRAS and PIK3CA, are associated with subclonal expansions in our cohort, with KRAS in C539 as the best example. We further expand on this point in the revised version of the manuscript (page 14, comment 2.10, text in red).

8. Fig. 3H&I: The resolution of BaseScope images is very low – please provide insets of higher magnification. It would also be helpful to see matching H&E staining and example of raw image that was used to score the glands as wt or mut.

We apologise for this. The requested inserts have been added to the new Figure 3 in the revised manuscript. The high resolution “.tiff” files are also available as supplementary figures (new Figures S29 and S30, page 10, comment 2.11, text in red).

G. References

Appropriate

H. Clarity

The manuscript is well written and apart from minor points mentioned above, it is clearly presented.

Referee #3 (Remarks to the Author):

Summary:

Housham et al. conducted a spatially resolved analysis of 27 CRCs assayed for RNA-seq and WGS focusing on tumor evolution and intra-tumor heterogeneity. WGS data was used to construct sample phylogenies for samples from each tumor. Interestingly, the expression of most genes did not correlate with the phylogeny. Next, the authors tested for association of gene expression with somatic alterations in an eQTL-like framework. ~1,400 genes had expression associated with somatic variation, which was largely driven by copy number. There was little evidence that somatic SNVs drive gene expression (though it was not clear whether this was due to low power to detect such events). A dN/dS analysis was carried out and revealed little sub-clonal selection. Likewise, individual putative sub-clonal drivers were largely consistent with benign/neutral effects. Simulations were used to test for overall evidence of subclonal selection in the spatial data, identifying 8/26 such instances.

Overall, this is a very interesting study of a cutting-edge dataset, which uses multiple different approaches to evaluate important questions about selection and plasticity. The findings of little phylogenetic signal in the gene expression and little evidence of sub-clonal selection, demonstrated through multiple different analyses, are very compelling and work to the strengths of this dataset. The statistical analyses are generally rigorous and well described. Given the key findings are null/neutral (but important!), my primary concerns are

whether these results are actually indicative of a null/neutral architecture or are just underpowered, as well as some lack of clarity on the various selection analyses.

We thank the reviewer for describing our dataset as cutting-edge and are glad that they find our work interesting. We address their thoughtful critique in detail below.

Statistical power in our study is an important and interesting point for discussion. We had previously looked at this aspect for phylogenetic signal (original Figure S5, now Figure S9) and have now investigated power for the eQTL analysis (see below, Figure R18 and new Figure S19). Overall, we are powered to see large effects that happened early during the tumour's evolution, and we find that there are (perhaps surprisingly) few examples of this. It is, of course, possible that there are many small effects but if the effects are small then they could be argued to be of limited relevance to cancer biology. We've now made this clear in the discussion (page 15, comment 3.0, text in red).

Comments:

* It is interesting that CMS/CRIS classifications were generally not homogenous, indicative of ITH of molecular subtypes. However it is not clear how much of this is due to (a) biological heterogeneity versus (b) uncertainty in the classification itself. Can it be shown that the findings are consistent with (a) vs (b), or otherwise comment on the plausibility of these two explanations. What would the molecular classifications look like for a "pseudo-bulk" analysis where all measure from a tumor are merged and then classified. Finally, assuming this molecular subtype ITH is really biological, how does this fit in with the broader observation of that gene expression appears to be largely neutral and carry little phylogenetic signal? Should molecular subtype ITH also be interpreted as a largely neutral process with subtypes not conferring selective advantage?

The reviewer has highlighted the interesting finding of CMS/CRIS heterogeneity. We note that subtype heterogeneity in CRC, particularly for CMS, has previously been shown. For instance, Alderdice et al. 2018 (PMID: 29412457) found heterogeneity for CMS but not CRIS and researchers originally involved in the CMS project have since acknowledged this and other biases (Fontana et al. 2019; PMID: 30796810).

To examine the impact of classification confidence, we have plotted only those samples which were confidently classified (Figure R13A&B below), revealing similar levels of intra-tumour heterogeneity to that seen in the original Figure S2A&B (now Figure S4A&B). Additionally, both classifications output a centroid distance score for each class (it was the classes with the minimal scores that were plotted in original Figure S2). Figure R13C&D shows a case study of tumour C550, revealing the centroid distance of each sample to all CMS and CRIS classifications respectively (i.e., smaller distance = greater match).

Figure R13. Extra analysis of CMS and CRIS heterogeneity. (A) CMS assignments per tumour, only samples which could be confidently classified ($FDR < 0.05$) are shown. **(B)** As for (A) but for CRIS. **(C)** Heatmap of centroid distances of each sample from CMS classes, black squares indicate the minimum (most likely) class for each sample for tumour C550, and stars represent significance of classification. **(D)** As for (C) but for CRIS.

Figure R13A&B demonstrate that CMS and CRIS classifications are heterogeneous within tumours even when only using confidently classified samples. Meanwhile, Figure R13C&D

display the degree of heterogeneity and uncertainty for each sample and class for tumour C550, underlining the fact that both classifications suffer from ITH bias due to widespread expression heterogeneity.

In terms of the interrelationship between subtypes and tumour evolution, we think that gene expression programmes, including subtype defining genes, are frequently plastic and may often be driven by response to extrinsic factors (including microenvironment composition), rather than representing a clonally selected expression programme. This is examined in detail in response to reviewer 1 point 1. Indeed, using a computational approach we find only a few examples of a positively selected subclone and none of those clones clonally express a molecular subtype distinct to the rest of the tumour.

We now report this new analysis in new panels Figure S4E-H and discuss it in the revised version of the manuscript (page 4, comment 3.1, text in red).

* Conventional eQTL analyses typically regress out many latent factors of expression (Principal Components or PEER factors, see PMC3398141 and GTEx Consortium workflow for example) which accounts for a large fraction of variance and generally increases power. However, the analysis in this manuscript only included covariates for Purity and Tumor/Normal. It is thus important to investigate whether adding latent factors as covariates or regressing them out of the expression has an impact on the results. It was also not clear if MSI cases were excluded from this analysis or if a covariate for MSI status should be added.

It is indeed a good point that germline variation between patients impacts gene expression. To address this, we compiled a binary matrix with the germline SNPs of the 19 patients involved in the eQTL analysis. We then performed principal component analysis to determine if there was any patient clustering by germline variation (see Figure R14).

Figure R14. Plot of the top two principal components from PCA analysis of germline SNPs in patients used in the eQTL analysis. Patients are coloured according to MSI status.

Figure R14 shows patients C516, C548 and C549 to be outliers in terms of germline variation compared to all other patients, although it should be noted that these top two principal components only account for 16.6% of the explained variation. The tumours are also coloured by MSI status, showing that principal component 1 slightly separates MSS from MSI tumours. Principal component 3 contributed 6.2% of explained variation and separated only C516 from the rest of the cohort (not shown).

Overall, it is important to note that we cannot claim causality because some effects can be driven by germline variations (and/or other unassessed trans effects). We have now made these caveats clear in the revised version of the manuscript and demonstrate the above analysis in new Figure S20 (pages 8 and 15, comment 3.2a, text in red).

In relation to MSI status, MSI tumours were included in the original eQTL analysis but this was not accounted for in the model: we apologise that this was not made clear. We therefore assessed the impact of MSI on the eQTL analysis by rerunning the analysis twice, first using only MSS tumour samples (n=149 across 15 tumours) and second using only MSI tumour samples (n=18 across 3 tumours). Out of the 29,949 mutation-gene combinations analysed in the original analysis, only two are present in both new analyses (i.e., only two mutations are shared in both an MSS and MSI tumour – *KRAS* G13D in MSI C548 and MSS

C561, and an enhancer mutation for *CAPG* in MSI C516 and MSS C550) and neither are significant for the mutation in either new analysis.

We then tested if mutations had different associations with *cis* gene expression between MSS and MSI cohorts. Given the large difference in sample size and therefore power, in order to make the two analyses comparable, only mutations with very large (>1.5) effect sizes were considered. The absolute mutation effect sizes of 73 eQTLs from the MSS analysis were therefore compared to 293 eQTLs from the MSI analysis.

Figure R15. QQ plot comparing the quantiles of MSS and MSI significant mutation eQTL effect sizes.

Figure R15 (and new Figure S21), showing the QQ-plot comparing these two datasets, reveals there is a difference in the distribution of effect sizes of significant eQTLs between the MSS and MSI analyses. Specifically, there are a higher proportion of MSI eQTLs at very large effect sizes in comparison to the MSS analysis. This is interesting as it suggests a difference in the genetic control of gene expression between MSS and MSI tumours and has been added to the revised version of the manuscript (comment 3.2b, page 8, text in red).

We next sought to determine the effect of adding MSI as a cofactor in the eQTL analysis (see Figure R16 and new Figure S22).

Figure R16. Genetic control of expression with eQTL analysis with MSI status added as a cofactor. (A) The number of genes with significant models for each data type. **(B)** The distribution of regression coefficients (effect sizes) for each data type. **(C)** and **(D)** Volcano plots highlighting selected genes that were significant for CNA and Mut eQTLs respectively. **(E)** In comparison to non-synonymous SNVs (NS), enhancer (Enh) mutations tended to have large effect sizes and a higher proportion of positive effect sizes. **(F)** The proportion of subclonal mutations that were associated with detectable changes in cis gene expression tended to be lower than for clonal eQTL mutations. **(G)** Visualisation of Fisher's exact tests showing that gene-mutation combinations were more likely to be eQTLs if they were associated with recurrent phylogenetic genes (genes found to be phylogenetic in at least 3 tumours) for subclonal mutations and that this was not significant for clonal mutations.

When compared with Figure 2A-G in the original manuscript Figure R16 reveals that adding MSI status as a cofactor in the eQTL analysis has a minor effect on the results. Notably, there was a small decrease in the number of significant eQTL genes (Figure R16A&B), non-coding enhancers were no longer significantly associated with increases in expression ($p=0.08$; Figure R16E) and subclonal mutations were no longer more likely to be eQTLs ($p=0.17$; Figure R16F). However, it should be noted that the direction of these effects did not change.

Finally, the distribution of R^2 values were compared between the original analysis (without MSI as a covariate) and the analysis with MSI as a covariate. Figure R17 (and new Figure S23) shows that, for models that were significant in both analyses, the R^2 values were highly correlated ($p < 1e^{-16}$, $R^2=0.855$).

Figure R17. Scatter plot showing correlation of R^2 values for models significant in both the original eQTL analysis and the analysis with MSI included as a covariate.

It is worth noting that the R^2 values tend to be higher for the analysis with MSI, and this was found to be significant (paired Wilcoxon signed rank test; $p=1.071e^{-241}$). Therefore, including MSI as a covariate marginally increases the amount of variance explained by each model, but the R^2 values are very highly correlated with the original analysis.

Overall, our analyses show some evidence of differences in genetic control of gene expression between MSS and MSI tumours, but the main finding of a lack of genetic control applies to both subtypes. The above analyses and figures have been added to the revised version of the manuscript (comment 3.2b, page 8, text in red).

* There are several claims that somatic SNVs do not influence gene expression (e.g. "Thus,

while most somatic SNVs do not cause a direct change in cis gene expression") but it seems just as likely that the study was underpowered to detect all such changes. Can the authors evaluate the statistical power to detect SNV → eQTL effects and either confirm that the study is sufficiently well powered to rule out such effects or clarify what effect size range they are well powered to rule out.

This is a very important point. Since expression values were normalised into a z-score, the mutation regression coefficient of each model could be used as the effect size to calculate power. Effect sizes were binned into intervals (0.5-3 by 0.5, and also 3-7) and then power was calculated using `pwr.f2.test` from the `pwr` R package, where inputs were `u`(the number of coefficients)=4, `v`(degrees of freedom)= $n-u-1$ (where $n=19$, the number of tumours), `sig.level`=0.01 and `f2` = regression coefficient of Mut in each model tested.

Figure R18. Boxplot showing post-hoc power analysis of mutation eQTL effect sizes. Effect sizes have been binned and the number of models in each bin are shown at the bottom of the plot.

Figure R18 shows the results of this post-hoc power analysis. Notably, we were powered to detect effect average sizes greater than 0.94 (standard deviation in expression change).

We now report this analysis in the revised version of the manuscript (new Figure S19) and discuss it in page 8, comment 3.3, text in red. We have also rephrased the sentence highlighted by the reviewer to emphasise we only had power to detect large effects.

* It is surprising and interesting that non-coding somatic mutations had generally stronger effect sizes than coding variants. Here again, is this a consequence of different statistical power; for example, fewer non-coding somatic events in a typical enhancer than a typical gene means the effect sizes in the former must be higher to be detectable.

We thank the reviewer for this important point. As we treat each mutation independently (i.e., we don't assess the cumulative effect of multiple mutations in the same gene/enhancer region), we believe that there are no differences in power between the coding and non-coding mutation analyses. For reference, the percentage of non-coding (enhancer) mutations we analysed that were found to be eQTLs (2.7%; 744/27742) was slightly higher than the percentage for coding mutations (2.4%; 60/2504).

* It's somewhat unclear how to reconcile the dN/dS analysis finding no significant subclonal selection ("For subclonal variants, we found no evidence of subclonal selection for truncating variants ($dN/dS \sim 1$) and missense mutations had dN/dS slightly higher than 1 although the difference was non-significant, suggesting that only a small subset of putative CRC drivers mutations were actually under positive selection in the growing tumours") with the simulation results that 8/26 tumors had strong evidence of subclonal selection and an additional 4 had some evidence of subclonal selection. This is made more confusing by the claim that "dN/dS analysis on the IntOGen driver gene list in neutral versus selected tumours confirmed the computational modelling results" which seems to contradict the previously quoted statement (although none of the blue lines in Figure 40 appear statistically significant, so unclear which claim is right) There's a big difference between no significant selection and 12/26 tumors having evidence of selection. Are the simulation-based results more sensitive than dN/dS or could they possibly be inflated? Are they diverging model results evidence of widespread subclonal selection on non-coding variants (which dN/dS cannot evaluate) or the presence of weak/mini-drivers?

Apologies for the lack of clarity here. The difference between the two analyses is the cohort-wide "ensemble" analysis (dN/dS) versus individual tumour analysis (mathematical modelling). We first tested selection with dN/dS on all subclonal truncating and missense variants across our cohort. No set was statistically significant, however whereas the truncating variants showed a point estimate of 1, the point estimate for the missense subclonal mutations was higher than one (Figure 2I). One may argue that we just do not have enough tumours to observe significance and that there are indeed some subclonal missense mutations that are under positive selection in some tumours that push dN/dS point estimate above one. We were frankly excited to find out that when we separated tumours classified as experiencing subclonal selection from tumours only with detectable neutral evolution using our spatial inference framework (which is completely orthogonal to dN/dS), the dN/dS subclonal value of missense mutations in tumours predicted to have selection indeed showed a point estimate higher than one. In contrast, the tumours predicted to be neutrally evolving confirmed a subclonal dN/dS point estimate value of 1

(Figure 4O). In the revised version of the manuscript we now clarify this point better (page 14, comment 3.4, text in red).

* It would be interesting to see if the coding eQTL genes also showed evidence of selection by dN/dS analysis. As the authors note, eQTL signal alone is not evidence of selection.

We thank the reviewer for this interesting suggestion. We have modified the dndscv package by Martincorena et al. (2017; PMID = 29056346) to allow us to calculate dN/dS estimates across eQTL sites for the coding genes and the results of this analysis are shown in Figure R19. In brief we found little difference between eQTL and non-eQTL variants, suggesting that eQTL mutations in coding genes were indeed not generally subject to strong positive selection. For clarity, we note that dN/dS is unable to capture any selection acting on non-coding eQTLs.

Figure R19. dN/dS estimates of eQTL and non-eQTL mutations.

* Pagel's lambda was used to quantify the phylogenetic signal, but I believe this statistic can be strongly effected by branch lengths even when the overall tree structure is similar. Can it be ruled out that lack of phylogenetic signal is not due to noise in the RNA/WGS data and is robust to the choice of scaling/normalizing the RNA-seq data?

It is true that phylogenetic signal, when measured by Pagel's lambda, is affected by branch length. It was for this reason that the median Lambda reported in the original submission, and associated p-value, was obtained from a set of 100 trees with varying branch lengths for the added lpWGS samples, to mitigate this issue (methods section 5, page 19).

To assess whether the lack of phylogenetic signal result is robust to the choice of RNA-seq normalisation, the gene phylogenetic signal analysis was rerun with expression normalised by standard log transformation (i.e., $\log_2(n+1)$ after normalisation for library size) as opposed to DESeq2's variance stabilising transformation (VST) normalisation. Figure R20 shows the percentage of expressed genes with phylogenetic signal for the original "VST" normalisation compared to the new "LogNorm" analysis.

Figure R20. The impact of RNA-seq normalisation method on phylogenetic signal analysis. (A) Bar chart showing the impact of RNA-seq normalisation method on the percent of genes that were found to be phylogenetic per tumour. (B) Clustered bar chart showing in each tumour, of the genes found to be phylogenetic in at least one analysis, the percentage that were found in both, only VST or only LogNorm respectively. VST=variance-stabilising transformation, LogNorm=log-normalisation.

Figure R20A demonstrates that the choice of RNA-seq normalisation method has little effect on the percentage of genes which were found to be phylogenetic in each tumour. Even though the LogNorm analysis finds notably more phylogenetic genes for C554 and C559, fewer phylogenetic genes were found for the LogNorm analysis in tumours C544, C552 and C560.

Lastly, it was checked that similar genes (i.e., not just similar numbers of genes) were being identified as phylogenetic between the two analyses (see Figure R20B). Figure R20B reveals that, for most tumours, of the genes identified as phylogenetic in at least one analysis, the majority were identified in both analyses. There was a high overlap between the genes found to be phylogenetic, indicating that the normalisation method has a negligible impact on the results.

We now report this new analysis in the revised version of the manuscript (Figures S7), and discuss in page 6, comment 3.6, text in red.

* In addition to testing for neutrality using dN/dS, there are orthogonal approaches that evaluate the Variant Allele Fraction spectrum (see: Williams et al. Nat Genet PMID 29808029 or Salichos et al. Nat Comms PMID 32024824). Given the focus of this manuscript on apparent neutrality, it would be helpful to confirm using one of these orthogonal statistical approaches.

Indeed, methods developed by us (Williams et al.) and others enable measurement selection from single samples using the VAF spectrum. As we previously showed in Williams et al. 2018 (PMID: 29808029) however, such methods work well in single large bulk samples

(which capture a large portion of the spatial composition of the tumour and hence its clonal structure) sequenced at high depth (>100X). In this study, our experimental design is different and in fact was designed to overcome the limitations of our previously developed methodologies. We aimed to map the clonal structure using gland-by-gland analyses, so not to have the confounding factors of bulk sequencing. Instead of one large sample containing many different lineages sequenced at high depth (>100X), we now opted for many spatially segregated samples containing clonal lineages (crypts/glands), sequenced at moderately lower depth (35X). We note that having a median of 8.5 WGS samples per patient implies a total coverage per patient of ~300X at whole genome level. For these types of data we had to design an entirely new method that incorporated the spatial information in our data and that could do inference on multi-region WGS. Hence the spatial cellular automaton we present in this study.

Interestingly, we note that phylogenetic trees do, in a way, capture the VAF distribution, as one could look at the clade size distribution as a measurement of the clonal architecture. Specifically, the phylogenetic trees in Figure 3 provided good estimates of time of emergence of different lineages (their Most Recent Common Ancestor – MRCA) measured relative to the total age of the tumour from trunk to leaves, calculated as the mean number of SNVs per lineage in the clade divided by mean SNVs per from root to leaf for every lineage. Moreover, for each lineage we had an estimate of its clade size because of the many samples per clade. Under neutral evolution, only early diverging lineages lead to large clade sizes, and in general clade sizes are predicted to follow a power-law $\sim 1/f$ distribution generating more and more lineages at smaller and smaller frequency as the tumour expands. In comparison, positive selection causes late emerging lineages to undergo subclonal expansion, reaching clade sizes that are larger than expected under neutral evolution. The clearest example of this phenomenon in our cohort is case C539, where a late arriving subclonal lineage in region A underwent a large expansion (Figure 3H). The relationship between MRCA time t_{MRCA} (how far into the tumour's past that a subclone was generated) and the size of the clade f (expressed as the proportion of samples within the mutated clade) should follow under neutrality as:

$$t_{MRCA} = -\frac{\ln(\pi f)}{\lambda} \quad (2)$$

Where π is the copy number of the locus and λ is the tumour growth rate, assuming exponential growth $N(t) = e^{\lambda t}$. We calculated t_{MRCA} vs f for every gene mutation, dividing into five categories:

- non-cancer gene mutations (background)
- synonymous mutations driver genes
- missense mutations in driver genes scored as functionally *benign* by PolyPhen
- missense mutations in driver genes scored as functionally deleterious by PolyPhen
- truncating mutations in driver genes

When examining our data, we found that most clones carrying driver genes mutations evolved as the predicted from neutral theory (equation 2; Figure R21). Clones with benign missense mutations as expected evolved neutrally, as did clones with truncating mutations

in genes with $dN/dS \sim 1$. A few noticeable exceptions of functionally deleterious missense mutations (which indeed showed slightly higher dN/dS) show signs of deviating from the neutral expectation, most of all KRAS G12C in C539, but also a subset of the subclonal PIK3CA mutations. Evidently, putative drivers such as FAT3 and FAT4, which are very large genes that show often benign mutations typically evolved neutrally, are unlikely to be true drivers in colorectal cancer.

Figure R21. Relative clade size vs relative age of MRCA for each putative driver variant versus background (non-cancer genes) shows that probably only RNF43 in truncating mutations, and a few variants in KRAS and PIK3CA show clear signs of deviating from the neutral expectation in red.

* It's not clear what the ATAC-seq analysis is showing and it is presented very tersely. ATAC-seq is introduced, but then the focus is on differentially expressed genes; then SCAs are mentioned in one sentence (though the term is never actually defined) but what "analogous analysis" was actually carried out is not explained and the findings are not explained.

We do appreciate the point raised by this reviewer. The reason why we did not focus more on the ATACseq data is because that is the central topic of our related manuscript that is under review in parallel the same journal. In the revised version of the manuscript we clarify that, and we now make better reference to the paper focussing on chromatin accessibility (Heide, Househam et al.) – see page 3, comment 3.8, text in red.

* Given the overall observations of neutrality, can the authors comment on whether there is any utility to collecting additional spatial data in untreated primary CRCs?

This is a good point. We did struggle to find genetic subclonal driver mutations beyond a few usual suspects. The dN/dS analysis on subclonal variants also excludes extensive weak

selection that we may have not picked up with the inference (e.g., minidrivers hypothesis). Nevertheless, we did find cases where the subclonal expansion was evident in the tree but was caused by an unknown event. We cannot exclude that such events may be numerous and require extensive spatial sampling. However, as we are now collecting prospective tissue at progression, such as liver metastatic deposits, we will be able to profile the clones that really count in terms of disease relapse and can go back to the original primary samples to see where we find those. This is the subject of future work however, as tissues are being still collected. We have extended the discussion around this topic in the revised version of the manuscript (page 16, comment 3.9, text in red).

* A 1% FDR cutoff was selected to call eQTL genes which is reasonable, but it would be helpful to also report the estimated fraction of non-null effects (i.e. the Storey π_0 statistic). It would also be of interest to know how many eQTL genes are detectable at a 10% FDR, which is the more liberal threshold typically used in conventional eQTL scans.

Using the q-value R package (<https://github.com/StoreyLab/qvalue>), the estimate of the overall proportion of true null hypotheses (π_0) was found to be 0.1007. This value is now reported in the revised version of the manuscript (page 7, comment 3.10a, text in red), and the method used to calculate it is reported in the methods (page 21, comment 3.10a, text in red).

Using a threshold of 10% FDR has only a small impact on the number of detectable eQTL genes (Figures R22 & R23). Notably the significance of results initially downstream of the FDR cut-off remain the same, with the exception of the subclonal analysis in Figure 2G (Figure R22G), where, in the 10% cut-off analysis only subclonal eQTLs are significantly associated with phylogenetic genes (as opposed to 'All' and 'Subclonal' being significant in the 1% cut-off analysis).

This analysis has now been reported in the revised manuscript (page 7, comment 3.10b, text in red). Replicas of Figures 2A-G and original Figure S7 (now Figure S17) with a 10% cut-off are available as Figure R22 and R23 respectively:

Figure R22. Genetic control of expression with eQTL analysis using 10% FDR threshold. (A) The number of genes with significant models for each data type. **(B)** The distribution of regression coefficients (effect sizes) for each data type. **(C)** and **(D)** Volcano plots highlighting selected genes that were significant for CNA and Mut eQTLs respectively. **(E)** In comparison to non-synonymous SNVs (NS), enhancer (Enh) mutations tended to have large effect sizes and a higher proportion of positive effect sizes. **(F)** The proportion of subclonal mutations that were associated with detectable changes in cis gene expression was significantly lower than for clonal eQTL mutations. **(G)** Visualisation of Fisher's exact tests showing that gene-

mutation combinations were more likely to be eQTLs if they were associated with recurrent phylogenetic genes (genes found to be phylogenetic in at least 3 tumours) for subclonal mutations and that this was not significant for clonal mutations or all mutations.

Figure R23. Frequency of associations between gene copy number alteration and change in gene expression, by direction of correlation and average locus-specific copy number. Using 10% FDR-threshold eQTL analysis results. X-axis: direction of copy number-expression correlation. Y-axis: proportion of samples across the whole cohort with specified copy number.

* The abstract mention of "eQTLs" reads somewhat confusingly as if a conventional germline eQTL analysis was carried out, I recommend modifying this to something like "somatic eQTLs" to make the distinction clear.

We thank the reviewer for pointing this out and have made the modification in the abstract so that the sentence now reads: “A somatic expression quantitative trait loci (eQTL) analysis identifies...” (page 2, comment 3.11, text in red).

In preparing the manuscript revision, we realised that the total number of significant eQTL genes was incorrectly reported in the main manuscript. This has now been corrected and does not change the interpretation of these data in any meaningful way.

Author Rebuttals to First Revision:

Referee #1 (Remarks to the Author): The authors have provided a comprehensive and thoughtful response to this reviewer's queries. I am satisfied with all their responses and appreciate their openness in acknowledging the limitations of their study, which paradoxically makes their conclusions even the more impactful. This manuscript and the accompanying sister manuscript are a great advance for the field and provide interesting avenues for further study. I think the SCNA data and discussion should stay in place (Point 2) as whilst superficial it is an interesting finding.

We thank the reviewer for their comments, and we appreciate the amount of time spent reviewing our manuscript, as well as their thoughtful insights.

Finally, I think some sort of visual schematic describing their findings would be useful, either within the manuscript or perhaps as an accompanying commentary given the scope of work and novelty. This decision can clearly be editorially decided on.

We agree that a visual schematic to describe our findings would be a good idea. Please see new Extended Data Figure 9 which depicts the main results.

Referee #2 (Remarks to the Author): The authors addressed all comments very well. This is a very thorough study and the conclusions will be of high interest to many investigators in the field.

We thank the reviewer for their useful critiques throughout the peer review process and for taking the time to review our manuscript.

Referee #3 (Remarks to the Author): The authors have addressed my major comments thoroughly, and I appreciate the detailed and thoughtful additional analyses. I have two outstanding minor comments which do not influence the main findings:

We appreciate the reviewer's positive appraisal of our responses and thank them for their time spent reviewing our manuscript in such detail.

With respect to Figure R14, my prior concern was regarding latent *non-genetic* factors of expression (cell type heterogeneity / data quality/ batch / etc). However my concerns are now largely addressed by CIBERSORTx analysis requested by Reviewer #2, which will also capture such latent heterogeneity. As for Figure R14, it's plausible that those 2-3 outliers are simply individuals of non-European ancestry. Perhaps the authors can cross-reference against self-reported race and, if it is the case, note this somewhere in their data/study description for future analyses.

We apologise for misunderstanding the reviewer's comment regarding latent factors, but we are glad that the CIBERSORTx analysis addressed their concerns.

For Figure R14 (now Figure S16 in the edited manuscript), the reviewer makes a good point that the genetic ancestry of our patients could explain the afore mentioned outliers. Unfortunately, we did not collect self-reported race data for our patients and so can't test

this hypothesis. We now note this as a potential confounder in the edited version of the manuscript (comment 1, pages 6 & 11, text in red).

With respect to Figure R15 (new Figure S21), the authors say "Specifically, there are a higher proportion of MSI eQTLs at very large effect sizes in comparison to the MSS analysis" but I believe the figure is showing the opposite: for example, a very large MSS effect size (x-axis) of 6 corresponding to a less large MSI effect size (y-axis) of ~4.5. Please check that the axis labels match the text or perhaps clarify the phrasing in the text.

We thank the reviewer for correctly pointing out mistake in Figure R15 (now Figure S17 in the edited manuscript). The axis labels are right, so the correct interpretation is that there are a higher proportion of **MSS** eQTLs at very large effect sizes in comparison to the **MSI** analysis. This also makes more sense to us as MSI tumours should be more likely to have many low-effect mutations due to their defective mismatch repair mechanisms. The corresponding point in the text now reads: "SNVs in **MSS** tumours were more frequently associated with large effects on gene expression" (comment 2, page 6, text in red).

Please also note that an error in the code to make Figure R15 (now Figure S17) has been rectified, leading to a slight adjustment in the figure that has no bearing on the interpretation.

Additional note from authors

Figure S16 in the original revised manuscript (now Figure S12 in the edited manuscript) was generated using analysis that erroneously included Group 4 genes. The new figure now only includes genes from Groups 1-3, which has had a negligible impact on the statistics for panels A and B (e.g. panel A R^2 was 0.219, it is now 0.210)